# PIBF1 regulates trophoblast syncytialization and promotes cardiovascular development

Jong Geol Lee [1,2,19,20], Jung-Min Yon[1,3,20], Globinna Kim[1,3], Seul-Gi Lee[4], C-Yoon Kim[5], Seung-A Cheong[1], Hyun-Yi Kim[6], Jiyoung Yu[1], Kyunggon Kim[1,7], Young Hoon Sung [1,3], Hyun Ju Yoo [1,7], Dong-Cheol Woo[1,8], Jin Kyung Rho [1,9], Chang Hoon Ha [1,9], Chan-Gi Pack [1,8], Seak Hee Oh[10], Joon Seo Lim[1], Yu Mi Han[11], Eui-Ju Hong[12], Je Kyung Seong [2,13], Han-Woong Lee [14], Sang-Wook Lee[2,15], Ki-Up Lee[1,9], Chong Jai Kim[1,16], Sang-Yoon Nam[17], You Sook Cho [1,18] ✉ & In-Jeoung Baek [1,2,3] ✉

Proper placental development in early pregnancy ensures a positive outcome later on. The developmental relationship between the placenta and embryonic organs, such as the heart, is crucial for a normal pregnancy. However, the mechanism through which the placenta influences the development of embryonic organs remains unclear. Trophoblasts fuse to form multinucleated syncytiotrophoblasts (SynT), which primarily make up the placental materno-fetal interface. We discovered that endogenous progesterone immunomodulatory binding factor 1 (PIBF1) is vital for trophoblast differentiation and fusion into SynT in humans and mice. PIBF1 facilitates communication between SynT and adjacent vascular cells, promoting vascular network development in the primary placenta. This process affected the early development of the embryonic cardiovascular system in mice. Moreover, in vitro experiments showed that PIBF1 promotes the development of cardiovascular characteristics in heart organoids. Our findings show how SynTs organize the barrier and imply their possible roles in supporting embryogenesis, including cardiovascular development. SynT-derived factors and SynT within the placenta may play critical roles in ensuring proper organogenesis of other organs in the embryo.

Successful placentation at the first trimester is essential for the placental function of the following gestational periods and subsequent outcome of pregnancy: inadequate early placentation is associated with a higher risk of miscarriage[1], which is the most common form of pregnancy loss and occurs mainly in the first trimester[2] and can manifest in later pregnancy complications. In particular, proper development of the early placenta is crucial for the normal development of embryonic organs, such as the heart: the heart-cardiovascular system is the first organ to develop during organogenesis[3], and the placenta and the embryonic heart arise in the first trimester and develop concurrently in the gestational timeline. Their defects frequently co-exist,

as seen in vivo[4] and clinical studies[5–7], and the phenomenon of interrelated influence between the two organs has been referred to as the placenta-heart axis[8]. Over the past 20 years, numerous in vivo studies have suggested that proper placentation is a prerequisite for normal cardiovascular development[4,9–13]. However, the cellular and molecular mechanisms of how the placenta supports the development of the embryo proper, including the cardiovascular system, have yet to be well-defined.

Syncytiotrophoblast (SynT), a multinucleated layer formed by the continuous fusion of mononucleated cytotrophoblasts (CTB), serves as the primary barrier for exchange between maternal and fetal

circulations, and its adequate development is a hallmark of successful placentation: defective syncytialization and altered expression of its modulators underpin numerous pregnancy-related complications[14]. Moreover, various placental cell types, including vascular and perivascular cells, also participate in barrier formation[15]. Regarding the placenta-heart axis, as a part of efforts to elucidate its cause-effect relationship, one step has recently been achieved through a massive in vivo phenotyping study: among trophoblast cells, SynT is proven to be a leading contributor to developmental heart disease[16]. However, the factors that organize the materno-fetal interface and the mechanisms by which the primary placenta influences the developing embryo have not been elucidated.

Progesterone immunomodulatory binding factor 1 (PIBF1) carries an essential role in maternal immune tolerance during pregnancy, as pregnant lymphocyte-produced PIBF1 inhibits natural killer (NK) cell activity[17] and suppresses premature parturition under uterine stress[18]. The maternal level of PIBF1 increases continuously throughout pregnancy until term, and a low level of PIBF1 is associated with adverse pregnancy outcomes[19–24]. Concurrently, PIBF1 is highly expressed in SynT in the human placenta during the first trimester and its expression declines at term[25], indicating the spatiotemporally different role of PIBF1 depending on the compartments of the mother and conceptus.

In this study, we found that SynT, a product of trophoblast fusion, interacts with adjacent vascular cells to initiate the formation of the exchange barrier in the primary placenta, and this process is orchestrated by trophoblast-expressed and SynT-derived PIBF1. Our findings suggest that a healthy placenta with proper SynT is required for normal early embryonic organogenesis, including cardiovascular development. The enhanced manifestation of cardiovascular features seen in PIBF1-treated heart organoids further implies the possible role of PIBF1 in cardiovascular development.

## Results

### Essential roles of PIBF1 in trophoblast syncytialization

We first investigated whether PIBF1 mediates syncytialization using a BeWo cell line. CRISPR/Cpf1-mediated gene targeting against exon 2 or 4 of human *PIBF1* (Supplementary Fig. 1) generated *PIBF1* knockout (KO) BeWo cell lines (Supplementary Fig. 2a). While cell fusion was induced in wild-type (WT) cells by the addition of adenylate cyclase activator forskolin (Fig. 1a), syncytia failed to form in *PIBF1* KO BeWo cells (BeWo KO line #1 in Supplementary Fig. 2a) (Fig. 1a). This was accompanied by decreased expression of fusogenic gene *ERVFRD-1* (*Syncytin 2*) and endocrine hormone gene *chorionic gonadotropin beta* (*CGB*) (Fig. 1b). Human CG (hCG), which is mainly produced by SynT and secreted into the maternal compartment at high levels during the first trimester[26], was detected at significantly lower levels in terms of secretion and protein in *PIBF1*-deficient cells, regardless of forskolin stimulation (Fig. 1c, d). These results were also replicated in *PIBF1* exon 4-targeted KO cells (BeWo KO line #2) (Supplementary Fig. 2b–e), indicating the role of PIBF1 in BeWo trophoblast fusion.

Next, to determine whether PIBF1 mediates the differentiation of human trophoblast stem cells (hTSC) to SynT, we generated *PIBF1* KO hTSC (Supplementary Fig. 1 and TS KO line #1 in Supplementary Fig. 3a, b). As a result, *PIBF1* KO hTSC showed no changes in cell proliferation (Supplementary Fig. 3c), with TS markers slightly increased in stem cell conditions (Supplementary Fig. 3d, e). However, *PIBF1* KO hTSC displayed a defective differentiation capacity into β-hCG-expressing SynT in both 2D and 3D cultures (Fig. 1e). This was consistent with the observation that *PIBF1* KO hTSC at a basal state exhibited a declined expression of *GCM1* (Supplementary Fig. 3f), a classical master regulator of SynT formation[27]. The resultant SynT derived from *PIBF1* KO hTSC showed decreased levels of SynT markers (Fig. 1f) and β-hCG secretion (Fig. 1g), while the levels of TS markers remained abnormally high (Fig. 1h). In addition, *PIBF1* KO hTSCs also demonstrated a

defective differentiation capacity into extravillous trophoblasts (EVTs) (Supplementary Fig. 3g, h). Considering that *GCM1* was expressed at low levels in *PIBF1* KO hTSC and has recently been refocused for its contribution to EVT differentiation[27], the possible role of PIBF1 in EVT development needs further investigation.

To examine the in vivo differentiation potency, *PIBF1* KO hTSCs were engrafted subcutaneously into severe immunodeficient male mice (Fig. 1i). Hosts injected with hTSCs contained a substantial and detectable amount of hCG in serum and urine, while those with *PIBF1* KO did not (Fig. 1j, k). The engrafted lesions of *PIBF1* KO were indistinguishable from those of WT in terms of gross size (Fig. 1l), and histologically, both genotypes exhibited similar expression of the pan-trophoblast marker cytokeratin 7 (KRT7) (Fig. 1m). However, differentiated trophoblast populations marked with β-hCG or HLA-G were largely absent in lesions developed from *PIBF1* KO hTSCs (Fig. 1m). Additionally, *PIBF1* KO hTSC lines other than KO line #1 were obtained using CRISPR/Cpf1-mediated gene targeting (Supplementary Fig. 1 and Supplementary Fig. 4a, b), among which most of the results regarding TSC stemness and SynT/EVT differentiation capacities were also replicated in *PIBF1* KO hTSC line #2 (Supplementary Fig. 4c–f). These findings with hTSCs suggest that PIBF1 mediates the cell-to-cell fusion of trophoblasts and the differentiation of TSC into SynT.

Full-length PIBF1 (fPIBF) is a nucleus- or centrosome-associated 90 kDa protein, while the smaller 35 kDa secretory form (sPIBF) is localized cytoplasmically[28]. To further validate our findings, we overexpressed fPIBF and/or sPIBF in *PIBF1*-deficient BeWo cells (Fig. 1n). Consequently, the restoration of fPIBF or sPIBF induced cell-to-cell fusion, which was further enhanced by the simultaneous restoration of both variants (Fig. 1o). Notably, exogenous PIBF treatment (Supplementary Fig. 5a) in *PIBF1* KO BeWo cells did not result in any significant improvement in syncytial formation (Fig. 1p and Supplementary Fig. 5b, c), indicating that PIBF1 mediates the cell-to-cell fusion event only when expressed and produced endogenously.

In summary, our data demonstrate that endogenous PIBF1 plays indispensable roles in trophoblast differentiation and syncytialization.

### Angiogenic role of SynT-derived PIBF1

In the developing human placenta, a small fraction of mononuclear cytotrophoblasts (CTBs) retain the properties of trophoblast stem cells (TSCs) and differentiate into multinucleated syncytiotrophoblasts (SynT), which, along with villous endothelial cells (ECs), establish a materno-fetal exchange interface as well as a physical and immunological barrier[29,30]. However, the mechanisms by which these cell populations communicate to establish the multi-layered structure are yet unclear. In this context, we hypothesized that syncytialized trophoblasts could secrete PIBF1 to build the complex interface, as sPIBF can be secreted by immune cells and act as a cytokine[18,28]. To address this, we first measured cellular and secreted levels of PIBF1 in TSC, SynT, and EVT using multiple reaction monitoring, a highly selective and sensitive technique for quantifying targeted proteins[31]. As a result, the intracellular level of PIBF1 was comparable among TSC, SynT, and EVT (Fig. 2a), while secreted PIBF1 was significantly more abundant in SynT or EVT than in TSC (Fig. 2a).

Next, to gain more detailed insights into the interaction between SynT and EC, we designed a SynT-EC transwell co-culture to verify whether syncytialized trophoblasts were capable of recruiting vascular endothelial cells (Fig. 2b). We found that hTSC-derived SynT or BeWo cells fused by forskolin recruited a remarkably higher number of HUVECs compared to non-differentiated hTSC or non-fused cells (Supplementary Fig. 6a, b), implying SynT-derived paracrine or endocrine signaling to endothelial cells. When *PIBF1* was deficient in hTSC, as expected, this endothelial recruiting ability was lacking (Fig. 2c). *PIBF1* KO BeWo cells also displayed defects in endothelial recruitment, which was rescued by the restoration of fPIBF and/or sPIBF (Fig. 2d). Notably, reintroducing sPIBF resulted in a higher level

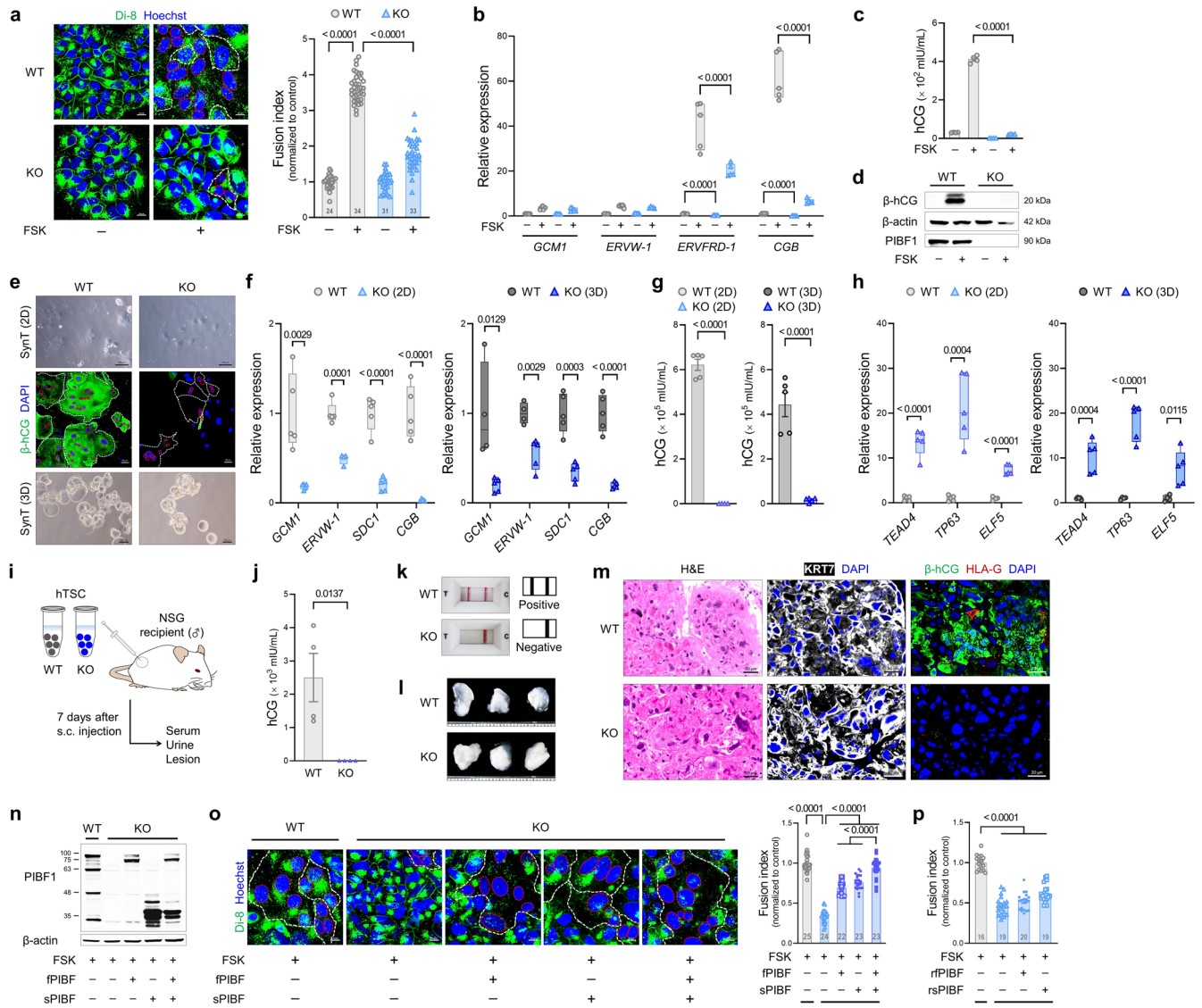

**Fig. 1 | PIBF1 is required for trophoblast syncytialization. a** Trophoblast fusion assay in WT and *PIBF1* KO BeWo cells. BeWo boundaries and nuclei were stained with Di-8-ANEPPS (Di-8; green) and Hoechst (blue), respectively. White and red dotted lines delineate the plasma membrane and nuclei within the fused cells. *n* = 24, 34, 31, 33 in each group. **b** Expressions of SynT markers in FSK-treated *PIBF1* KO BeWo cells. *n* = 5-6 per group. **c** Levels of hCG secreted by FSK-treated *PIBF1* KO BeWo cells. *n* = 4 per group. **d** Expressions of β-hCG and PIBF1 in FSK-treated *PIBF1* KO BeWo cells. β-actin was used as an internal control. **e** Phase-contrast images of *PIBF1* KO hTSC-derived syncytiotrophoblast (TS-SynT) in 2D (upper) and 3D cultures (lower) and immunostaining of β-hCG (green) in 2D *PIBF1* KO TS-SynT (middle). White and red dotted lines delineate the plasma membrane and nuclei within the fused cells. **f** Expressions of SynT markers in 2D and 3D *PIBF1* KO TS-SynT. *n* = 4-5 per group. **g** Levels of hCG secreted by 2D and 3D *PIBF1* KO TS-SynT. *n* = 4-5 per group. **h** Expressions of TS markers in 2D and 3D *PIBF1* KO TS-SynT. *n* = 4-5 per group. **i** Schematic diagram of *PIBF1* KO hTSC engraftment into immunodeficient NSG mice. **j** Serum hCG level in NSG mice subcutaneously injected with *PIBF1* KO hTSCs. *n* = 4 per group. **k** Representative images of over-the-counter hCG pregnancy test of urine samples collected from *PIBF1* KO hTSC-injected mice.

**l** Gross trophoblastic lesions that were developed from transplanted *PIBF1* KO hTSCs. **m** H&E and immunofluorescence staining of KRT7 (white), β-hCG (green), and HLA-G (red) in a *PIBF1* KO hTSC-derived lesion. **n** Expression of PIBF1 in FSK-treated *PIBF1* KO BeWo reintroduced with full-length (fPIBF) and/or secretory PIBF1 (sPIBF). **o** Fusion assay in *PIBF1* KO BeWo cells after the restoration of fPIBF and/or sPIBF. *n* = 25, 24, 22, 23, 23 in each group. **p** Fusion assay in *PIBF1* KO BeWo cells supplemented with recombinant fPIBF1 (rfPIBF) and/or sPIBF1 (rsPIBF). *n* = 16, 19, 20, 19 in each group. Fusion assay data (**a**, **o**, and **p**) are expressed relative to that of WT control. Transcript expression data were normalized to *RPS18* (**b**) or *GAPDH* (**f** and **h**) and expressed relative to that of WT. Data are presented as mean ± SEM (**a**, **c**, **g**, **j**, **o**, and **p**) or box-and-whisker plots where the interquartile range (IQR) of boxplot is between Q1 and Q3, the center line indicates the median value, whiskers of boxplot is extended to the maxima and minima, and maxima is Q3 + 1.5×IQR and minima is Q1 − 1.5×IQR (**b**, **f**, and **h**); *P < 0.05; **P < 0.01; ***P < 0.001; ****P < 0.0001 in one-way ANOVA (**a-c**, **o**, and **p**) or two-sided Student's t-test (**f-h** and **j**). The number *n* represents biologically independent cells in each group. Source data are provided as a Source Data file.

of endothelial recruitment compared to that of fPIBF (Fig. 2d), suggesting that fused trophoblast-derived signals encourage endothelial cell homing to the site of the materno-fetal interface, with sPIBF being the primary mediating factor.

As the placental blood vessels consist of several cell populations, including ECs and pericytes, we tested which vascular cell types were

affected by PIBF1 in angiogenesis. Endothelial cells derived from the umbilical vein and artery (HUVEC and HUAEC, respectively) were adopted due to their different profiles of endothelial identity[32,33]. As a result, recombinant sPIBF enhanced indices of endothelial migration and tube formation in a dose-dependent manner in HUVEC (Fig. 2e–g) via activation of angiogenesis-related signaling cascades (Fig. 2h and

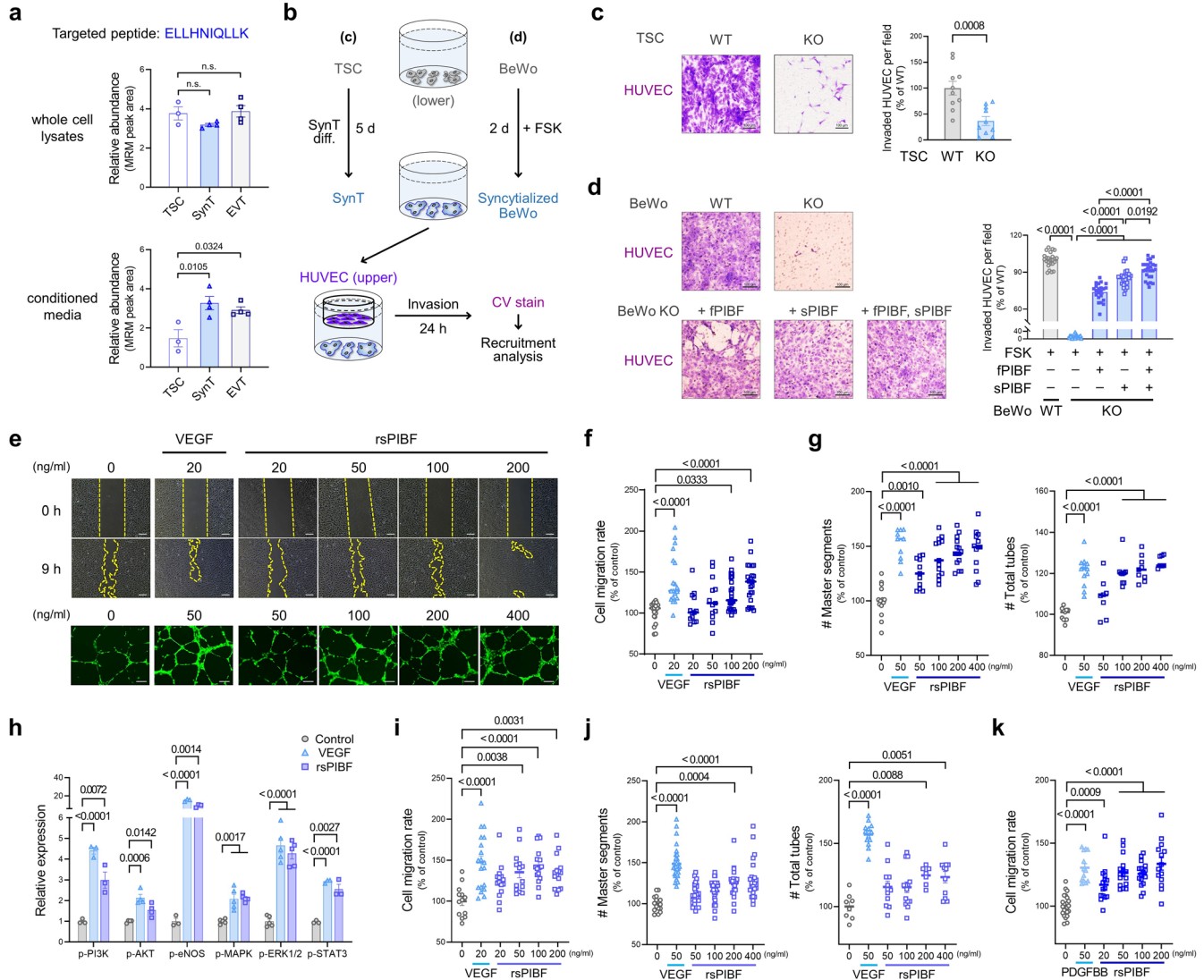

**Fig. 2 | Trophoblast-derived PIBF1 mediates vasculature formation in vitro.**
**a** Multiple reaction monitoring (MRM) analysis of intracellular and secreted PIBF1 in hTSC, TS-SynT, and TS-EVT. n.s., not significant. $n = 3$-4 per group. **b** Schematic diagram of endothelial recruitment by hTSC-derived SynT or fused BeWo cells. CV, crystal violet. **c** Recruitment analysis of HUVEC by differentiated SynT from WT and *PIBF1* KO hTSCs. $n = 10$ per group. **d** Recruitment analysis of HUVEC by WT and *PIBF1* KO BeWo cells restoration of fPIBF and/or sPIBF. $n = 24, 22, 22, 23, 25$ in each group. **e** Representative images for migration (upper panel) and tube forming assay (lower panel) in rsPIBF-treated HUVEC. Scale bar: 200 μm. **f, g** Quantified data from migration (**f**) ($n = 20, 19, 13, 13, 24, 23$ in each group) and tube forming assay (**g**) ($n = 15, 10, 11, 13, 15, 13$ in each group of master segments count, $n = 11, 12, 8, 9, 9, 7$ in each group of total tubes count) in rsPIBF-treated HUVEC. **h** Expressions of

angiogenesis-related signaling cascades in rsPIBF-treated HUVEC. Data were normalized to β-actin. $n = 3$–5 per group. **i, j** Migration (**i**) ($n = 13, 19, 15, 14, 17, 15$ in each group) and tube forming assay (**j**) ($n = 15, 31, 26, 27, 20, 24$ in each group of master segments count, $n = 8, 13, 12, 15, 7, 11$ in each group of total tube count) in rsPIBF-treated HUAEC. **k** Migration assay in rsPIBF-treated placental pericytes. $n = 25, 13, 15, 16, 16, 14$ in each group. Recruitment (**c, d**), migration (**f, i,** and **k**), tube forming (**g** and **j**), and the band intensity of protein data (**h**) are expressed relative to WT control and are presented as mean ± SEM; *$P < 0.05$; **$P < 0.01$; ***$P < 0.001$; ****$P < 0.0001$ in one-way ANOVA (**a, c, d, f–k**). The number $n$ represents biologically independent cells in each group. Source data are provided as a Source Data file.

Supplementary Fig. 6c), and similarly in HUAEC (Fig. 2i, j and Supplementary Fig. 6d). It also dose-dependently increased the migratory property of placental pericytes (Fig. 2k and Supplementary Fig. 6e).

The N-terminal region of PIBF1, ranging from amino acids 1-184, is a biologically active site[34] and is equivalent to the recombinant active domain containing a fragment of PIBF1 (aPIBF) used in our study (Supplementary Fig. 5a). The angiogenic effects of recombinant fPIBF and aPIBF proteins were validated in HUVEC (Supplementary Fig. 6f-h), HUAEC (Supplementary Fig. 6i–k), and placental pericytes (Supplementary Fig. 6l). In contrast, the C-terminal fragment of PIBF1 (cPIBF) showed no changes in endothelial migration, tube formation, or pericyte migration (Supplementary Fig. 6f–l). This indicates that the angiogenic property of PIBF1 is located in the N-terminal region

encoded by exons 2-4, which are known to exert its immunological activity[34].

Collectively, these results show that SynT-derived PIBF1 recruits vascular and perivascular cells to form vessel-like structures and promotes their angiogenesis, suggesting that PIBF1 may play a role in the development of the placenta's circulatory system and the feto-maternal interface.

## Defective syncytialization and formation of the materno-fetal interface in placenta lacking *Pibf1*

We next investigated whether the loss of Pibf1 affects placentation and subsequent embryogenesis by evaluating *Pibf1*-null placentas and embryos. *Pibf1*+/- intercrosses produced no *Pibf1*−/− offspring

(Supplementary Fig. 7a), which was not associated with the implantation potential of blastocysts (Supplementary Fig. 7b). Instead, *Pibf1*-null embryos experienced developmental retardation from E9.5-10.5 (Supplementary Fig. 7c, d). At this stage, when the primary placenta begins to form[35] and *Pibf1* is deleted in all cell lineages of the labyrinthine interhaemal membrane (Fig. 3a), the loss of *Pibf1* led to defective placentation and embryogenesis (growth restriction in size, failed tail turning, pericardial effusion) (Fig. 3b). *Pibf1*-null placentas were morphologically and functionally disrupted, as evidenced by an underdeveloped labyrinth with decreased fetal-derived blood vessels (Fig. 3c, d) and reduced fatty acid transfer capacity (Fig. 3e, Supplementary Data 1).

Possible mechanisms of Pibf1 regarding the failed development of *Pibf1*-null placenta and embryos can be explored in light of previous literature. Pregnancy state and outcome are associated with maternal PIBF1 levels[24]. However, *Pibf1*-null lethality did not result from maternal *Pibf1* haploinsufficiency, as the abnormal appearances of *Pibf1*-null embryos were also replicated in WT foster mothers (Supplementary Fig. 8a, b). Meanwhile, Pibf1 was first described as a suppressor of conventional cytolytic natural killer (NK) cells[17], while uterine NK (uNK) cells dampen IL17-producing helper T (Th17)-mediated inflammation, which occurs locally in the maternal decidua, to promote immune tolerance[36] (Supplementary Fig. 8c). We therefore examined whether embryonic loss of *Pibf1* impacted the uNK population. However, no significant change was observed in their proportion among immune cells or absolute number in *Pibf1*[+/-] decidua harboring *Pibf1*[-/-] embryos (Supplementary Fig. 8d, e). In addition, functional markers of uNK cells such as granzyme B (cytotoxic granule contents[37]), TNF-α and IFN-γ (pro-inflammatory cytokines[38]), and CD107a (a proxy for degranulation[39]) also did not show significant changes in *Pibf1*[+/-] decidua with *Pibf1*[-/-] embryos (Supplementary Fig. 8f), suggesting that embryonic loss of *Pibf1* did not have a substantial impact on the population and function of uNK cells. Furthermore, in a Th17-deficient maternal immune background (Supplementary Fig. 8g), *Pibf1*[-/-] embryos still exhibited morphological abnormalities (Supplementary Fig. 8h), similar to those detected in littermates from *Pibf1* WT or heterozygous mothers. These data suggest, at least in part, that the maternal compartments of the conceptus were genetically or immunologically non-pathogenic to the early defects of the *Pibf1*[-/-] embryo.

As PIBF1 plays essential roles in TSC differentiation and trophoblast fusion in vitro, we investigated whether *Pibf1*-null placentas have defects in SynT formation. In *Pibf1*[-/-] placentas, defective labyrinth formation was accompanied by decreased trophoblast proliferation (Fig. 3f). Among TSCs and their lineages, the labyrinthine trophoblast within the interhaemal membrane affected by *Pibf1* deletion was SynT-II: while s-TGCs developed typically (*Ctsq* in Fig. 3g, alkaline phosphatase positive in Fig. 3h), SynT-II markers (*Gcm1*, *Synb*, *Mct4*) were significantly decreased, with those of SynT-I (*Syna*, *Mct1*) not showing statistically significant changes ($P = 0.232$ and $0.495$ in *Syna* and *Mct1* in Student's *t*-test, respectively) (Fig. 3g). This was further confirmed by MCT1/4 co-staining (Fig. 3i). Moreover, at an ultrastructural level in *Pibf1*[-/-] labyrinth, the interhaemal membranes were few and abnormally increased in thickness, and most importantly, SynT-II failed to undergo fusion (Fig. 3j). SynT-I also exhibited visible abnormalities (excessively increased thickness, vacuoles, and electron-dense areas) (Fig. 3j). In terms of vascular development within the placenta, *Pibf1*[-/-] embryo-derived blood vessel formation (Laminin-positive) declined, along with an increased degree of apoptosis (TUNEL-positive) (Fig. 3k); moreover, the reduced placental vasculature was site-specific (only in the labyrinth) (Fig. 3l), which is consistent with our histological data.

Collectively, these data indicate that the loss of *Pibf1* leads to impaired cell-cell fusion in trophoblasts and their interaction with the endothelium, and consequently a failure to form a materno-fetal interface. This suggests that defective primary placentation is a major cause of the observed lethality.

## Syncytialization and trophoblast-endothelial interaction recovered in a functional placenta harboring *Pibf1*-intact trophoblast

Pibf1 displayed a ubiquitous expression pattern throughout the placenta and embryo (Supplementary Fig. 9a, b), making it difficult to determine which compartments are involved in the embryonic-lethal phenotype of *Pibf1*[-/-] embryos. To distinguish between trophoblast-intrinsic and embryo lineage-induced effects within the placenta, we generated conditional *Pibf1* KO (cKO) conceptuses using *Meox2*-cre (Supplementary Fig. 10), a transgenic line for epiblast-specific cre-recombinase expression (endothelium, fetal blood cell), while leaving PIBF1 expression intact in trophectoderm-derived cells (S-TGC, SynT-I, SynT-II) within the placental labyrinth[40] (Fig. 4a and Supplementary Fig. 11a). *Pibf1*-null embryos associated with a *Pibf1*-retained placenta (Δ/+) were normally detected at an expected Mendelian ratio (Fig. 4b), and both embryos and placentas were fully recovered at E10.5 (Fig. 4c–f), a stage when complete knockout counterparts were morphologically or histologically disrupted. Importantly, this rescue was accompanied by normal states of TSC and its differentiated lineages, especially SynT-II (Fig. 4g, h), and the labyrinthine vasculature (Fig. 4i).

The placenta also serves as a hematopoietic organ during this period[41,42]: briefly, primitive hematopoiesis first occurs in the yolk sac blood islands (<E8.5), then expands to the endothelial cells of major vessels in the conceptus, including the placenta (E9.5-10.5). Hematopoietic progenitors subsequently colonize the fetal liver during gestation and bone marrow after birth[43]. We observed that the hematopoietic potential of *Pibf1*[-/-] yolk sac and placenta were normal until E9.5 (Supplementary Fig. 12a–c), but aberrantly decreased at E10.5 (Fig. 4j and Supplementary Fig. 12a, d). This defect was recovered in the *Pibf1*-intact yolk sac and placenta, and even in the *Pibf1* cKO embryo at E10.5 (Fig. 4j). Moreover, the rescued placenta displayed normal morphometry and nutrient transport function at E13.5, as identified in histological assessment and measurement of fatty acid transfer capacity (Supplementary Fig. 13a, b, Supplementary Data 2). At this stage, the distribution of SynT-I and -II was also comparable to that of the control placenta (Supplementary Fig. 13c).

In addition, regarding its subcellular characteristics, the full-length 90 kDa PIBF1 is localized in the nucleus or centrosome[28]. Elevated apoptosis and expression of the apoptosis regulator Trp53 (Supplementary Fig. 14a) in *Pibf1*-null embryos prompted us to investigate the role of Trp53 in the viability of embryos lacking *Pibf1*. However, the embryonic function of Pibf1 was not related to the Trp53 pathway, as *Pibf1;Trp53* double KO embryos phenocopied *Pibf1* KO embryos (Supplementary Fig. 14b, c). The centrosome-associated protein PIBF1 is also reported to be essential for the formation of primary cilia[44–46], and its mutations were reported in some patients with ciliopathy[47–49]. We found that while *Pibf1*[Δ/Δ];*Meox2*[cre/+] fetuses had an expected Mendelian ratio until E18.5, no cKO newborns were detected after birth (Supplementary Fig. 14d). Also, *Pibf1*[Δ/Δ];*Meox2*[cre/+] fetuses displayed phenotypic features of ciliopathy (Supplementary Fig. 14e–h), which could potentially contribute to postnatal fetal demise, irrespective of a functionally normal placenta.

To provide further insights into the roles of trophoblast syncytialization and its interaction with vascular cells, we dissected endothelial lineages of the labyrinthine interhaemal membrane (Fig. 4k) using two endothelial-specific cre lines−*Cdh5*-cre and *Tie2*-cre−that express each promoter-driven cre recombinase in endothelial lineages (Supplementary Fig. 11a)[50,51]. *Cdh5*- or *Tie2*-cre-mediated *Pibf1* cKO embryos were detected at E10.5 with expected normal Mendelian inheritance (Fig. 4l) and no apparent morphological anomalies (Fig. 4m, o). The histological index of their placentas was not significantly different from the control (Fig. 4m, n). As *Tie2*- or *Cdh5*-cre-expressing cells share the identity of both endothelial and hematopoietic lineages[50,51], the endothelial-specific *Pibf1* cKO conceptuses were explored for functional hematopoietic activity in their hemogenic compartments, which displayed comparable levels among

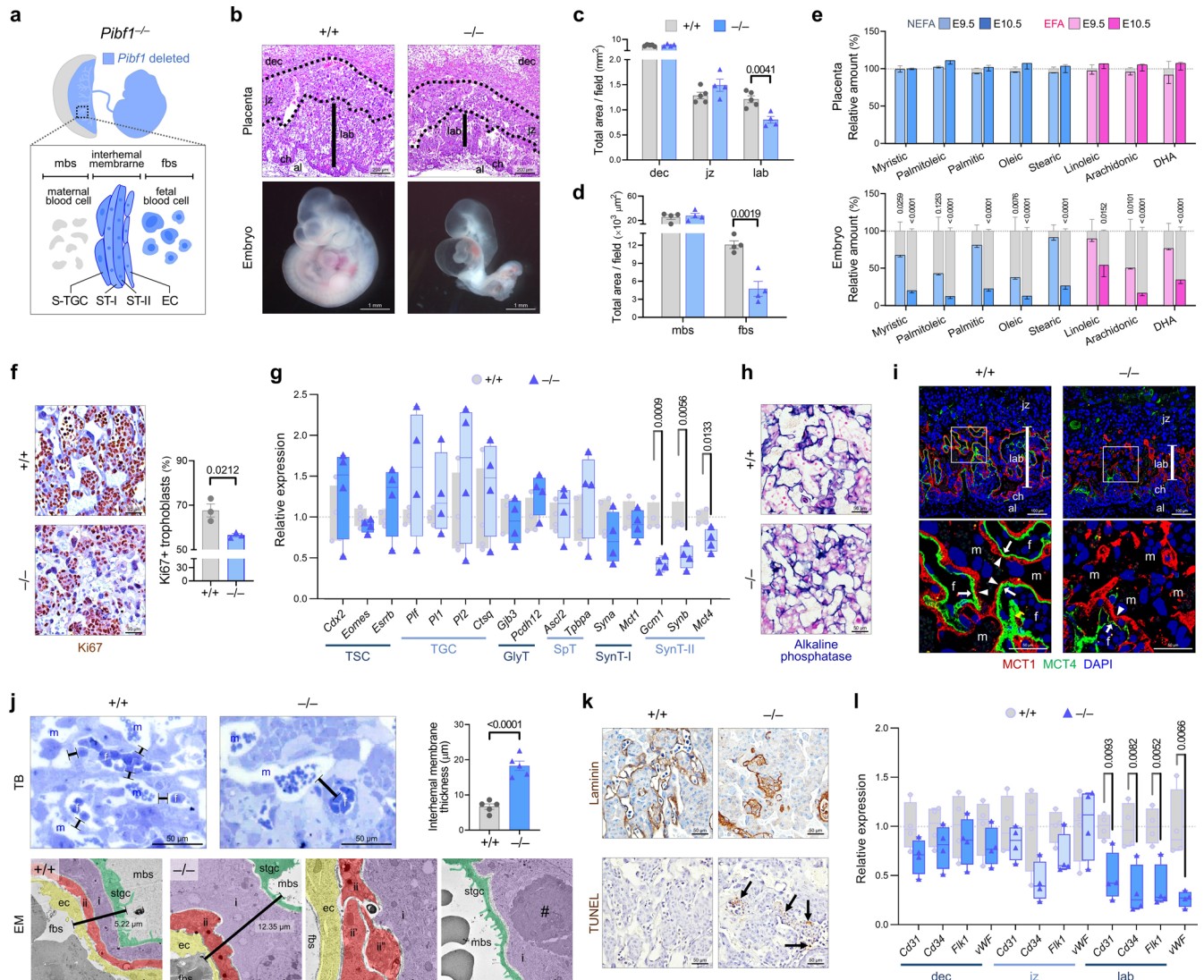

**Fig. 3 | Trophoblast syncytialization and vascular labyrinth integrity of the placenta are impaired in E10.5 *Pibf1*-null mice. a** Schematic diagram of the lineages containing *Pibf1* deletion in the *Pibf1*⁻/⁻ conceptus and labyrinth interhaemal membrane. S-TGC, sinusoidal trophoblast giant cell. ST-I and -II, syncytiotrophoblast type I and II, respectively. EC, endothelial cell. **b** Placental histology and gross appearance of *Pibf1*⁻/⁻ embryos. Solid black lines denote the maximum thickness of the labyrinth. dec, decidua. jz, junctional zone. lab, labyrinth. ch, chorion. al, allantois. **c, d** Gross histological assessment of placental morphology (**c**) and morphometric analysis of the labyrinth vascular beds (**d**) in *Pibf1*⁻/⁻ placentas. mbs and fbs, maternal and fetal blood space, respectively. *n* = 4–5 per group. **e** Fatty acid (FA) contents in *Pibf1*⁻/⁻ placenta and embryo at E9.5-10.5 as measured by gas chromatography coupled with mass spectrometry (GC-MS). Data are expressed as the relative amount (%) (colored bar) compared to that of wild type (WT; gray bar) (*n* = 6/genotype). The term "acid" was omitted except for docosahexaenoic acid (DHA). NEFA, non-essential FA; EFA, essential FA. *n* = 6 per group. **f** Proliferation of labyrinth trophoblasts in *Pibf1*⁻/⁻ placenta as determined by the proportion of Ki67-positive cells in labyrinth trophoblasts. *n* = 3 per group. **g** Expressions of markers for TSC and its lineages in *Pibf1*⁻/⁻ placenta. GlyT, glycogen trophoblast cell. SpT, spongiotrophoblast. *n* = 4 per group. **h** Distribution of maternal blood sinus-covered sinusoidal TGC (S-TGC) in *Pibf1*⁻/⁻ labyrinth labeled with endogenous alkaline phosphatase. **i** Immunofluorescence staining of MCT1 (red) and MCT4

(green) to detect SynT-I (arrowhead) and -II (arrow) layers, respectively, in *Pibf1*⁻/⁻ placenta. m and f, maternal and fetal blood sinus, respectively. **j** Ultrastructure of the interhaemal membrane in *Pibf1*⁻/⁻ labyrinth. (Upper panel) Toluidine blue (TB)-stained semithin section and thickness of the interhaemal membrane. (Lower panel) Pseudo-colored transmission electron microscopy (EM) images in ultrathin section (lower). ii, ii', and ii″ denote a lack of fusion within SynT-II. Hashtag denotes an excessive electron-dense area in SynT-I. ec, endothelial cell. i, SynT-I. ii, SynT-II. *n* = 5 per group. **k** Immunostaining of Laminin and TUNEL to label endothelial lining of fetal blood vessels and apoptotic cells in *Pibf1*⁻/⁻ placenta. TUNEL-positive signals were detected only in cells associated with the fetal-derived vessel or fetal red blood cells in *Pibf1*⁻/⁻ labyrinth (arrows). **l** Expressions of endothelial vasculature markers in laser micro-dissected *Pibf1*⁻/⁻ placenta sections. *n* = 4 per group. Experimental samples were obtained on E10.5 unless indicated otherwise. Transcript expression data (**g** and **l**) were normalized to *Gapdh* and expressed relative to that of WT. Data are presented as mean ± SEM (**c**–**f**, and **j**) or box-and-whisker plots where the interquartile range (IQR) of boxplot is between Q1 and Q3, the center line indicates the median value, whiskers of boxplot is extended to the maxima and minima, and maxima is Q3 + 1.5×IQR and minima is Q1 − 1.5×IQR (**g** and **l**); *$P < 0.05$; **$P < 0.01$; ***$P < 0.001$ in two-sided Student's t-test. The number *n* represents biologically independent embryos or placentas in each group. Source data are provided as a Source Data file.

genotypes (Fig. 4p). Additionally, when *Pibf1* was ablated in hematopoietic lineages using *Vav1*-, *Cdh5*-, or *Tie2*-cre lines, the hematopoietic functions of different *Pibf1* cKO organs were normal at E13.5 (cKO^Vav1 and cKO^Tie2 in Supplementary Fig. 15a–d) and postnatally in adult

(cKO^CdhS and cKO^Vav1 in Supplementary Fig. 15e, f). These hematopoiesis data suggest that proper syncytialization of trophoblasts may ensure normal hematopoietic development in both extra-embryonic and embryonic organs.

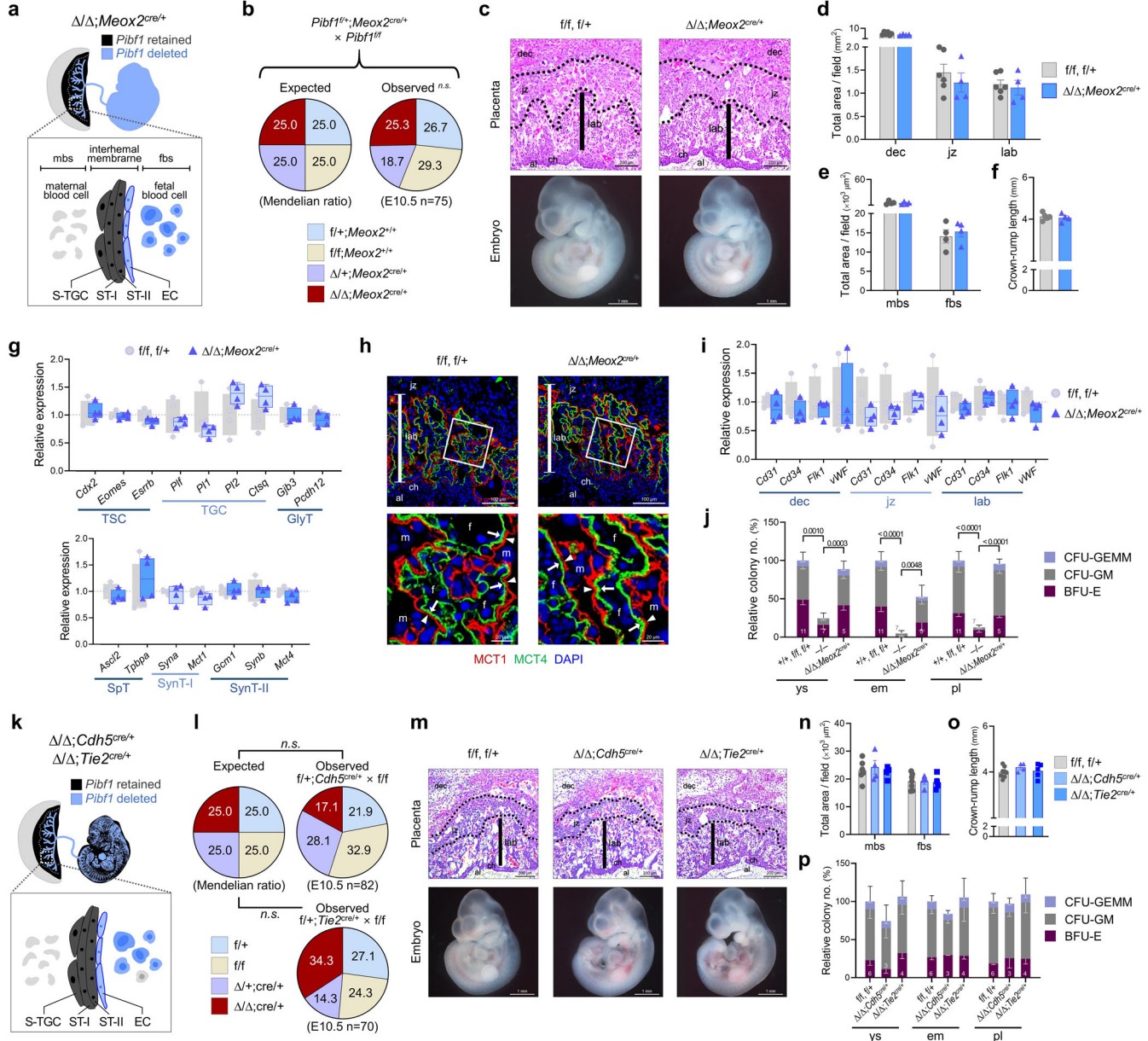

**Fig. 4 | *Pibf1* is required in trophoblast development, but not endothelial/hematopoietic lineages, for normal syncytialization, placental vascularization, and subsequent embryo survival in mice. a** Schematic diagram of the lineages containing *Pibf1* deletion in the conceptus and labyrinth interhaemal membrane with conditional knockout (cKO) driven by *Meox2cre* (cKO*Meox2*). **b** The expected and observed genotype distribution in cKO*Meox2* intercross. **c** Placental histology and gross appearance of cKO*Meox2* embryos. Gross histological assessment of placental morphology **d** (*n* = 4-6 per group) and morphometric analysis of the labyrinth vascular beds (**e**) (*n* = 4 per group) in cKO*Meox2* placentas. **f** Measurement of the crown-rump length in cKO*Meox2* embryos. *n* = 4–5 per group. **g** Expressions of markers for TSC and its lineages in cKO*Meox2* placenta. *n* = 4 per group. **h** Immunofluorescence staining of MCT1 (red) and MCT4 (green) to detect SynT-I (arrowhead) and -II (arrow) layers, respectively, in cKO*Meox2* placenta. **i** Expressions of markers for endothelial vasculature in laser micro-dissected sections of cKO*Meox2* placenta. *n* = 4 per group. **j** Hematopoietic potential in the yolk sac (ys), embryo proper (em), and placenta (pl) of *Pibf1* KO and cKO*Meox2* as determined by colony forming unit (CFU) assay. CFU-GEMM = CFU-granulocyte, erythroid, macrophage,

megakaryocyte. CFU-GM = CFU-granulocyte, macrophage. BFU-E = burst forming unit-erythroid. *n* = 11, 5, 7 in each group. **k** Schematic diagram of the lineages containing *Pibf1* deletion in the conceptus and labyrinth interhaemal membrane with endothelial-specific cKO (ecKO) driven by *Cdh5cre* (ecKO*Cdh5*) or *Tie2cre* (ecKO*Tie2*). **l** The expected and observed genotype distribution in ecKO*Cdh5* or ecKO*Tie2* intercross. **m** Placental histology and gross appearance of ecKO*Cdh5* and ecKO*Tie2* embryos. **n** Gross histological assessment of ecKO placentas. *n* = 7, 4, 4 in each group. **o** Measurement of crown-rump length in ecKO embryos. *n* = 8, 4, 5 in each group. **p** Hematopoietic potential in ecKO mutants as determined by CFU assay. *n* = 6, 3, 4 in each group. Experimental samples were obtained on E10.5 unless indicated otherwise. Transcript expression data (**g** and **i**) were normalized to *Gapdh* and expressed relative to that of the control group, and CFU data (**j** and **p**) were expressed relative to that of the control group. The number in each bar indicates sample size (n) in (**j**) and (**p**). Data are presented as mean ± SEM (**d**–**f**, **i**, **j**, and **n-p**) or mean ± minimum to maximum (**g**). n.s., not significant in one-sided χ² test (**b** and **l**); **P < 0.01; ***P < 0.001 in one-way ANOVA. The number *n* represents biologically independent embryos or placentas in each group. Source data are provided as a Source Data file.

Collectively, these results indicate that normally syncytialized trophoblasts and their interaction with endothelium, orchestrated by Pibf1, are prerequisites for establishing a materno-fetal interface, a functional placenta, and, ultimately, embryo survival.

## Impact of trophoblast syncytialization upon cardiovascular development of the embryo

As observed in the placenta with its global expression pattern, Pibf1 is also expressed in the developing heart of the early embryo

(Supplementary Fig. 9b). Under our *Pibf1* cKO strategies, the Cre lines are active in the whole heart (*Pibf1*$^{\Delta/\Delta}$;*Meox2*$^{cre/+}$) or part of the heart (endocardial lineages in *Pibf1*$^{\Delta/\Delta}$;*Cdh5*$^{cre/+}$, *Pibf1*$^{\Delta/\Delta}$;*Tie2*$^{cre/+}$) (Fig. 5a and Supplementary Fig. 11b). In line with the notion that the placenta and embryonic heart develop in parallel during early gestation and that their defects commonly co-exist[4–8], the abnormalities of *Pibf1*-null embryos rescued by the SynT-intact functional placenta also included a visibly underdeveloped heart (Fig. 3b). Moreover, syncytialized trophoblasts produce angiogenic factors such as PIBF1 (Fig. 2a), which might be able to potentially affect endothelial lineages or cardiac vasculature in the developing heart. In this context, we examined the impact of Pibf1-mediated trophoblast syncytialization on cardiovascular development in *Pibf1* mutants. As seen in the whole mount CD31 stain, the vasculature of the yolk sac, intersomitic region, and heart was severely disrupted in *Pibf1*$^{-/-}$ embryos; however, it was largely recovered in *Pibf1* cKO embryos (Fig. 5b). Particularly, in the cardiac region, CD31-stained areas were significantly decreased in *Pibf1*$^{-/-}$ hearts but were comparable among control and *Pibf1* cKO hearts (Fig. 5c, d). In gross and histological appearances, the *Pibf1*$^{-/-}$ hearts showed severe developmental delays, displaying failed right and left chamber specification, abnormally enlarged pericardial cavity, reduced chamber length, and decreased trabecular density. These cardiac defects were fully rescued in all three *Pibf1* cKO mutant hearts (Fig. 5e, f), which was further confirmed histologically by the myocardial marker cTnT (Fig. 5g, h). The early cardiogenesis-related markers (myocardium development and cardiac jelly regulation)[52] and coronary vascular development markers (pan-endothelial lineage and arterial/venous specification)[53] exhibited a similar pattern, as most of them were decreased in *Pibf1*$^{-/-}$ hearts and recovered in those of all *Pibf1* cKO mutants (Fig. 5i, j), under the influence of a functional SynT-bearing placenta. These data indicate that the cardiovascular development of the embryo depends on Pibf1-mediated trophoblast syncytialization and placentation, not the genetic state of *Pibf1* in the heart or vascular system itself.

Additionally, concerning its immunotolerant functions under certain pregnancy environments[17,18] and the relationship between its maternal level and pregnancy outcome[19–24,54], we addressed whether loss of *Pibf1* in the maternal immune system affects pregnancy and the offspring (Supplementary Fig. 16a). *Pibf1*$^{\Delta/\Delta}$;*Vav1*$^{cre/+}$ females (*Pibf1*-deficient in hematopoietic/immune cells[55]) displayed normal reproductive functions (Supplementary Fig. 16b), and their pups showed normal growth rate (Supplementary Fig. 16c), indicating that the genetic deletion of *Pibf1* in immune cells does not directly affect the maintenance of pregnancy in mice. Meanwhile, the function of immune cells might also be affected by PIBF1, which can be produced by other cells or tissues. For example, PIBF1 is highly expressed in endometrial stromal cells during the mid-secretory phase of the endometrium and is required for their decidualization[56]. Therefore, further investigation is needed to understand the role of PIBF1 in maintaining pregnancy in maternal environments.

### Manifestation of cardiovascular features facilitated by PIBF1 in heart organoids

To further investigate the effects of PIBF1 on in vitro cardiovascular features, we employed an induced pluripotent stem cell (iPSC)-derived human heart organoid (hHO) model, which can mimic the three-dimensional characteristics of the heart[57]. After differentiating hHOs from iPSC-derived embryonic bodies under the influence of sPIBF (Fig. 6a), we confirmed that cTnT-positive cardiomyocytes were significantly more abundant in sPIBF-treated hHOs than in controls (Fig. 6b). Additionally, co-staining of CD31 and PDGFRβ, markers for endothelial cells and pericytes, respectively, revealed an increased distribution of CD31-positive cells in sPIBF1-treated hHOs compared to controls, while that of PDGFRβ remained unchanged (Fig. 6b). A wide range of rsPIBF doses (50–1000 ng/ml) also showed increases in

endothelial and cardiomyocyte distributions within hHO (Supplementary Fig. 17a).

Considering that hHO is a mixed population of several cells, such as endothelial cells and cardiomyocytes[57], it remained uncertain whether the impact of sPIBF on hHO is due to its direct effect on cardiomyocyte differentiation or indirect effect resulting from enhanced endothelial angiogenesis. To address this issue, we conducted a two-dimensional culture of human iPSC-derived cardiomyocytes[58], which revealed the direct impact of PIBF1 on cardiomyocyte differentiation (Fig. 6c).

Since human iPSC-derived hHOs exhibit electrophysiological signals that can mimic depolarization/repolarization coupling as functional features[57], we measured changes in the leading indicators of field potential (FP), an electrophysiological signal, using a multi-electrode array (MEA). There were no significant differences in beat period and field potential duration (FPD) (Fig. 6d); however, the spike amplitude was significantly higher in sPIBF-treated samples than in controls (Fig. 6d). Moreover, the angiogenic effect of sPIBF observed in endothelial cells and pericytes was further confirmed in hHOs using a Matrigel-based sprouting assay: Most of them were PDGFRβ positive (Fig. 6e). Notably, sPIBF-treated hHOs displayed a more vascularized appearance with longer sprouting lengths than vehicle-treated hHOs (Fig. 6e), which was increased in a dose-dependent manner with sPIBF (Supplementary Fig. 17b).

To better understand how sPIBF affects developing hearts, we analyzed the transcriptomes of vehicle- or sPIBF-treated hHOs and found that 240 genes were differentially regulated by sPIBF, with 105 differentially expressed genes (DEGs) increased and 135 decreased (Supplementary Fig. 18). Gene Ontology enrichment analysis of the DEGs (Supplementary Data 3 and 4) revealed that in hHO, compared to the undifferentiated embryonic body (EB), there was increased expression of markers not only for cardiac muscle but also for other mesenchymal tissues such as bone and fat (Fig. 6f, column 1 vs. columns 2-4 in heatmap). However, when treated with rsPIBF, the expression of other mesenchymal cell fate-related genes was significantly suppressed. Notably, the expression of cardiomyocyte-related genes was significantly increased (Fig. 6f, columns 2-4 vs. columns 5-7 in heatmap), strengthening the fate decision in muscle and heart development by sPIBF (Fig. 6g).

Taken together, PIBF1 encourages differentiation/maturation of cardiomyocytes and endothelial angiogenesis in the in vitro heart development. Along with our findings on the essential roles of SynT-derived PIBF1 in angiogenesis and primary placentation, this may contribute to the development of the cardiovascular system in the embryo.

## Discussion

PIBF1-intact trophoblast cells led to their syncytialization and played a role in the formation of multiple layers within the interhaemal membrane, which is essential for the proper development of the labyrinthine structure through PIBF1 derived from SynT. Under the influence of the placenta harboring properly syncytialized trophoblasts, the abnormalities observed in *Pibf1*-null early embryos were restored in vivo, and severe cardiovascular defects were also recovered by the functional placenta. Further in vitro evidence using human heart organoids proposes a possible role of PIBF1 in cardiovascular development. These data collectively indicate the essential roles of SynT in primary placentation and embryo survival, and imply the possible existence of placenta-derived factors that affect embryogenesis, including cardiovascular development.

We sought to determine what makes the placenta a functional unit at such an early period, represented as the proper development of trophoblasts and lipid transport-associated signatures of the placenta in mice[35]. We demonstrated that SynT is a key trophoblast subtype that contributes to primary placentation and early cardiovascular

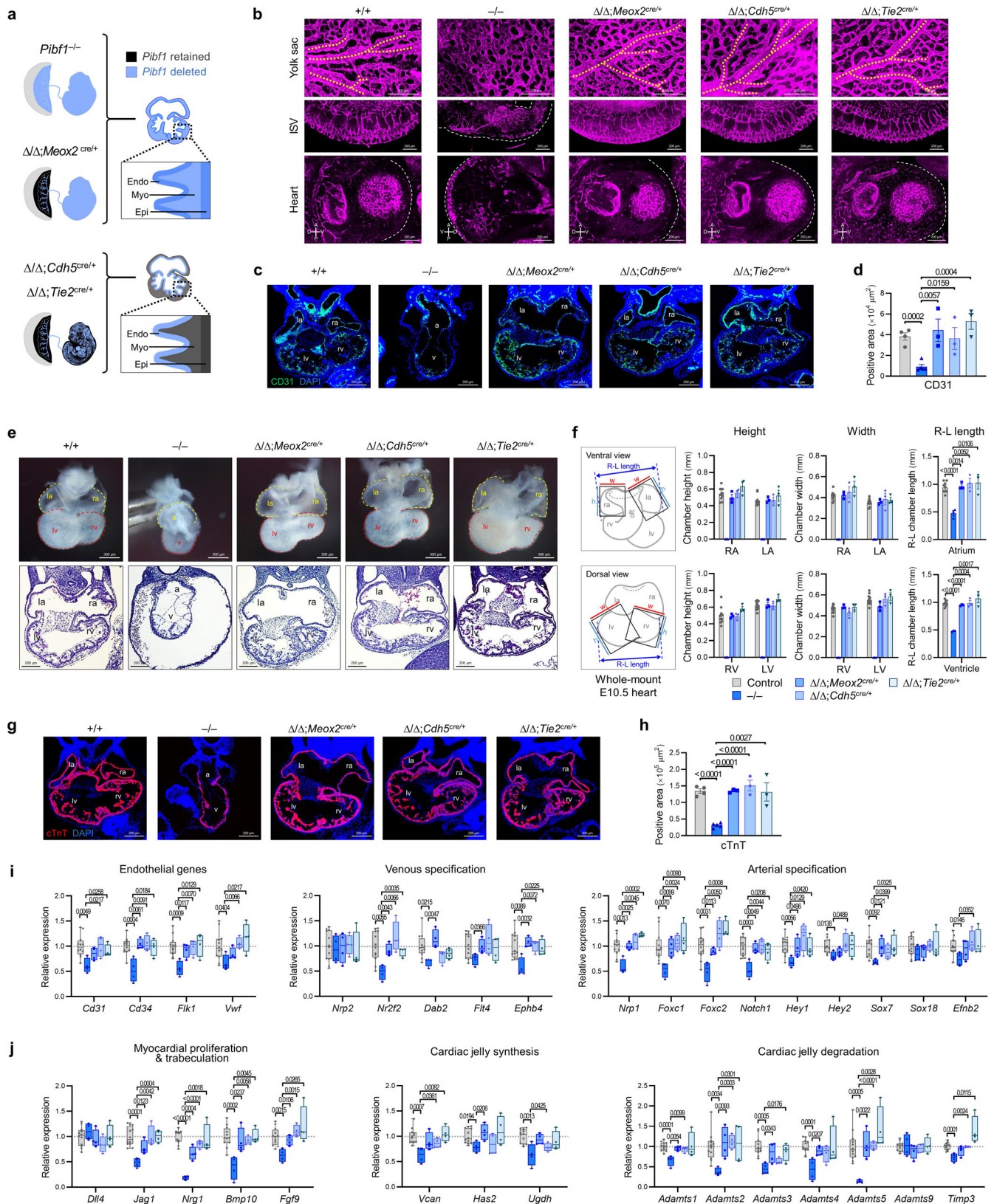

development. Disrupted cell-cell fusion in SynT-II is a significant finding of *Pibf1*−/− interhaemal membrane and a defective SynT-I, where ultrastructural anomalies are also detected. This phenomenon is similarly seen in *Ly6e* KO[59] or *Atp11a* KO[60] placentas, where unfused SynTs affect the morphologies of adjacent cells within the interhaemal membrane. In line with our findings, a recent study reported that

defective SynT-I is a shared phenotype among mutants with mid-gestational congenital heart disease (CHD)[16].

Interestingly, mice deficient with *Synb*, a fusogenic gene required for SynT-II fusion, showed an altered syncytialization in SynT-II but resulted in viable offspring, albeit with a slightly reduced number[61]. This suggests that the defective formation of SynT itself

**Fig. 5 | Placenta with properly syncytialized trophoblasts maintains embryonic development of vasculature and heart in mice. a** Schematic diagram of the lineages containing *Pibf1* deletion in the heart with whole-body *Pibf1* KO (*Pibf1⁻/⁻*), epiblast-specific cKO (cKO*^Meox2^*), or endothelial-specific cKO (cKO*^CdhS^*, cKO*^Tie2^*). Endo, endocardium, Myo, myocardium, Epi, epicardium. **b** Whole-mount CD31 stain of the yolk sac, intersomitic vasculature (ISV), and heart in *Pibf1* mutant embryos. Yellow dotted lines in the images of the yolk sac mark continuous vessels with a diameter ≥ 40 µm. Experiments in **b** are representative of three independent experiments with similar data. Immunostaining of CD31 in histological sections (**c**) and quantification of CD31-stained areas (**d**) in *Pibf1* mutant hearts. a, atrium. la and ra, left and right atrium. v, ventricle. lv and rv, left and right ventricle. *n* = 4, 5, 3, 3 in each group. **e** Whole-mount images and histological examination of *Pibf1* KO and cKO hearts. White lines indicate the maximum length from edge to edge of the right and left chambers. Note that left or right chambers were not specified in the *Pibf1*

KO heart. **f** Comparison of the chamber size of *Pibf1* KO and cKO hearts as measured from the whole-mount images. *n* = 11, 3, 3, 3 in each group. Immunostaining of cTnT in histological sections (**g**) and quantification of cTnT-stained areas (**h**) in *Pibf1* mutant hearts. *n* = 4, 5, 3, 3 in each group. **i, j** Expressions of markers for coronary vascular development (**i**) and cardiogenesis (**j**) in *Pibf1* KO and cKO hearts. *n* = 12, 4, 4, 4 in each group. Experimental samples were obtained on E10.5 unless indicated otherwise. Transcript expression data (**i** and **j**) were normalized to *Gapdh* and expressed relative to that of the control group. Data are presented as mean ± SEM (**d**, **f**, and **h**) or box-and-whisker plots where the interquartile range (IQR) of boxplot is between Q1 and Q3, the center line indicates the median value, whiskers of boxplot is extended to the maxima and minima, and maxima is Q3 + 1.5×IQR and minima is Q1 − 1.5×IQR (**i** and **j**); *P < 0.05; **P < 0.01; ***P < 0.001 in one-way ANOVA. The number *n* represents biologically independent embryos or placentas in each group. Source data are provided as a Source Data file.

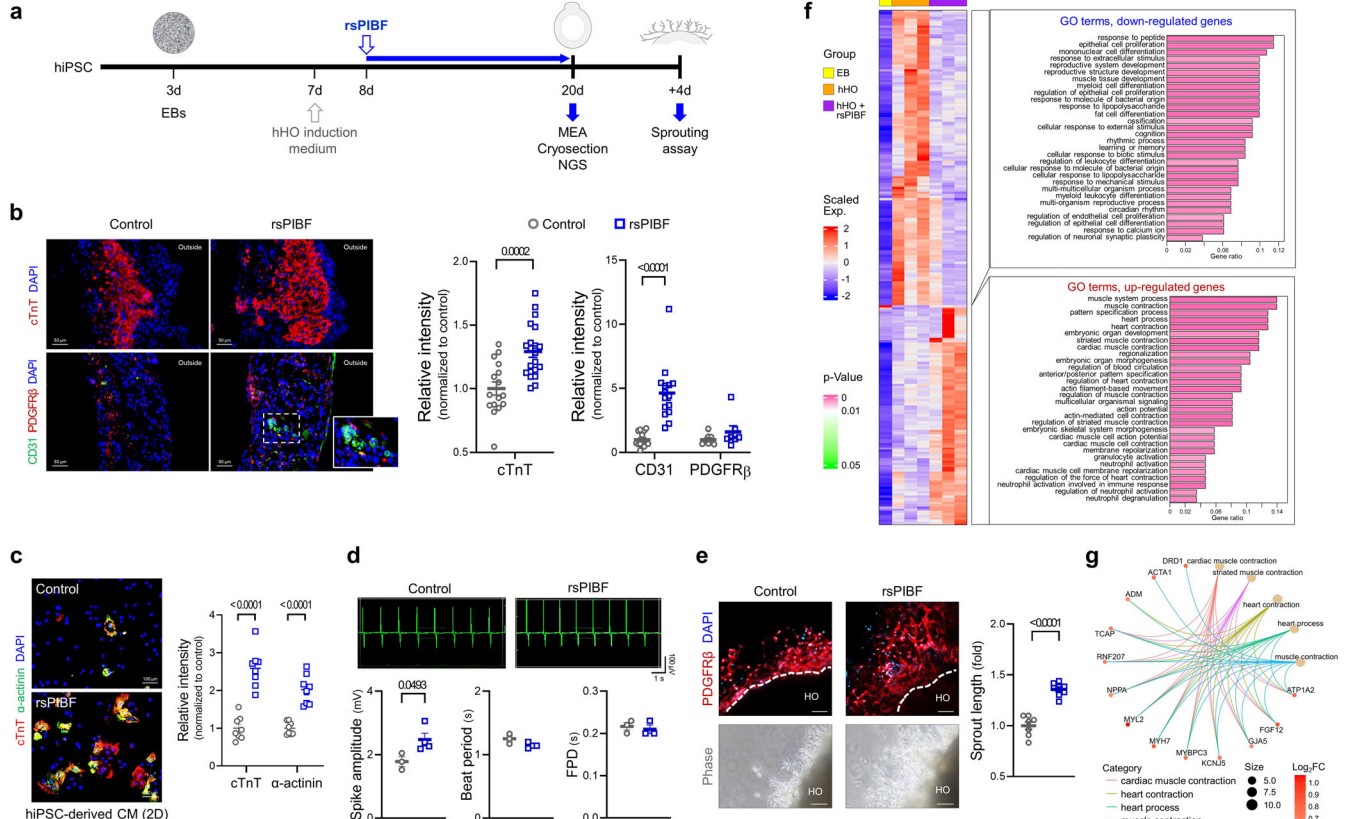

**Fig. 6 | PIBF1 facilitates the manifestation of cardiovascular features in human heart organoids. a** Schematic diagram of differentiation from iPSC into hHO and application of rsPIBF. **b** Immunostaining of cTnT, CD31, and PDGFRβ in rsPIBF-treated hHO. *n* = 16, 21 (cTnT), *n* = 13, 15 (CD31), *n* = 8, 7 (PDGFRβ) in each group. **c** Immunostaining of cTnT and α-actinin in rsPIBF-treated hiPSC-derived 2D-cultured cardiomyocytes. *n* = 8 per group. **d** Multi-electrodes array (MEA) in rsPIBF-treated hHOs. FPD, field potential duration. *n* = 3-4 per group. **e** Sprouting assay of rsPIBF-treated hHOs. Sprouts were viewed as phase-contrast and PDGFRβ-stained images. *n* = 7-8 per group. Scale bar: 100 µm. **f** Expression pattern of significantly up-or down-regulated genes in sPIBF-treated hHO compared to the non-treated group was visualized using a heatmap. Two clusters of genes were identified by hierarchical clustering based on their expression patterns, and representative GO

terms of each cluster were visualized using a barplot (width and color of the bars indicate gene ratio and p-value, respectively). EB, embryonic body. **g** The top 5 enriched GO terms (big dots) on significantly up-regulated genes by sPIBF treatment were visualized with their associated genes (small dots). The size of big dots indicates the number of associated genes. The color of small dots indicates the binary logarithm of the fold change of the genes by sPIBF treatment. The lines connecting the GO term and the associated gene were uniquely colored by the GO term. The intensity of fluorescence-labeled cells (**b**, **c**) and sprout length data (**e**) are expressed relative to control. Data are presented as mean ± SEM (**b-e**); *P < 0.05; ***P < 0.001; ****P < 0.0001 in two-sided Student's t-test (**b-e**). The *n* number represents biologically independent hHOs in each group. Source data are provided as a Source Data file.

may not be the sole event contributing to the early pathogenesis of CHD. In this regard, our data further delineate the roles of SynT in terms of communicating with adjacent placental vascular cells via secreted PIBF1 to promote the formation of the materno-fetal interface in the primary placenta. SynT-derived PIBF1 mediates the angiogenic properties of vascular ECs and perivascular pericytes,

which form the cellular structure of capillary wall in developing organs, including the heart[62], and also play essential roles in placental vasculature development[63–66]. Given that SynT recruits ECs and activates their angiogenesis signaling, it is highly likely that other SynT-derived angiogenic or vasculogenic factors also serve as organizers of the placental barrier.

In the genetic approach using a *Meox2*-cre driver line, the vascular interaction between placenta and embryo (chorioallantoic connection and umbilical cord vessels) was evident in both *Pibf1*$^{-/-}$ and *Pibf1*$^{Meox2}$ cKO conceptuses. However, the two conceptuses differed in terms of the existence of functional trophoblasts, especially SynT. Notably, we have demonstrated that SynT is essential for primary placentation and plays a crucial role in the process, which can have significant implications for embryo organogenesis, including the development of the cardiovascular system. Rescued abnormalities of *Pibf1*-null embryos under a normal SynT-bearing placenta might be a result of restored placental nutritional supply. However, considering the unique molecular and physiological aspects of PIBF1 as proven in this study, it is important to take into account the presence of placenta-derived factors beyond the classical placenta-heart concept which has been described as a parallel development of the two organs, and PIBF1 may be one of the factors that mediate this axis. Considering that the blood level of PIBF1 is detectable in both non-pregnant and pregnant women throughout gestational periods[24], it would be possible to detect PIBF1 in placental or embryonic circulation as well. This would allow for the elucidation of the range of PIBF1 levels in normal and pathological conditions of pregnancy. Likewise, SynT may be able to produce other factors essential for heart development in an endocrine manner. Identifying the placenta-derived factor critical for embryonic cardiovascular development is an important finding, considering its potential for clinical application in early pregnancy complications associated with fetal CHD.

However, several limitations could not be addressed in this study. As evidenced in our study, the state of *Pibf1* on the maternal side did not impact *Pibf1*-null embryonic lethality, indicating that maternally derived PIBF1 is unable to cross the placental barrier and reach embryonic circulation. The presence of placenta-resident cells or tissues that can act as a source of PIBF1, delivering it directly into the embryonic bloodstream and ultimately to developing organs, is crucial. SynT is considered one of the potential candidates for this role, as it is in direct contact with maternal blood space in humans and interacts with the endothelium of embryonic vasculature in mice. In this context, we have demonstrated that PIBF1 is a factor derived from SynT that is essential for primary placentation and plays a crucial role in in-vitro cardiogenesis. However, in this study, we did not prove that placentally produced PIBF1 enters the fetal blood circulation to directly exert effects on the developing fetal heart. Thus, it is possible that the dysfunctional placenta of Pibf1 mutants affects heart development in ways unrelated to secreted PIBF1. Another limitation of this study is the temporal discrepancy between the expression of Pibf1 in early embryos before E5.5[67] and the periods when cre recombinase becomes active, which occurs since E5.5 in the *Meox2*-cre driver line[40]. Although *Pibf1*-null embryos displayed a normal ICM/TE ratio in vitro (Supplementary Fig. 7b) and a normal gross phenotype until E8.5 in vivo (Supplementary Fig. 7c), the intracellular events in the blastocysts that can affect fate decisions of embryonic stem cells and subsequent lineages cannot be thoroughly determined solely by such gross phenotype detection. Thus, the role of Pibf1 in the peri-implantation embryonic cells before E5.5, a period outside of the *Meox2*-cre activity window, needs to be further investigated.

In addition, the ciliopathy-related phenotypes and the prenatal lethality of *Pibf1*$^{Meox2}$ cKO fetus are most likely embryonic proper in origin because extraembryonic lineages in mice lack primary cilium whereas epiblast-derived ones are ciliated[68]. Unlike that in mice, human trophoblasts have abundant primary cilia[69]. The primary cilia in the placenta are recently re-examined with their potential involvement in the pathogenesis of preeclampsia[70], and *PIBF1* KO TSC fails to differentiate into EVT; therefore, further work is needed to understand the relationship between PIBF1, ciliogenesis, and EVT development and to develop animal models that can recapitulate the findings of human trophoblasts and placental development with or without cilia.

Also, it is noteworthy that the expression level of *GCM1* was reduced following *PIBF1* KO in hTSC and its differentiated SynT. GCM1 is exclusively expressed in the placenta of humans and mice[71–73], and is classically considered a master regulator of SynT formation[74,75]. It has recently been shown that GCM1 also plays a role in EVT differentiation: when *GCM1* was knocked down in hTSC, there was a decrease in their ability to differentiate into both SynT and EVT[27], which is highly similar to that observed in *PIBF1* KO hTSC, suggesting that GCM1 may be an intrinsic factor in PIBF1-mediated trophoblast syncytialization. The relationship between PIBF1 and GCM1 needs to be further investigated.

In terms of early pregnancy management and complications, placenta-derived factors that can reach the embryo through fetoplacental circulation have garnered significant interest[76], as they can impact prenatal organ development and postnatal health as seen in the cases of serotonin[77] and ELABELA[78]. Our study uncovers the previously unknown role of PIBF1 as a placenta-derived factor and offers insights into the essential role of SynT in the placenta-heart axis (Fig. 7). Furthermore, considering the emerging field of neuroplacentology[79], the placenta-heart axis concept we have rediscovered here may be applied more broadly to other organs that develop simultaneously with the primary placenta.

## Methods
### Cell culture
BeWo cells (CCL-98, ATCC) were cultured in Dulbecco's modified Eagle's medium–Ham's F12 (DMEM/F12) medium (11320-033, Gibco) supplemented with 10% FBS (16000-044, Gibco). HUVECs were isolated from the human umbilical cord of donated pregnant women and were cultured in Medium 200 (M200500, Gibco) supplemented with large vessel endothelial supplement (LVES; A1460801, Gibco). HUAECs (8010, ScienCell Research Laboratories) were cultured in Endothelial Cell Growth Medium (1001, ScienCell Research Laboratories). Human pericytes from the placenta (Pl-PC; C-12980, PromoCell) were cultured in Pericyte Growth Medium 2 (C-28041, PromoCell). The culture medium for BeWo, HUVEC, HUAEC, and Pl-PCs contained 1% antibiotic-antimycotic solution (penicillin 100 U/ml, streptomycin 100 μg/ml, amphotericin B 25 μg/ml; 15240-096, Gibco).

Human trophoblast stem (hTS) cells (RCB4936) were provided by Riken BRC. The cells were cultured and maintained as previously described[80]. Briefly, hTS cells were seeded in 5 μg/ml collagen IV (354233, Corning)-coated dish and cultured in TS medium (DMEM/F12 (Gibco)) supplemented with 0.1 mM 2-mercaptoethanol (2-ME; 16141-079, Gibco), 0.2% FBS (Gibco), 0.5% antibiotic-antimycotic, 0.3% BSA (017-22231, Wako), 1% ITS-X supplement (094-06761, Wako), 1.5 μg/ml L-ascorbic acid (013-12061, Wako), 50 ng/ml EGF (053-07871, Wako), 2 μM CHIR99021 (038-23101, Wako), 0.5 μM A83-01 (035-24113, Wako), 1 μM SB431542 (031-24291, Wako), 0.8 mM valproic acid (227-01071, Wako) and 5 μM Y27632 (036-24023, Wako)], and the culture medium was replaced every two days.

All cells were cultured in a humidified atmosphere with 5% $CO_2$ at 37 °C.

### Establishment of *PIBF1* KO cell lines
To target the *PIBF1* gene in human cell lines, we used a lentiviral vector expressing CRISPR/AsCpf1 (Acidaminococcus sp. BV3L6 Cpf1), pY108 (Addgene plasmid #84739; http://n2t.net/addgene:84739; RRID:Addgene_84739). CRISPR RNAs specific for the human *PIBF1* gene were designed using Benchling (https://www.benchling.com), and the following oligomers were purchased from Macrogen, Inc.: 5′–agatGAAACAACAGTTCCTACGGA–3′ and 5′–aaaaTCCGTAGGAACTGTTGTTTC–3′ for *PIBF1* exon 2-specific crRNA; 5′–agatTTGAGACAGAAACAACTAGA–3′ and 5′– aaaaTCTAGTTGTTTCTGTCTCAA–3′ for *PIBF1* exon 4-specific crRNA. Each pair of oligomers was annealed and cloned into pY108 using BsmBI restriction enzymes (New England BioLabs).

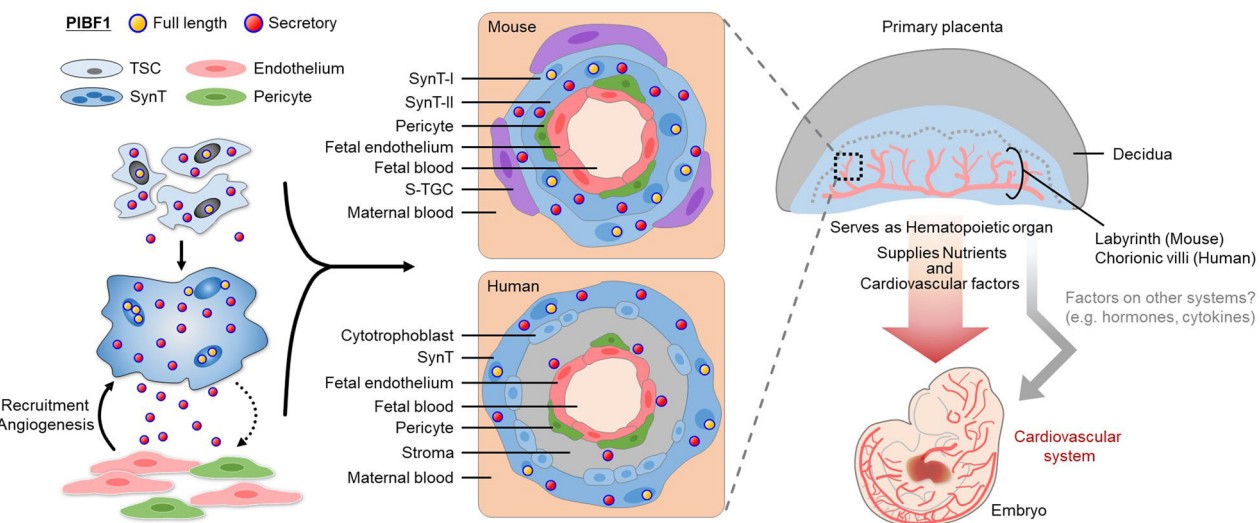

**Fig. 7 | PIBF1 is crucial for trophoblast syncytialization, and syncytiotrophoblast-derived PIBF1 mediates the formation of a multi-layered feto-maternal interface in the placenta, contributing to the proper embryo development, including the cardiovascular system.** Trophoblast stem cell (TSC) differentiates into syncytiotrophoblast (SynT) through cell-to-cell fusion via PIBF1. The resulting SynT communicates via secreted PIBF1 with adjacent vessel-forming cells such as endothelial cells and pericytes to promote their recruitment and angiogenesis, thereby establishing the interhaemal membrane that constitutes the materno-fetal interface of the primary placenta. SynT-I and SynT-II, syncytiotrophoblast type I and II; S-TGC, sinusoidal trophoblast giant cell.

BeWo and hTS cells were transduced with a 1:1 mixture of culture medium and lentiviral supernatant containing hPIBF1 exon 2 or exon 4 targeting Cpf1-expressing vectors in the presence of polybrene (2.5 µg/ml; 107689, Sigma). After selection with puromycin (2 µg/ml; A1113803, Gibco) for 48 h, the transduced and selected cells were dissociated and seeded in 100-mm dishes at the lowest possible cell concentration that allows for single-cell growth. The cell was then allowed to grow until it formed a visible colony, and the single-cell colonies ($n = 60$ for each targeted exon) were selected after being seeded in a 96-well plate.

After examining the PIBF1 expression of BeWo and hTSC colonies using Western blotting, the colonies were expanded to larger culture plates. For subsequent sequence analysis in *PIBF1* KO suspect colonies, the mutant PIBF1 locus was amplified by PCR reaction using the following primers: hPIBF1 exon 2, 5′-cctccccgcaggaactg-3′ and 5′-tgccgtttgaagacaataaagca-3′ (580 bp), hPIBF1 exon 4, 5′-tactgtcaggtttctgatttcctaa-3′ and 5′- aacaccattcaccatccatagc-3′ (443 bp). The PCR products from the colonies were cloned using a T-Blunt PCR Cloning kit (SOT02-K020, SolGent). The mutations were identified by Sanger sequencing analysis (Bionics, Seoul, Republic of Korea) of 20-25 clones from each colony. *PIBF1* KO TS cells were used at passages 6–8.

**Recombinant PIBF1**
Recombinant proteins of full length (rfPIBF; H00010464-P01) and C-terminal fragments of PIBF1 (rcPIBF; H00010464-Q01) were purchased from Abnova. Human monomeric Fc of immunoglobulin fused form of PIBF1 (1 ~ 298 a.a, rsPIBF), which was expressed and purified from CHO cell, was purchased from Ybiologics (Daejeon, Republic of Korea).

**Construction of the raPIBF1 expression plasmid.** For the production of raPIBF, cDNA for the N-terminal of PIBF1 (1-184 a.a) was synthesized at Cosmogenetech (Seoul, Republic of Korea). The cDNA was used as the template for polymerase chain reaction (PCR) analysis. The N-terminal PIBF1 (1–184 amino acids) was amplified from the cDNA template using the upstream primer 5′-CCA GAA TTC ATG TCT CGA AAA ATT TCA AAG GAG TC introduced an EcoRI restriction site, while the downstream primer 5′-AAG GTC GAC AAG GTC TTC ATC TTT TGT TTC TTA GAC C contained a SalI site. PCR was performed as follows:

95 °C for 4 min, followed by 30 cycles at 95 °C for 25 s, 65 °C for 25 s, and 70 °C for 60 s; the reaction was then held at 70 °C for 20 min and finally cooled to 4 °C for standby. Subsequently, the PCR products of PIBF1 were double-digested with restriction enzymes, cloned into the pET-28a vector, and transformed into E. coli DH5α competent cells.

**Induction and purification of raPIBF1.** The pET-28a-raPIBF1 recombinant plasmid was transformed into E. coli BL21(DE3) competent cells using a heat-shock method at 42 °C for 100 sec and then transferred to ice for 5 min and cultured in LB medium containing kanamycin (50 µg/ml) at 37 °C for 12 h. The transformed E. coli suspension was inoculated into culture tubes containing 5 mL LB medium (50 µg/ml kanamycin). Following incubation at 37 °C, the tubes were centrifuged at 13 x *g* until the optical density (OD$_{600}$) reached 0.7, isopropyl β-D-1 thiogalactopyranoside (IPTG) was added at a concentration of 0.1 mM. The LB media was cultured at 20 °C for 12 h to express the recombinant protein. The bacterial cells were harvested by centrifugation at 15,000 x g for 30 min and resuspended in 10 mL of 50 mM Tris-HCl (pH 8.5) buffer and separated on 10% SDS-PAGE gels to determine the solubility of raPIBF1. The raPIBF1 protein was purified using a 6xHis Tag and Ni-NTA affinity chromatography. Bacterial pellets were lysed using a lysis buffer (50 mM Tris-HCl (pH 8.5), 150 mM NaCl buffer) with sonication for 10 min at 4 °C. After centrifugation at 15,000 x *g* for 30 min, the resulting supernatant was loaded onto a Ni-NTA column that had been equilibrated with lysis buffer. After washing with lysis buffer, elution buffer (200 mM NaCl and 300 mM imidazole (pH 8.5)) was used for elution. The purified raPIBF1 proteins were dialyzed using a 10 kDa cutoff centricon with a PBS solution.

**BeWo experiments**
**Cell-to-cell fusion and fusion index quantification.** BeWo cells were incubated with 20 µM forskolin (3828, Cell Signaling Technology) to induce cell fusion. After 48 h, the cultured cells were stained with Hoechst 33342 (H1399, Invitrogen) and 2 µM Di-8-ANEPPS (D3167, Invitrogen) for 30 min, and subsequently imaged by Zeiss LSM 880 Fast Airyscan Confocal system (Carl Zeiss, Jena, Germany).

In fluorescent images of six randomly selected fields from three or more independent experiments, the number of nuclei within non-fused or fused cells was counted. Cells with more than two nuclei were

identified as fused cells, and the fusion index in each group was calculated as follows: (the number of nuclei in fused cells) / (total number of nuclei). The fusion index data were expressed relative to the forskolin-untreated WT control.

**PIBF1 restoration in PIBF1 KO BeWo cells.** The fragment of the cDNA region encoding the amino acids 1-757 (fPIBF) or 1-224 + 684−757 (sPIBF) of the human PIBF1 (NP_006337.2) was synthesized and inserted into pCAGGS expression vector[81] by Cosmogenetech (Seoul, Republic of Korea). The resulting construct DNA (1 μg/ml) expressing fPIBF or sPIBF or empty vector control was transfected into *PIBF1* KO BeWo cells using Lipofectamine LTX with PLUS reagent (15338100, Invitrogen) following the manufacturer's instruction.

**Treatment with recombinant PIBF1 on PIBF1 KO BeWo cells.** *PIBF1* KO BeWo cells were incubated with rsPIBF (400 ng/ml) or rfPIBF (400 ng/ml) simultaneously with 20 μM forskolin. After 48 h, the fusion index was obtained as described above.

**In vitro angiogenesis assays**
For migration assay, three scratches were made using a 200-μl pipette tip across the plates. After washing, cells were incubated without or with rsPIBF, rfPIBF, raPIBF, and rcPIBF at a dose of 20, 50, 100, and 200 ng/ml for 9 h (HUVEC), 14 h (HUAEC), or 8 h (Pl-PC). VEGF (20 ng/ml; 293-VE, R&D Systems) and PDGF-BB (50 ng/ml; PeproTech, 100-14B) were used as a positive control for ECs and Pl-PCs, respectively. The images of cell migration at 0 h and endpoint were analyzed with their migrated area using ImageJ, and the cell migration rate was calculated as follows: [(area at 0 h)−(area at endpoint)]/(area at 0 h) × 100 (%).

For tube forming assay, ECs were labeled with 10 μM CellTracker Green CMFDA Dye (C7025, Invitrogen) for 30 min, and then were seeded onto a Matrigel-coated plate and incubated without or with the recombinant PIBF1 isoforms at a dose of 50, 100, 200, and 400 ng/ml for 9 h. VEGF (50 ng/ml) was used as a positive control. The images were captured using EVOS M5000 Imaging System (Invitrogen), and the capillary-like structures formed by HUVECs and HUAECs were analyzed using ImageJ software (NIH).

To investigate rsPIBF-induced changes of angiogenesis-related signaling cascades in HUVECs, HUVECs were incubated with VEGF (20 ng/ml), rsPIBF (200 ng/ml), or vehicle for 30 min, and the cells were processed for Western blot analysis.

**hTS cell experiments**
**Differentiation of hTS cells.** Differentiation of TS cells into SynT or EVT was induced as previously described[80]. Briefly, for SynT differentiation in a 2D [SynT(2D)] culture, TS cells were seeded in a collagen IV-coated dish and cultured in SynT(2D) medium [DMEM/F12 supplemented with 0.1 mM 2-ME, 0.5% antibiotic-antimycotic, 0.3% BSA, 1% ITS-X, 2.5 μM Y27632, 2 μM forskolin, and 4% KnockOut™ Serum Replacement (KSR; 10828028, Gibco)]. The cells were replaced with a fresh medium on day 3 and analyzed on day 6. In the case of SynT differentiation in a 3D [SynT(3D)] culture, TS cells were cultured in SynT(3D) medium [SynT(2D) medium supplemented with 50 ng/ml EGF]. The cells were treated with fresh medium on day 3 and analyzed on day 6.

For EVT differentiation, TS cells were seeded in a collagen IV-coated dish and cultured in EVT medium [DMEM/F12 (Gibco) supplemented with 0.1 mM 2-ME, 0.5% antibiotic-antimycotic, 0.3% BSA, 1% ITS-X, 7.5 μM A83-01, 2.5 μM Y27632, 4% KSR, and 100 ng/ml NRG1 (26941, Cell Signaling Technology)] and Matrigel was added to the medium at a final concentration of 2%. At day 3, the cells were replaced with EVT medium lacking NRG1 and supplemented with Matrigel at a final concentration of 0.5%. At day 6, when the cells reached ~80% confluence, the cells were passaged and cultured in the EVT medium

lacking NRG1 and KSR with 0.5% Matrigel for an additional two days and analyzed on day 8.

In addition, the viability of TSCs was measured using Ez-cytox cell viability assay (EZ-1000, DoGenBio) according to the manufacturer's instructions.

**Engraftment of TS cells.** $10^7$ TSCs were suspended in 200 μl of a 1:2 mixture of Matrigel and DMEM/F12 containing 0.3% BSA and 1% ITS-X, and then subcutaneously injected into 6- to 8-week-old severe immune-deficient (SID) male mice. Lesions, urine, and serum were collected seven days after injection.

**Trophoblast-endothelial transwell co-culture**
A trophoblast-endothelial co-culture system was adapted with a slight modification of the previous method[82]. Briefly, hTS cells were plated in the bottom of 24-well chambers coated with 2.5 μg/ml collagen IV and were cultured in TS or SynT(2D) medium for 5 days, with a fresh medium change on day 3. BeWo cells in the bottom of the same 24-well chambers were transfected with fPIBF-expressing and/or sPIBF-expressing constructs or control vector for 24 h, and then stimulated with forskolin for 48 h.

After 5 days of differentiation or 48 h of forskolin stimulation, the culture media were replaced with a 1:1 mixed medium of Medium200/LVES and modified SynT(2D) (SynT(2D) medium lacking EGF, Y27632, and forskolin, which can affect endothelial invasion capacity[83–87]) in HUVEC-SynT, or Medium200/LVES and DMEM/F12/10% FBS in HUVEC-BeWo.

Concurrently, HUVECs were placed for 24 h in an 8-μm pore size transwell insert (CLS3422, Corning) coated with Matrigel (354234, Corning), and then the endothelial inserts were transferred onto the differentiated SynT or forskolin-stimulated BeWo cultures. After 24 h of co-culture, HUVECs that invaded the lower surface of the membrane were stained with crystal violet. The number of invaded cells was counted in six fields randomly selected from each sample using ImageJ software (NIH).

**MRM analysis**
**MRM method establishment for PIBF1 quantification.** Representative tryptic peptide for PIBF1 was determined to be ELLHNIQLLK (molecular weight: 1219.72 Da, N-terminus of PIBF1 (55-64 aa)) using PeptideaAltas (http://www.peptideatlas.org/) archive. Stable isotope-labeled internal standard (SIS) peptide for ELLHNIQLLK (molecular weight: 1227.72 Da) was synthesized by incorporating a heavy isotope labeled lysine at the C-terminal (L-*Lysine*-13C6,15N2) with 98% purity (Peptron, South Korea) to perform multiple reaction monitoring (MRM) analysis. A list of possible b- and y-series product ions for a doubly charged peptide of ELLHNIQLLK was generated theoretically using Skyline software (64 bits, version 19.1.0.193, University of Washington, MacCoss Lab) with the m/z range from 300 to 1,400. Those theoretical transitions were optimized with default instrument parameters of HPLC system (1290 Infinity, Agilent Technologies) and triple quadrupole mass spectrometry (MS) with a jet stream ESI source (6495 Agilent, Agilent Technologies). The MS source settings were set as below: gas temperature was 250 °C, gas flow was 15 L/min, nebulizer was 30 psi, sheath gas temperature was 350 °C, sheath gas flow was 10 L/min, capillary was 3500 V for positive ion funneling, nozzle voltage was 300 V, iFunnel parameters for high-pressure RF was 90 V and low-pressure RF was 60 V. Mixture of light peptide and SIS peptide was injected directly into a reversed-phase analytical column (Zorbax Eclipse Plus C18 rapid resolution HD, 2.1 × 100 mm, 1.8-micron column, Agilent Technologies) with 40 °C oven temperature. The peptide sample was separated and eluted at 0.3 ml/min on a linear gradient of mobile phase B from 5% to 30% B in 19 min (Mobile phase A: water/0.1% FA, mobile phase B: acetonitrile/0.1% FA). The gradient was ramped to 70% B for 3 min and ramped to 90% for 3 min. And 5% B for 10 min to

equilibrate the column for the next run. Resulting in the top three most intense transitions for the light peptide (Q1: 407.2550 m/z (triple charged), Q3: 501.3395 m/z (y4, singly charged), 433.2663 (y7, doubly charged) and 489.8084 (y8, doubly charged)) and SIS peptide (Q1: 410.2550 m/z (triple charged), Q3: 509.3537 (y4, singly charged), 437.2734 (y7, doubly charged) and 493.8155 (y8, doubly charged)) were selected for collision energy optimization and retention time determination, respectively. Targeted MS acquisitions were performed using 3-min detection windows, a 100-msec cycle time. MRM was performed using 6 points of concentration samples ranging from 0 to 50 fmol with a blank matrix to determine the response curve. The limit of detection (LoD) using 3 times of standard deviation plus the average peak area of three blank runs and the limit of quantitation (LoQ) using 10 times of standard deviation plus the average peak area of three blank runs were determined using the reverse response curve.

**Preparation of cell lysates and conditioned media for MRM analysis.** hTSC, SynT, and EVT were incubated in the media without BSA for 24 h. Cell pellets were suspended in lysis buffer [5% SDS, 50 mM triethylammonium bicarbonate (TEAB), 1× Halt™ protease inhibitor cocktail (78429, Thermo Fisher Scientific), pH 8.5] and were boiled at 80 °C for 10 min. Supernatants were harvested by centrifugation at 18,000 × g for 20 min. Conditioned media (CM) of 1 ml from each cell were lyophilized to reduce the volume. Each dried pellet from the lyophilized CM was suspended in 300 µl of lysis buffer. The protein concentration of each supernatant from cell lysate and CM was measured with a BCA protein quantification kit (Pierce™ BCA Protein Assay Kit; 23225, Thermo Fisher Scientific). A 100 µg aliquot of proteins was digested using suspension-trapping (S-Trap) mini columns (CO2-mini-80, ProtiFi) as per the manufacturer's protocol with Lys-C/trypsin mixture (V5071, Promega) and incubation for 16 h at 37 °C[88]. The eluted peptide mixture was lyophilized using a cold trap and stored at −80 °C until use.

**Quantification of PIBF1 using MRM.** The peptide mixture from the cell lysate and the CM was dried and reconstituted using 0.1% FA to be 1 µg/ul. SIS peptide was spiked to be 50 fmol and targeted MRM analysis was performed using established MRM methods. Quantity information for each transition was extracted using Skyline, and the concentration of PIBF1 was calculated using a peak area of 50 fmol SIS peptide.

**Measurement of hCG**
The presence of hCG in the mouse urine sample was qualitatively detected using a commercial over-the-counter hCG pregnancy test kit. The quantitative level of hCG in the samples of mouse serum or conditioned media obtained from BeWo, SynT(2D), or SynT(3D) culture was measured using an ELISA kit (KA4005, Abnova) following the manufacturer's instruction.

**Human heart organoid (hHO) experiments**
**Generation of hHO and rsPIBF1 treatment.** iPSC-derived hHOs were generated as previously described[57]. Briefly, human iPSCs were cultured in mTeSR-TM1 medium (StemCell Technologies) containing 10 µM Y-27632 (1254, Tocris Bioscience) and 10% Matrigel (345277, Corning) for 3 days on a shaking incubator to form embryonic bodies (EBs). EBs were then cultured in RPMI1640 containing B27 supplement without insulin and 6 µM CHIR99021 (4423, Tocris Bioscience) for 2 days, and then in RPMI1640 containing B27 supplement without insulin and 2 µM C-59 (S7037, Selleckchem) for 2 days. On day 7, cells were cultured in Advanced MEM (AD-MEM) containing 1% GlutaMAX and 1% antibiotic-antimycotic (hHO medium).

To investigate the effect of rsPIBF on the development of hHO, cells were cultured in hHO medium (control group) or hHO medium supplemented with 50, 200, and 1000 ng/ml sPIBF (sPIBF group) on

day 8 (one day after hHO induction). The medium was replaced every two days until days 19-21.

**Sprouting assay.** For sprouting assay, a 20-µl cold drop of Matrigel containing control or rsPIBF-treated hHOs was solidified at room temperature for 5 min, added with AD-MEM, and cultured for 4-6 days on a shaking incubator with the medium replaced every two days.

**Multi-electrodes array.** For multi-electrodes array (MEA), after control or sPIBF-treated hHOs were loaded in the MEA device (Maestro Edge version; Axion BioSystems, GA, USA), raw waveforms representing the field potential (FP) of each group were recorded for 3 minutes in AxIS software. Three leading indicators of recorded FP (spike amplitude, beat period, and field potential duration) were analyzed using the Cardiac Analysis Tool and AxIS Metric Plotting Tool (Axion BioSystems)[57,58].

**Histology.** For immunostaining of hHOs, control or rsPIBF-treated hHOs were fixed in 4% PFA overnight at 4 °C, dehydrated in a sucrose gradient (15-30%), embedded in OCT compound, cryosectioned in 9 µm thickness, and stored at −80 °C until following procedures.

**RNA seq and Gene ontology (GO) analysis.** Total RNA was isolated from EBs (day 3), control, or sPIBF-treated hHOs (day 20) using a Trizol Reagent kit (15596026, Invitrogen). Next-generation sequencing (NGS) was conducted by LifeGenomics (Suwon, Republic of Korea). Differentially expressed gene analysis was performed using DESeq2 version 1.36.0[89]. The genes up- or down-regulated over 1.5 times with adjusted p-value (Benjamini-Hochberg correction) were regarded as significant. GO analysis was performed using clusterProfiler version 4.4.4[90]. The statistical analysis and visualization were performed under the R version 4.2.2 and R studio Build 353 environments.

**hiPSC-derived cardiomyocyte experiments**
A two-dimensional culture of hiPSC-derived cardiomyocytes was performed as previously described[58] with slight modifications. Briefly, hiPSCs were seeded on Matrigel-coated dishes with mTeSRTM1 (StemCell Technologies) media containing 10 µM Y27632. The mTeSRTM1 media was changed every day until the confluency of hiPSCs reached 90%. First, to differentiate into the mesodermal lineage, hiPSCs were cultured for 1 day in RPMI1640 media containing B27 supplement without insulin (B27 media) supplemented with 6 µM CHIR99021 (Day 0). The next day (Day 1), it was replaced with B27 media containing 3 µM CHIR99021 and cultured for 1 day. Next, for induction into cardiac progenitors, B27 media containing 2 µM C59 (Sellckem), a Wnt signal inhibitor, was replaced on day 2. The next day (Day 3), it was replaced with B27 media containing 2 µM C59 and cultured for 1 day. From day 4 to day 10, the Advanced MEM (Gibco) medium (AD-MEM) containing 1% penicillin/streptomycin and 1% GlutaMAX (Gibco) was replaced every 2 days until the differentiated cells began beating.

To investigate the effect of rsPIBF on the differentiation of cardiomyocytes, sPIBF (200 ng/ml) was treated for 6 days (Day 4-10). After confirming beating, cells were detached using TrypLE (Gibco) and reseeded on Matrigel-coated dishes. On day 12, the hiPSC-CMs were fixed in 4% PFA for 20 min and stained with antibodies against cTnT and α-actinin, as listed in Supplementary Data 6.

**Mice**
All animal experiments were performed per the Korean Ministry of Food and Drug Safety guidelines. Experimental procedures were reviewed and approved by the Institutional Animal Care and Use Committees (IACUC) of Asan Institute for Life Sciences (approval number: 2019-12-217). All mice were maintained in the specific pathogen-free facility of the Laboratory of Animal Research at Asan

Medical Center (AMCLAR) at an ambient temperature of $23 \pm 1\,°C$ and $50 \pm 5\%$ humidity, with ad libitum access to a commercial CHOW diet (Purina Laboratory Rodent Diet, 38057) and standard laboratory water and a 12-h dark/light cycle, which was applied throughout all mouse studies. The protocol for conventional knockout of *Pibf1* (*Pibf1*[em1Hwl]/J)[91] and *TrpS3* (*TrpS3*[em1Baek]/J)[92] in mice was described previously. ICR and SID (NOD/J-*Prkdc*[em1Baek]*Il2rg*[em1Baek]/J) mice were purchased from Orient-Bio (Gyeonggi-do, Republic of Korea) and GEM Biosciences (Chenogju, Republic of Korea), respectively. *Meox2*-cre (B6.129S4-*Meox2*[tm1(cre)Sor]/J, 003755), *Vav1*-iCre (B6N.Cg-*Commd10*[Tg(Vav1-icre)A2Kio]/J, 018968), *Cdh5*-cre (B6;129-Tg(Cdh5-cre)1Spe/J, 017968), and R26R reporter (B6.129S4-*Gt(ROSA)26Sor*[tm1Sor]/J, 003474) mice were purchased from the Jackson Laboratory (Bar Harbor, ME, USA). *Tie2*-cre (B6.Cg-Tg(Tek-cre)1Ywa/J, JAX 008863) and *Il17a* knockout (B6.129P2-*Il17a*[tm1Yiw]) mice were generously provided from Dr. Chang Hoon Ha and Dr. Kyung Jin Lee, respectively, from Asan Medical Center (Seoul, Republic of Korea).

To obtain tissue samples of conceptus at different gestational ages, 7-week-old female mice were mated at a 1:1 ratio with male mice in the evening. The day of the vaginal plug formation was considered embryonic day (E) 0.5. Pregnant mice were then randomly assigned to groups with different experiments, euthanized at the designated time point, and the tissue samples from the different parts of the conceptus were snap-frozen and stored at $-80\,°C$ or fixed in 4% PFA for further analysis.

## Generation of *Pibf1* cKO mice

To generate single guide RNAs (sgRNAs) used for floxed *PIBF1* mice, the following oligomers were purchased from Macrogen, Inc.: 5′−taggGCTATTTCAGTTTCTCTAAG−3′ and 5′−aaacCTTAGAGAAACTGAAATAGC−3′ for *Pibf1* intron 1-specific sgRNA; 5′−taggACTCTATTTCTGCTTCCCGA−3′ and 5′−aaacTCGGGAAGCAGAAATAGAGT−3′ for *Pibf1* intron 17-specific sgRNA. Each pair of oligomers was annealed and cloned into pUC57-sgRNA vector, a gift from Xingxu Huang (Addgene plasmid # 51132; http://n2t.net/addgene:51132;RRID:Addgene_51132), using BsaI restriction enzymes (New England BioLabs), and the cloned vectors were used as a template for sgRNA in vitro transcription. The following single-stranded deoxynucleotide donor DNAs were purchased from Integrated DNA Technologies, Inc.: 5′−tgtaactcccttgtttaaatacgacacgatttagtaaaatagttatagaactagctatttcagtttctctATAACTTCGTATAATGTATGCTATACGAAGTTATaagaggaacgctaaagaaggaggtctgtgtggtgggggggaacaggaagagcttcagtacaatccgcctc−3′ for inserting a loxP into *Pibf1* intron 1; 5′−caattctcctccttcaatatattgtgaatttctgttgaactgtgaaacagataactcatttctgcttccATAACTTCGTATAATGTATGCTATACGAAGTTATcgagggctgaaattacaaccatgaaccaccatacccaggtgaaactatttttttttagagatataacata−3′ for inserting a loxP into *Pibf1* intron 17. Microinjection of fertilized eggs and their transfer to foster mothers were performed as previously described[91]. Genomic DNA samples obtained from the tail of embryos or 3-week-old pups were genotyped by using the primers listed in Supplementary Data 5.

## Embryo outgrowth culture

Blastocysts were individually cultured on 0.1% gelatin-coated multi-well dishes in KnockOut DMEM (10829018) supplemented with 20% Knock-Out Serum Replacement (10828010), 2 mM GlutaMAX (35050061), 0.1 mM non-essential amino acids (11140050), 50 U/ml penicillin-50 µg/ml streptomycin (15070063), 55 µM 2-mercaptoethanol (21985023), and 5 ng/ml leukemia inhibitory factor (A35933) (all reagents were purchased from Gibco) at $37\,°C$ and 5% $CO_2$. On the 7th day of culture, outgrowths were photographed, collected, and subsequently genotyped by PCR.

## Morphometric analysis of embryo and heart

The images of embryo outgrowths and dissected parts of the conceptus, including the yolk sac, embryo, and heart, were photographed using a ProgRes C5 camera and Capture Pro software (Jenoptik Optical Systems, Jena, Germany). Quantitative measurements of the areas of inner cell mass and trophectoderm in the outgrowths, the crown-rump length in whole-mount embryos, and the width and height of heart chambers in whole-mount hearts were carried out using ImageJ software (NIH).

## Histology

**Morphometric analysis of placenta.** H&E-stained placental sections were examined for the areas of three main layers of the placenta (decidua basalis, junctional zone, labyrinth) and those of labyrinthine vascular spaces under a light microscope. Fetal blood sinuses were distinguished from maternal ones by their primitive nucleated erythrocytes. The blood sinuses in the labyrinth were examined and compartmentalized in three random non-overlapping ×40 fields of each sample. The extent of each area was quantified using ImageJ software (NIH).

**Immunostaining.** Heat-induced antigen-retrieved tissue sections were subjected to Vectastain Elite ABC-HRP system (PK6101, Vector Labs) for immunohistochemistry according to the manufacturer's instructions. Primary antibodies were as follows: anti-Ki67 (1/10000; ab15580, Abcam), anti-Laminin (1/750; L9393, Sigma-Aldrich), anti-Perforin (1/200; ab16074, Abcam), and anti-PIBF1 (1/500; PMID:14634107[34]). In the case of staining Perforin, peroxidase-conjugated donkey anti-rat IgG (712-035-150, Jackson ImmunoResearch Laboratories) was applied as a secondary antibody. Labyrinthine Ki67-positive trophoblast cells or decidual Perforin-positive cells were counted in three randomly selected ×40 fields of each sample.

For immunofluorescence staining, antigen-retrieved tissue sections were subjected to primary antibodies, subsequently captured by Alexa Fluor-conjugated secondary antibodies. In vitro samples, including fixed cells and hHO cryosections, were permeabilized with 0.1% Triton X-100 and subjected to antibodies. The sections were mounted with DAPI (D1306, Molecular Probes), and examined under Zeiss LSM 880 confocal microscope system (Carl Zeiss, Jena, Germany). CD31, cTnT, or PDGFRβ-stained areas were quantified using ImageJ software (NIH). The antibodies used for immunofluorescence staining are listed in Supplementary Data 6.

**Alkaline phosphatase activity.** OCT-embedded placentas were 6 µm-thick cryosectioned, incubated in alkaline phosphatase buffer (0.1 M NaCl, 0.1 M Tris-HCl, 0.05 M $MgCl_2$, 0.1% Tween 20, pH 9.5) containing 5-bromo-4-chloro-3-indolyl-phosphate (BCIP) and nitro blue tetrazolium (NBT) color development substrate (S3731, Promega) according to the manufacturer's instructions, and then counterstained with nuclear fast red.

**Transmission electron microscopy.** Isolated placentas were fixed in 2.5% glutaraldehyde at $4\,°C$ for 4 h, fixed in 1% $OsO_4$ for 1 h at room temperature, and then embedded in Epon. Semithin sections (500 nm) were stained with toluidine blue, and the thickness of interhaemal membranes was measured in five randomly selected ×100 fields of each sample. Ultrathin sections (60 nm) were contrasted with uranyl acetate and lead citrate and observed using a Hitachi H-7100 transmission electron microscope (Hitachi, Nackashi, Japan) at 80 kV.

**In situ hybridization.** For in situ hybridization (ISH) of mouse *Pibf1*, samples were fixed in 10% neutral-buffered formalin, embedded in paraffin, and sectioned into 5-µm-thick slices on SuperFrost Plus adhesion slides (J1800AMNZ, ThermoFisher Scientific). Following the manufacturer's instructions, a chromogenic RNA ISH assay was performed using the RNAscope 2.5 HD detection kit (Brown) for FFPE tissues (322371, Advanced Cell Diagnostics). RNAscope probes (all from Advanced Cell Diagnostics) were used for mouse *Ppib* (313911),

bacterial *dapB* (310043), and mouse *Pibf1* (408871) was used as the positive control, negative control, and target probe, respectively.

## Measurement of fatty acids

Gas chromatography coupled with mass spectrometry (GC/MS)-based measurement of placental and embryonic fatty acids (FAs) was previously described[35]. Quantified data were expressed as a relative amount (%) to the wild-type group.

## Laser micro-dissection

Dissecting the placenta into three main layers (decidua basalis, junctional zone, labyrinth) using laser micro-dissection of frozen sections was previously described[35].

## Quantitative real-time PCR

cDNA synthesized with the SuperScript III First-Strand synthesis system (18080-051, Invitrogen) was employed as a template for quantitative real-time PCR reactions using the iQ SYBR Green Supermix (170-8882AP, Bio-Rad) and the primer sets (Supplementary Data 5) in the QuantStudio3 Real-Time PCR system (Applied Biosystems, CA, USA). Data were normalized by an internal control gene, such as *RPS18*, *GAPDH*, and *mGapdh*, and expressed relative to the expression level of the wild-type or control group.

## In vitro fertilization and embryo transfer

Oocytes were collected from females superovulated with 5 IU pregnant mare serum gonadotropin (G4877, Sigma) and 5 IU human chorionic gonadotropin (CG10, Sigma) at a 48 h interval. The in vitro fertilized 2-cell stage embryos from *Pibf1* heterozygous were transferred to pseudopregnant wild-type ICR or *Il17a* knockout female mice. Embryos at E10.5 to E11.0 were collected, photographed, and genotyped by PCR.

## Fluorescence-activated cell sorting (FACS)

The mesometrial regions of conceptus covering decidua basalis were dissected and sampled at E8.5 or E10.5, chopped, and digested using 1 mg/ml collagenase type II (C6885, Sigma) and 1 U/ml DNase I (04536282001, Roche) at 37 °C for 30 min. The prepared single cells were Fc-blocked with anti-CD16/CD32 (553142, BD Biosciences) and then stained with antibodies listed in Supplementary Data 6. All incubation steps were carried out in D-PBS containing 2% FBS. Stained samples were assessed using the FACSCanto II system (BD Biosciences, CA, USA) and further analyzed using FlowJo software (Tree Star, OR, USA).

## Whole-mount X-gal staining

4% PFA-fixed isolated parts of conceptus were incubated in detergent buffer (2 mM MgCl$_2$, 0.02% IGEPAL, and 0.01% sodium deoxycholate in PBS, pH 7.4) at 4 °C overnight. Samples were then incubated in detergent buffer containing 4 mM potassium ferricyanide, 4 mM potassium ferrocyanide, and 1 mg/ml X-gal (B4252, Sigma) at 37 °C for overnight in the dark, post-fixed in 4% PFA overnight, and visualized under a stereo microscope. For histological analysis, X-gal-stained tissues were paraffin-embedded, 4 μm-thick cuts, and counterstained with nuclear fast red.

## Hematopoiesis experiments

**Colony-forming unit assay.** Dissected parts of conceptus at different gestational ages were mechanically minced and enzymatically digested by incubating them in D-PBS containing 2% FBS and 0.1% collagenase type I (C0130, Sigma) at 37 °C for 30 min. In the case of preparing adult bone marrow samples, the femur and tibia of 4-month-old male mice were dissected, and their proximal and distal epiphysis were cut. Bone marrow was collected by flushing them with 2% FBS-containing D-PBS and dissociated by passing through a 70 μm cell strainer. Cells were cultured in a Methocult methylcellulose medium (M3434, Stem Cell Technologies) according to the manufacturer's instructions. Colonies were morphologically distinguished and scored after 10-12 days of culture.

**Complete blood count with white blood cell differential.** Whole blood samples of 4-month-old male mice collected from the inferior vena cava were analyzed for hematologic parameters using ADVIA 2120i automatic blood analyzer (Siemens, Munich, Germany).

**Whole-mount CD31 staining.** 4% PFA-fixed embryos or yolk sacs were blocked in 5% goat serum, incubated with anti-CD31 (1:200; 553370, BD Biosciences), and subsequently with AF594-conjugated secondary antibody (1:200; A21209, Invitrogen) in the dark. The immuno-stained whole mounts were visualized under Zeiss LSM 880 laser scanning confocal microscope system (Carl Zeiss, Jena, Germany). All incubation steps were performed in PBTx (PBS with 0.1% Tween 20 and 0.1% Triton X-100) at 4 °C overnight.

**Western blotting.** Protein lysates from cell or tissue samples were prepared using cell extraction buffer (FNN0011, Invitrogen) supplemented with protease inhibitor cocktail (Sigma, P2714) and phenylmethylsulfonyl fluoride (0754, Amresco), loaded in SDS-PAGE gels, transferred onto polyvinylidene difluoride membrane (IPVH00010, Millipore), and immunoblotted with primary and secondary antibodies as listed in Supplementary Data 6. The signals were detected using a SuperSignal West Pico chemiluminescent system (34080, Thermo Fisher Scientific).

**Magnetic resonance imaging.** MR imaging was performed at a 7.0 T/ 160 mm Parmascan preclinical scanner on Paravision 6.0 and equipped with a 72 mm diameter volume transmission coil, and a surface receive coil (Bruker Inc., Germany). The embryos are then placed into sealed plastic containers in preparation and immersed in 4% formaldehyde solution in PBS during MR scanning. The high-resolution 2D rapid acquisition with relaxation enhancement (2D RARE) sequence, fast spin-echo based T2 weighted (T2w) imaging was performed with axial (Ax) and sagittal (Sag) views – TR/TE = 4500/ 50 ms; average = 40; field of view (FOV) = 12 × 12 mm (Ax), 14 × 14 (Sag); RARE factor = 8; matrix = 256 × 256; slice thickness = 0.4 mm; slice number = 20 (no gap); and total scan time = 3 h 12 m. AsanJ-Morphometry™ is a dedicated software for ImageJ (NIH) used to indicate abnormalities in an embryo. The AsanJ-Morphometry software is open to the public for research purposes and downloadable at http://datasharing.aim-aicro.com/morphometry.

## Statistical analysis

Experiments were replicated at least two times, and no data were excluded from the analyses. The $\chi^2$ test was applied to test whether the genotype distribution of the mice from various breeding schemes obeyed the expected Mendelian inheritance patterns. Other data were analyzed by Student's two-tailed unpaired *t*-test or one-way ANOVA followed by Tukey's multiple comparison test. Statistical significance was determined using GraphPad Prism software (GraphPad Software, CA, USA). All data points in diagrams represent biological replicates. *P* values less than 0.05 were considered statistically significant.

## Reporting summary

Further information on research design is available in the Nature Portfolio Reporting Summary linked to this article.

## Data availability

The RNA-seq data for rsPIBF-treated hHO generated in this study have been deposited in the GEO database under accession code GSE229265. All other data are available within the main text and the Supplementary Information files. Source data are provided with this paper.

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

## Acknowledgements

This study was supported by grants from the National Research Foundation of Korea (NRF) funded by the Ministry of Science, ICT and Future Planning (2018M3A9D5A01087977 and 2020R1A2C2012912). This work was also supported by grants (2020-028, 2022-013, and 2023-062) from the Asan Institute for Life Sciences, Asan Medical Center, Seoul, Korea. We thank the core facilities of the GEAR Core, Confocal Microscope Core, Clinical Proteomics Core, Comparative Pathology Core, Electron Microscopy Core, Flowcytometry Core, Magnetic Resonance Core, Mammalian Genetics Core, Metabolomics Core, and Dep. of Laboratory Animal Research at the ConveRgence mEDIcine research center (CREDIT) at Asan Medical Center for the use of their shared equipment, services, and expertise. We are also grateful to Dr. B. Polgar and Dr. J. Szekeres-Bartho (University of Pecs, Hungary) for generously providing anti-PIBF1 antibodies.

## Author contributions

I.-J.B. designed research. J.G.L., J.-M.Y., S.-G.L., C.-Y.K., J.Y., K.K., J.K.R., and C.H.H. carried out in vitro assays. J.G.L., J.-M.Y., G.K., S.-A.C., Y.H.S., and H.J.Y. carried out in vivo experiments. S.-G.L. and C.-Y.K. performed heart organoid experiments. D.-C.W. and C.-G.P. performed imaging experiments. K.K. and Y.S.C. produced recombinant proteins. J.G.L., J.-M.Y., H.-Y.K., S.-G.L., C.-Y.K., K.K., Y.S.C. and I.-J.B. analyzed data. I.-J.B. supervised the project and secured funding. J.G.L., J.-M.Y., C.-Y.K., J.S.L., Y.S.C. and I.-J.B. wrote the original paper. S.H.O., Y.M.H., E.-J.H., J.K.S., H.-W.L., S.-W.L., K.-U.L., C.J.K., S.-Y.N., and all other authors revised and edited the paper.

## Competing interests

J.G.L., J.-M.Y., K.K., Y.S.C., and I.-J.B. have filed a provisional patent application in the Korea patent office (composition for preventing and treating pregnant complications related to placenta and/or fetal cardiovascular abnormalities by targeting PIBF1). All the other authors declare no competing interests.

## Ethics

HUVEC preparation was approved by the Institutional Review Board of the Asan Medical Center (approval ID: 2022-0664). Written informed consent was obtained from all participants or their families in accordance with the Declaration of Helsinki.

## Additional information

¹Asan Institute for Life Sciences, Asan Medical Center, Seoul 05505, Korea. ²Korea Mouse Phenotyping Center (KMPC), Seoul 08826, Korea. ³Department of Cell and Genetic Engineering, Asan Medical Center, University of Ulsan College of Medicine, Seoul 05505, Korea. ⁴Department of Stem Cell Biology, School of Medicine, Konkuk University, Seoul 05029, Korea. ⁵College of Veterinary Medicine, Konkuk University, Seoul 05029, Korea. ⁶NGeneS Inc, Ansan 15495, Korea. ⁷Department of Digital Medicine, Asan Medical Center, University of Ulsan College of Medicine, Seoul 05505, Korea. ⁸Department of Biomedical Engineering, Asan Medical Center, University of Ulsan College of Medicine, Seoul 05505, Korea. ⁹Department of Biochemistry and Molecular Biology, Asan Medical Center, University of Ulsan College of Medicine, Seoul 05505, Korea. ¹⁰Department of Pediatrics, Asan Medical Center, University of Ulsan College of Medicine, Seoul 05505, Korea. ¹¹Research Institute of Medical Science, Sungkyunkwan University School of Medicine, Seoul 06351, Korea. ¹²College of Veterinary Medicine, Chungnam National University, Daejeon 34134, Korea. ¹³College of Veterinary Medicine, Seoul National University, Seoul 08826, Korea. ¹⁴Department of Biochemistry, College of Life Science and Biotechnology, Yonsei University, Seoul 03722, Korea. ¹⁵Department of Radiation Oncology, Asan Medical Center, University of Ulsan College of Medicine, Seoul 05505, Korea. ¹⁶Department of Pathology, Asan Medical Center, University of Ulsan College of Medicine, Seoul 05505, Korea. ¹⁷College of Veterinary Medicine, Chungbuk National University, Cheongju 28644, Korea. ¹⁸Division of Allergy and Clinical Immunology, Department of Internal Medicine, Asan Medical Center, University of Ulsan College of Medicine, Seoul 05505, Korea. ¹⁹Present address: Biological Resources Research Group, Bioenvironmental Science & Toxicology Division, Gyeongnam Branch Institute, Korea Institute of Toxicology (KIT), Jinju 52834, Korea. ²⁰These authors contributed equally: Jong Geol Lee, Jung-Min Yon. ✉e-mail: yscho@amc.seoul.kr; ijbaek@amc.seou.kr

