## [Peer Review File · Nature Communications]

PIBF1 regulates trophoblast syncytialization and promotes cardiovascular developmentREVIEWER COMMENTS

Reviewer #1 (Remarks to the Author):

NCOMMS-23-25470-T

This manuscript by Lee et al investigates the role of progesterone immunomodulatory binding factor 1 (PIBF1) in trophoblast biology and its impact on fetal development, specifically on cardiogenesis. The authors find that PIBF1 has an essential role in facilitating trophoblast fusion into syncytiotrophoblast, and KO cells exhibit an overall decreased ability to differentiate at all. Somewhat surprisingly, this effect is seen both, with the soluble as well as with the intracellular form of PIBF1, but not when the sPIBF1 protein is supplied exogenously. Using recombinant versions of the protein, they also ascribe pro-angiogenic as well as heart differentiation-promoting properties to PIBF1 in Transwell and heart organoid culture experiments, respectively. The authors then assess KO as well as multiple conditional KO mice and find that PIBF1 ablation is embryonic-lethal around mid-gestation when placental function becomes critical. Perhaps most importantly, embryo-specific deletion of PIBF1 using the Meox-Cre driver rescues the early lethality and the placental phenotype, implying that PIBF1 is critical in the trophoblast compartment for early embryogenesis. They also generate Vav1-, Cdh5-, and Tie2-Cre conditional knockouts to assess endothelial and hematopoietic roles of PIBF1, but these conditional KOs develop normally, as expected from the Meox-Cre data.

Overall, this manuscript contains a large amount of data, albeit the informative value of a significant amount of them is relatively limited and the manuscript is inflated for data that do not add substance.

The most important point is that the authors' key conclusion is flawed and unjustified, i.e. that placental syncytiotrophoblast-produced PIBF1 is the signaling molecule that governs heart development, which is highlighted in the abstract and summary figure. This conclusion cannot be drawn, as PIBF1 is fairly ubiquitously expressed, and the mere fact that direct exposure of cells to PIBF1 promotes cardiac differentiation does not mean that this effect originates from a syncytiotrophoblastic source. Moreover, how PIBF1 would enter the fetal blood circulation, how it traverses the placental endothelial cell layer, or whether indeed syncytiotrophoblast-derived PIBF1 ever crosses the interhaemal barrier, remains unclear. Rescuing the placental defect restores all of embryonic development, and hence the effect on cardiogenesis may be indirect and due restored nutritional supply and/or due to normal tissue function of other embryonic cell types. The entire focus around this key conclusion needs to be completely revised and omitted.

Secondly, the KO phenotype is extremely severe with the entire embryo exhibiting profound developmental delay, lack of turning, and many other defects. Pericardial edema is a common feature of embryos with such severe phenotypes. Thus, the entire focus around the placenta-heart axis is not necessarily justified either, as development of all organ anlagen is impeded.

With these major flaws in mind, this reduces the data to the conclusion that PIBF1 is another example of the collection of genes that are essential in the placenta for embryonic development to term. This is important, but not in the strong and unidirectional interpretation offered by the authors.

Instead of focussing on the somewhat contrived focus on heart development, it would be far more interesting and legitimately justified by the data to investigate how PIBF1 facilitates syncytiotrophoblast fusion. If it was acting as a soluble protein, then the exogenous supply of sPIBF1 should have an effect, but this is not the case. Hence it must act intra-cellular to exert this function. What is the nuclear role of PIBF1, how would it even enter the nucleus (NLS?), or how would a centrosome-associated localization lead to cell fusion? These functional aspects are critical to elucidate in order to heighten the insights gained from this study beyond the observational level of fusion defects.

The Vav1-, Cdh5-, or Tie2-Cre mediated conditional knockouts are relatively meaningless as their lack of phenotype can be expected from the Meox-Cre results. As such, they inflate the manuscript but do not add substantial value. It is possible that the chronological sequence of events was such that the authors started by first generating those cKOs, before using the Meox-Cre. Any value beyond trophoblast-mediated rescue of embryonic lineages should be highlighted, but does not deserve the amount of space occupied in the figures and text.

Given the cilia-related mortality of embryo-specific KOs in late gestation (that is mentioned late and should be stated earlier), the centriole-related function of PIBF1 may be far more important for its normal role in development.

Additional points include:

- Where precisely in the placenta and embryo is PIBF1 expressed? These data are essential to infer cell types where it might be functioning.
- It is impossible to interpret the BeWo cell fusion photos. What is highlighted by the dotted lines, and how would one see that?
- How does the "heart organoid" model differ from traditional EB differentiation experiments towards cardiomyocytes? Is this terminology merely a 'sexing up' of the experimental strategy? Using "EB differentiation" as term is fine.
- The methods around the generation of CRISPR KOs in hTSCs must be expanded, as this is by no means a trivial approach with hTSCs being somewhat refractory to single cell cloning. The authors should expand on how these cells were generated, how many KO clones they obtained, how many were tested and how many exhibited the same phenotype. An N=3 should be used at a minimum.
- How were the recombinant proteins isolated and purified, and at what concentration were they used? Are the pro-angiogenic and heart differentiation-promoting effects dose-dependent?
- The display for qPCR (and other) datasets differs almost between every figure element, and the replicate data points of the controls cannot be made out in many of these graphs. Please homogenize the display to at least a common display type for each type of data.
- Given the fact that RNA-seq was performed on heart organoids, it is barely surprising that heart-related terms stand out in the GO terms. This should be reflected in the description of these data.

Reviewer #2 (Remarks to the Author):

In this study, the authors explore the role for trophoblast-derived PIBF-1 in differentiation of placental syncytiotrophoblast and the link to vascularization and cardiogenesis in development. Using cell culture approaches, it was demonstrated in a series of elegant experiments that PIBF-1 is essential for syncytialisation of human trophoblast cells. Co-culture and sPIBF-treatment approaches using human foetal endothelial cell lines and iPSC-derived heart organoids, the authors show the impact of PIBF on angiogenesis and cardiogenesis in vitro. The assessment of a Pibf KO mouse showed a remarkable co-occurrence of a syncytiotrophoblast layer II defect in the placenta and abnormal heart formation. A conditional embryo-specific KO demonstrated that a wildtype placenta not only recovered the placental defect, but rescued the heart defect as well. Together, this work provides a compelling mechanism linking the placenta-heart axis in development, a significant contribution to the field.

Major comments:

1. The data presented in section 2 of the results show the effect of syncytialised BeWo cells on the angiogenic properties on foetal endothelial cell lines (lines 137-175). a) Given that synT cells are adjacent to maternal endometrial vasculature, synT-produced PIBF-1 may also be involved in remodelling maternal vascular endothelium. Is there an appropriate cellular model to assess this aspect or can the authors comment on this? b) Can the authors also justify why syncytialised BeWo cells were used instead of syncytialised hTSCs, the latter of which may be more physiologically similar to primary cells?

2. Is there evidence that PIBF is secreted into human foetal circulation, facilitating its potential impact on cardiomyocyte differentiation, given the effects observed in vitro (Figure 3)?

3. The authors use Meox2-cre to generate embryo-specific cKOs, a nice strategy to demonstrate the importance of placental PIBF and the Cre appears to be fully penetrant by E13.5 in the genotyping provided (Supp Figure 9b). To exclude the possibility that early expression of Pibf in the embryo is impacting the embryonic phenotype, can the authors report at what stage in development the Meox2-Cre become expressed? Is there any evidence that Pibf is expressed in any embryonic cell type up to this stage?

4. The contrast to the synb KO in the discussion is valuable (lines 345-353). However, the lack of embryonic lethality seen in the synb KO may not necessarily exclude that PIBF from the synT cells is the primary contributor of CHD. It is my understanding that the synb KO does not completely lack the SynT-II compartment, rather that the cells just fail to fuse but may still differentiate. Hence it is possible in this model, PIBF is still sufficiently produced. Whereas the stainings in the Pibf null model suggest that SynT-II almost completely fails to differentiate or form appropriately.

Minor comment:

1. The authors could choose a more representative figure to show the measured relative increase in cTnT staining in sPIBF-treated hHOs (Figure 3b), as the current image shows no apparent difference.
2. The font in Figures 3e and 3f is so small that it is illegible.
3. Can the authors show full images of the placental sections shown in Figures 4b, 5c, and 5m?
4. It is unclear which placental stages have been analysed throughout Figure 4, the text provides a range of E9.5-E10.5, but the embryos appear to be E10.5. Can the authors please add this to the figure legend and labels?

Reviewer #3 (Remarks to the Author):

This study shows that PIBF is indispensable for trophoblast differentiation and syncytiotrophoblast formation. Syncytiotrophoblast and adjacent vascular cells communicate via PIBF, to promote the development of the vascular network in the primary placenta. This process plays a role in the early development of the embryonic cardiovascular system.

This is a very good and extremely meticulous work. The study is well designed the methods are appropriate, the data are interesting and sound. The paper is fairly well written. The conclusions are reasonable, except for those concerning the involvement of the immune system.

- (p. 8) the authors investigate whether the decidual NK cells contributed to the early defects of the Pibf1^{-/-} embryo, and found no significant changes either in the % of NK cells among immune cells or their absolute number in Pibf1^{+/-} decidua harbouring Pibf1^{-/-} embryos. Furthermore, in a Th17-deficient maternal immune background, Pibf1^{-/-} embryos still exhibited morphological abnormalities. They conclude that the maternal immunological environment did not play a role in the early defects of the Pibf1^{-/-} embryo. This is likely however, the absolute number or % of decidual NK cells is not meaningful without functional changes (altered lytic activity), which they did not investigate.

- (p.12) the authors addressed whether the loss of Pibf1 in the maternal immune system affects pregnancy and the offspring. They show that Pibf1 Δ/Δ ;Vav1cre/+ females (Pibf1-deficient in hematopoietic/immune cells) displayed normal reproductive functions, and their pups showed normal growth rate, indicating that Pibf1 is dispensable in the maternal immune system for maintaining pregnancy, at least under an unchallenged condition.

This conclusion is not justified.

The fact that hematopoietic/immune cells are PIBF deficient, does not mean that their function cannot

be affected by PIBF produced by the neighbouring cells.

Though uterine NK cells do not express progesterone receptors (Henderson et al., Steroid receptor expression in uterine natural killer cells *J Clin Endocrinol Metab* . 2003; 88(1):440-9.) thus they cannot produce PIBF, they still do contain PIBF in their cytoplasmic granules (Bogdan et al., PIBF positive uterine NK cells in the mouse decidua. *J Reprod Immunol*. 2017 Feb;119:38-43.), Furthermore in PIBF deficient mice, PIBF+ decidual NK cells decrease in number together with an increased decidual NK activity (Csabai et al., Altered Immune Response and Implantation Failure in Progesterone-Induced Blocking Factor-Deficient Mice. *J. Front Immunol*. 2020 Mar 11;11:349.) Knocking out the PIBF receptors on immune cells would be a more relevant approach to investigate the involvement of the maternal immune system.

In conclusion; this paper shows that the inability of the trophoblast to produce PIBF interferes with organogenesis. This in itself is an important new information, sufficient to merit publication.

Point-by-point response to reviewer comments

Reviewer #1 (Remarks to the Author):

This manuscript by Lee et al investigates the role of progesterone immunomodulatory binding factor 1 (PIBF1) in trophoblast biology and its impact on fetal development, specifically on cardiogenesis.

The authors find that PIBF1 has an essential role in facilitating trophoblast fusion into syncytiotrophoblast, and KO cells exhibit an overall decreased ability to differentiate at all. Somewhat surprisingly, this effect is seen both, with the soluble as well as with the intracellular form of PIBF1, but not when the sPIBF1 protein is supplied exogenously. Using recombinant versions of the protein, they also ascribe pro-angiogenic as well as heart differentiation-promoting properties to PIBF1 in Transwell and heart organoid culture experiments, respectively. The authors then assess KO as well as multiple conditional KO mice and find that PIBF1 ablation is embryonic-lethal around mid-gestation when placental function becomes critical. Perhaps most importantly, embryo-specific deletion of PIBF1 using the Meox-Cre driver rescues the early lethality and the placental phenotype, implying that PIBF1 is critical in the trophoblast compartment for early embryogenesis. They also generate Vav1-, Cdh5-, and Tie2-Cre conditional knockouts to assess endothelial and hematopoietic roles of PIBF1, but these conditional KOs develop normally, as expected from the Meox-Cre data.

Overall, this manuscript contains a large amount of data, albeit the informative value of a significant amount of them is relatively limited and the manuscript is inflated for data that do not add substance.

→ Thank you for reviewing our study in detail and providing helpful comments.

Regarding the disappearance of embryonic lethality, the results of the Cre-TG mice used in this study can be considered equivalent. We conducted this study and developed our hypothesis, focusing on the significance of the morphological completion of the placenta. To investigate the role of PIBF1 in placental constituent cells, which are believed to be involved in early functional placenta during placentation, and to ensure the accuracy of the *in vitro* results, we used various Cre-TG mice (Cdh5-cre and Tie2-cre for vascular and vascular-derived hematopoietic cells and Vav-cre for blood cells) in addition to Meox2-cre (which allows for PIBF1 rescue in placental trophoblasts) in the *in vivo* mouse model.

Ultimately, we demonstrated the function of PIBF1 derived from placental trophoblast cells, particularly SynT. Paradoxically, we also confirmed that PIBF1 secreted from immune cells by progesterone during pregnancy does not have a direct effect on the differentiation of placental trophoblast cells. By revising our questions, we gained valuable insights and improved the overall quality of the paper with additional results that further supported our hypotheses. Please see the point-by-point response to the specific comments.

The most important point is that the authors' key conclusion is flawed and unjustified, i.e. that placental syncytiotrophoblast-produced PIBF1 is the signaling molecule that governs

heart development, which is highlighted in the abstract and summary figure. This conclusion cannot be drawn, as PIBF1 is fairly ubiquitously expressed, and the mere fact that direct exposure of cells to PIBF1 promotes cardiac differentiation does not mean that this effect originates from a syncytiotrophoblastic source.

→ We thank the reviewer for pointing this out. As you indicated, we performed the immunohistochemistry (IHC) and in situ hybridization (ISH) assays. PIBF1 displayed a ubiquitous expression pattern throughout the embryo and placenta, both in its mRNA and protein expressions (Fig. R1). This global expression pattern indicates that PIBF1 may be developmentally essential for all types of cells, tissues, and organs. However, it also makes it difficult to identify which compartments are involved in the embryonic-lethal phenotype of *Pibf1*-null embryos.

Fig. R1. Ubiquitous expression of Pibf1 in the mouse placenta and embryo at embryonic day (E) 10.5.

a *In situ* hybridization of *Pibf1* using a *Pibf1*-specific probe. Probes detecting mouse *Ppib* and bacterial *dapb* were used as positive and negative controls. Note that the signals (brown dots) were ubiquitously present throughout the placenta and embryo. a, atrium. v, ventricle.

b Immunohistochemistry of *Pibf1* showing ubiquitous signals throughout the placenta and embryo. Sections without primary antibody were used as a negative control. dec, decidua. jz, junctional zone. lab, labyrinth. ra and la, right and left atrium. rv and lv, right and left ventricle.

→ To explore the developmentally essential role of *Pibf1*, we employed conditional knockout strategies using various Cre lines. Surprisingly, we found that *Pibf1* is dispensable for embryogenesis and placentogenesis in specific cell populations, such

as endothelial and hematopoietic lineages. Most importantly, we demonstrated the necessity of PIBF1 for SynT differentiation and formation. Further, it sheds new light on PIBF1 as a SynT-derived factor that affects the cardiovascular development of the embryo and placenta, which is supported by *in vitro* (hTSC, BeWo, and perivascular/vascular cells) and *in vivo* studies (*Pibf1*^{-/-} and cKO mice with various cre lines). Those *in vitro* and *in vivo* data are interconnected and complement each other. For instance, the *PIBF1* KO studies, which were not performed in endothelial cells such as HUVEC or HUAEC, were complemented by the *in vivo* studies using *Cdh5*- and *Tie2*-cre lines, demonstrating that *Pibf1* is not essential for the development of endothelial lineages. Such data, along with each intermediate conclusion, are interlinked and aimed at addressing the initial overarching topic of the "placenta-heart axis," which has been garnering the attention of researchers in this field.

Recently, experimental evidence has demonstrated the interaction between the placenta (including its compartments and derived factors) and the development of specific embryonic organs¹⁻³.

In this regard, the unique properties of PIBF1 in trophoblast include:

1. Its indispensable role in SynT differentiation and formation
2. The existence of its secretory PIBF1
3. Its secretion increased after differentiation, which led us to explore the physiological implication of SynT in the axis.

Moreover, how PIBF1 would enter the fetal blood circulation, how it traverses the placental endothelial cell layer, or whether indeed syncytiotrophoblast-derived PIBF1 ever crosses the interhaemal barrier, remains unclear.

→ We appreciate the reviewer's criticism. As we have mentioned above, the primary focus of this study was to highlight the production and secretion of PIBF1 by the trophoblast syncytialization event, and how the secreted PIBF1 impacts the cardiovascular development of both the placenta and embryo. Such functional changes within the placenta are generally accompanied by active anatomical and structural alterations of placental compartments. So, we organized and identified the significance of developmentally important events during pregnancy. Considering the period when *Pibf1*-null phenotypes manifested, we selected a critical event, as you indicated, "interhaemal membrane formation," which should precede the subsequent development of the placenta and embryo. This logic is based on the series of events that occur during the morphological development of the initial placenta. Rather than focusing on how PIBF1 traverses the barrier, we concentrated on the fundamental physiological significance of PIBF1: where PIBF1 is developmentally required, which cells produce PIBF1, what kind of developmental role PIBF1 has, and ultimately how it functions in the placenta-heart axis.

The interhaemal barrier is composed of various cell populations, including various trophoblast subtypes and vascular endothelial cells, that enable the formation of a feto-maternal vascular network. In our study, we specifically focused on SynT as a

leading performer, which is in line with the recent literature³. Our data show that proper differentiation into SynT is the crucial process that initiates the structural development of the interhaemal membrane. SynT differentiation and formation enable the production of SynT-derived factors, such as PIBF1, which recruit perivascular/vascular cells.

Such a sequential series of events enables the proper alignment of each layer, resulting in the anatomical formation of the interhaemal membrane. Surprisingly, PIBF1 is involved in both SynT differentiation and the recruitment of perivascular/vascular cells. Due to its multifaceted role, the absence of PIBF1 severely impaired the initial development of the feto-maternal network, and subsequent events (cardiovascular development encouraged by SynT-derived PIBF1) were also defective as expected. Additionally, embryonic development may be normal if PIBF1, derived from maternal immune cells, crosses the interhaemal barrier and directly regulates fetal cardiac development. However, considering the embryonic lethality of PIBF1 KO, this may not be the case. To avoid exaggerating the conclusion about the placental cardiac axis involving SynT-derived PIBF1, we either normalized (Supplementary Fig. 9a, b) or abrogated (Supplementary Fig. 16) the secretion of PIBF1 from maternal immune cells. We did not observe a significant effect on the development of defective embryos.

Rescuing the placental defect restores all of embryonic development, and hence the effect on cardiogenesis may be indirect and due restored nutritional supply and/or due to normal tissue function of other embryonic cell types. The entire focus around this key conclusion needs to be completely revised and omitted.

- As the reviewer indicated, it can be seen that normal cardiovascular development seen in *Pibf1*-null embryos under *Pibf1*-rescued placenta is due to restored placental function, such as nutritional supply or any unknown support provided by normally developed embryonic tissues/organs. This raises confusion as to whether it appears to be an indirect effect.

However, since the heart and vascular system are the first organs to develop during embryogenesis^{4,5}, if the cardiovascular defect was rescued or alleviated, it is possible that abnormalities in other organs connected to the cardiovascular system could be indirectly restored or ameliorated, which was also observed in cKO^{*Meox2*} embryos.

This study provides evidence that the structural integrity of the placenta, including the fetal blood vessels derived from the splanchnic mesoderm, may influence the overall development of the fetal cardiovascular system, which plays a critical role in early embryonic development. Our *in vitro* and *in vivo* data, which we present here, are interconnected and mutually complement each other. One of our hypotheses was how SynT affects embryogenesis. Through *in vitro* studies, we found that (1) endogenously expressed PIBF1 directly induces trophoblast syncytialization, (2) SynT recruits vascular cells and promotes their angiogenesis through the secretion of PIBF1, and (3) sPIBF1 further enhances hHO development, suggesting that SynT-derived PIBF1 could play a role in embryonic cardiovascular development. *In vivo* studies using

various cre lines showed defects in SynT and cardiovascular development in *Pibf1*-null mice, which was rescued in cKO^{Meox2} mice, confirming the unique properties of PIBF1.

In addition to the *in vitro* study using human heart organoids, considering the characteristics of hHO, which is a mixed population of several cell types such as endothelial cells and cardiomyocytes⁶, we could not demonstrate in the original manuscript whether the effect of PIBF1 in hHO is due to its direct effect on cardiomyocyte or an indirect effect resulting from enhanced endothelial angiogenesis. Thus, we conducted a 2D culture using human iPSC-derived cardiomyocyte⁷, and found that rsPIBF1 could enhance the differentiation of cardiomyocytes (Fig. 3c), confirming the direct impact of PIBF1 on cardiomyocytes. These new data have been included in the revised manuscript.

Secondly, the KO phenotype is extremely severe with the entire embryo exhibiting profound developmental delay, lack of turning, and many other defects. Pericardial edema is a common feature of embryos with such severe phenotypes. Thus, the entire focus around the placenta-heart axis is not necessarily justified either, as development of all organ anlagen is impeded.

With these major flaws in mind, this reduces the data to the conclusion that PIBF1 is another example of the collection of genes that are essential in the placenta for embryonic development to term. This is important, but not in the strong and unidirectional interpretation offered by the authors.

- ➔ As the reviewer indicated, the severe phenotypes of *in-utero* lethal mutants have made it difficult to assess specific phenotyping, especially in cases where there are limited or no previous preliminary studies or literature available for certain genes. Fortunately, PIBF1 is not the case due to the existence of reports by various dedicated researchers and our *in vitro* data.

As the reviewer indicated, the gross phenotypes observed in the *Pibf1* global KO embryo (i.e., developmental delay, pericardial edema) are frequently observed in embryos with severe abnormalities. Therefore, it cannot be concluded that these embryos only have cardiovascular defects. To support the existence of cardiovascular defects, we thoroughly evaluated cardiovascular phenotypes using multiple parameters, including histology (H&E, whole mount IF, and IF-FFPE), morphometry of whole hearts, and expression of cardiogenesis/endothelial markers in hearts.

Using mouse genetics tools should be performed to clarify whether the cardiovascular defects are of placental origin. However, most of the gene KO embryos were not rescued by the Meox2-Cre mediated trophoblast gene restoration in the presence of placenta malformations, resulting in embryonic lethality. For example, *ankyrin repeat and SOCS box containing 2 (Asb2)* knockout embryos display cardiovascular abnormalities (pericardial edema) with severe gross phenotypes^{8,9}, which are, however, of cardiac origin¹⁰ and are not linked to the genetic state of placental *Asb2*, as demonstrated in the *Meox2* cre-cKO strategy (**Fig. R2**, unpublished data). In contrast to *Asb2*, *Pibf1*-null embryos with *Pibf1*-rescued placenta (cKO^{Meox2}) have

displayed normal gross and cardiovascular phenotypes, as demonstrated by various *in vivo* parameters. These findings, together with our *in vitro* data using perivascular/vascular cells, their co-culture with SynT, and hHO, indicate the essential role of trophoblastic PIBF1 (endogenously expressed and secreted) in the placenta-heart axis.

Fig. R2. Phenotypes of *Asb2* KO embryos at E10.5.

Genotype distribution in $Asb2^{+/-} \times Asb2^{+/-}$ or $Asb2^{f/+}; Meox2^{cre/+} \times Asb2^{ff}$ (a, b), gross phenotypes of yolk sac vasculature and embryo (c, f), crown-rump length (d, g), and proportion of embryos with or without heart beating (e, h) in $Asb2^{-/-}$ and *Meox2* cre-mediated *Asb2* cKO embryos. Blue arrows indicates yolk sac vasculature filled with blood. Red arrowheads indicate pericardial effusion. n.s., ** $P < 0.01$ in χ^2 test (a, b, e, h). *** $P < 0.001$ in Student's t-test (d, h).

Additionally, we utilized a precise and sensitive mass spectrometry technique called multiple reaction monitoring (MRM), also known as selective reaction monitoring (SRM), which allows for the selective quantification of compounds in complex mixtures. We demonstrated the significant increment of secreted PIBF1 in functionally differentiated trophoblasts. Moreover, as described earlier in this response letter, in addition to the hHO data, we demonstrated the direct effect of PIBF1 on cardiomyocyte differentiation (Fig. 3c), further highlighting the essential role of PIBF1 in this axis.

Instead of focussing on the somewhat contrived focus on heart development, it would be far more interesting and legitimately justified by the data to investigate how PIBF1 facilitates syncytiotrophoblast fusion. If it was acting as a soluble protein, then the exogenous supply of

sPIBF1 should have an effect, but this is not the case. Hence it must act intra-cellular to exert this function. What is the nuclear role of PIBF1, how would it even enter the nucleus (NLS?), or how would a centrosome-associated localization lead to cell fusion? These functional aspects are critical to elucidate in order to heighten the insights gained from this study beyond the observational level of fusion defects.

→ We appreciate the reviewer's suggestion. You have pointed out exactly what we had been considering for a long time. The physiological function of PIBF1 in developmental biology can be understood based on previous literature: while full-length PIBF1 is a centrosome-associated protein, its secretory isoform has an immunomodulatory function in immune cells during pregnancy¹¹. As you indicated, investigating the reversibility of abnormalities in *Pibf1*^{-/-} embryos through exogenous PIBF1 treatment was not feasible in our present experimental design, because the supplementation of exogenous PIBF1 did not rescue fusion defects in *PIBF1* KO BeWo (Fig. 1p), which was also observed *in vivo* (data not shown). Moreover, as we demonstrated using *Vav1* cre-cKO that *Pibf1* is dispensable in maternal immune cells, at least in an unchallenged normal pregnancy condition, we excluded the possibility of PIBF1 derived outside the placenta (exogenously derived and immune cell-derived ones) and were able to focus on PIBF1 expressed or derived from SynT.

Several trophoblast fusogen proteins, which are syncytin proteins of retroviral origin, are exclusively expressed in placental trophoblasts under normal physiological conditions¹²⁻¹⁴. As per your suggestion, the identification of new fusogens has greatly advanced our understanding of the mechanisms of cell-cell fusion. In this study, we identified that *PIBF1* acts as a fusion gene. However, as mentioned earlier, the molecular mechanisms required for PIBF1 to function as a fusion protein have yet to be thoroughly investigated, as we focused on understanding the physiological significance of SynT and the role of PIBF1 in mediating it. Therefore, further research is needed to investigate the impact of PIBF1 on cell-to-cell fusion and its relationship with various diseases, including infertility, myopathies, osteoporosis, and cancer, by regulating the functions of related cells.

Although we could not present the in-depth molecular mechanism of PIBF1 in trophoblast fusion, we tried to pay attention to glial cell missing 1 (GCM1) as an intrinsic factor for PIBF1-mediated trophoblast syncytialization. GCM1 expression was significantly down-regulated in *PIBF1* KO hTSCs and its derived SynT (Fig. 1f and Supplementary Fig. 3f). GCM1 is expressed exclusively in the placenta in humans and mice¹⁵⁻¹⁷, is traditionally regarded as a master regulator of SynT formation, and has recently been shown to also play a role in EVT differentiation¹⁸. For instance, knockdown of GCM1 in human TSCs led to a decreased capacity of differentiation into both SynT and EVT¹⁸, which closely resembles the phenotype observed in *PIBF1* KO TSCs. Thus, the relationship between PIBF1 and GCM1 needs to be further investigated. We have added these discussions in the Discussion section of the revised manuscript.

Centrosome-associated full-length PIBF1 is involved in primary ciliogenesis¹⁹. Although cilia are not observed in trophoblast cells of the mouse placenta²⁰, their development is essential in various pregnancy-related complications in the trophoblast of the human placenta^{21,22}. In this regard, it is necessary to analyze the role of

centrosome proteins to confirm the function of cilia in early placental development. In this regard, we identified defects in placental SynT and observed embryonic lethal phenotypes, such as the deletion of *PIBF1*, in embryos where *intraflagellar transport 88* (*Ift88*), one of the well-known ciliogenic genes, was deleted. However, the phenotypes were not of placental origin, as they were not rescued in the *Meox2-cre* cKO strategy; this indicates that the SynT defects seen in the *Ift88*^{-/-} placenta were a result of defective ciliogenesis in the embryo proper (data not shown). We are currently studying the meaning of ciliogenesis and the molecular mechanisms in placental cells other than trophoblasts.

As seen in the cases of *Pibf1* and *Ift88*, which are centrosome-associated genes that also have crucial developmental functions in different parts of the placenta, it is evident that other cells surrounding trophoblasts also contribute to proper placentogenesis. Furthermore, defects in one compartment can have an impact on the surrounding areas. In this regard, one of the unique features of PIBF1 is its ability to mediate the communication between SynT and perivascular/vascular cells, promoting their recruitment and encouraging angiogenesis.

The Vav1-, Cdh5-, or Tie2-Cre mediated conditional knockouts are relatively meaningless as their lack of phenotype can be expected from the Meox-Cre results. As such, they inflate the manuscript but do not add substantial value. It is possible that the chronological sequence of events was such that the authors started by first generating those cKOs, before using the Meox-Cre. Any value beyond trophoblast-mediated rescue of embryonic lineages should be highlighted, but does not deserve the amount of space occupied in the figures and text.

→ Many thanks for the great suggestion. We have received some of your suggestions, and through this point-by-point response, we have once again stated the reason and importance of using Vav1-, Cdh5-, or Tie2-Cre in this study.

As the reviewer pointed out, the data obtained from using *Pibf1* cKO with endothelial/hematopoietic lineage-specific cre lineages (*Cdh5*-, *Tie2*-, or *Vav1-cre*) appear to be relatively insignificant, as they did not show significant changes from an embryonic standpoint. However, as the studies progressed and the data accumulated, our hypothesis became clearer as our *in vitro* and *in vivo* data were all interconnected, leading us to answer a significant question: "What is responsible for mediating the placenta-heart axis?"

The chronological order of studies using mutations or cKOs is as follows: *Pibf1*^{-/-} was the first, and since we suspected that PIBF1 had centrosomal protein and immunological functions based on its known functions as the cause of embryonic lethality, we used *Pibf1*^{-/-}; *Trp53*^{-/-} and *Il17*^{-/-} mice and WT mice to evaluate *Pibf1* haploinsufficiency. After confirming that there was no significant relationship between the previously known function of PIBF1 and embryonic lethality, we conducted experiments to investigate the role of PIBF1 in the development of placental cells.

Although the function of PIBF1 in SynT differentiation was directly evaluated through *PIBF1* KO induction in various trophoblasts *in vitro*, its function could not be directly confirmed in each placental cell, including fetal endothelium (FE), pericytes, and FE-

derived hematopoietic cells. The endothelial cell lines (HUVEC, HUAEC) have unique characteristics, which makes it difficult to achieve complete KO cell lines.

Although cKO^{Meox2} displayed an embryonic phenotype that was rescued by trophoblast-specific gene restoration, with Pibf1 expression distributed throughout the embryo/placenta, we needed to cross-validate whether PIBF1 is truly necessary for the development of any cells in the placenta *in vivo*. Therefore, we directly identified the function of Pibf1 in the development of individual vascular endothelial cells and its derived blood cells (cKO^{Cdh5}, cKO^{Tie2}) required to form the placenta during embryonic development. Accordingly, we decided to present the results in the main figures, focusing on identifying the source cell of Pibf1 in the placenta that connects the axis with the developing heart and arranged the results of the cre-TG mouse studies as described in the original manuscript in chronological order, following the experimental flow based on the aforementioned logic.

To aid readers' understanding, we have modified the order of Cre-TG mouse results as per your suggestion: studies involving cKO^{Meox2} mutants at a later stage (E15.5~) are now mentioned earlier in the Results section of the revised manuscript (Supplementary Fig. 14d-h).

Given the cilia-related mortality of embryo-specific KOs in late gestation (that is mentioned late and should be stated earlier), the centriole-related function of PIBF1 may be far more important for its normal role in development.

- Concerning the ciliogenic function of PIBF1, we previously discussed in this response letter a comparison between Pibf1 and Ift88 in terms of *Meox2* cre-mediated cKO data, implying the significance of primary ciliogenesis in placenta development. There is an interspecies difference in the location of primary cilia formed within the placenta: human trophoblasts have abundant primary cilia²¹, while extraembryonic lineages in mice lack primary cilium²⁰. Therefore, we focused on the physiological significance of trophoblast syncytialization, a fundamental event for SynT differentiation and formation. In order to avoid confusion, the centrosome-associated PIBF1 features-related descriptions (*Pibf1*^{-/-}; *Trp53*^{-/-} and later-stage (E15.5~) cKO^{Meox2} data) are now described earlier in the Results section of the revised manuscript (Supplementary Fig. 14).

Additional points include:

- Where precisely in the placenta and embryo is PIBF1 expressed? These data are essential to infer cell types where it might be functioning.

- As the reviewer suggested, we additionally carried out IHC and ISH assays. We found that PIBF1 displayed a ubiquitous expression pattern throughout the embryo and placenta, both in its mRNA and protein expressions (**Fig. R1**), as stated earlier in this Response Letter. The global expression of PIBF1 mRNA and protein might result from the characteristics of the antibody and probe: the antibody was raised against the N-terminal 48 kDa portion of PIBF1, while the probe targets most of the isoforms of mouse Pibf1²³. These IHC and IHC data are only provided in this response letter.

- It is impossible to interpret the BeWo cell fusion photos. What is highlighted by the dotted lines, and how would one see that?

→ Given that fused BeWo or syncytiotrophoblasts contain multiple nuclei in a single cell, for quantifying trophoblast fusion, we adopted a live cell imaging method to quantify trophoblast fusion. This method utilizes Di-8-ANEPPS (Di-8), which can effectively label the plasma membrane of BeWo as previously described^{14,24}. Cells containing a single nucleus or multiple nuclei were counted, and the fusion index was quantified as described in the Methods section. In the images of fused BeWo cells (Fig. 1a, o, Supplementary Fig. 2b, 4b, c), white and red dotted lines indicate the Di-8-labelled plasma membrane and the Hoechst-labelled nuclei within the fused cells. As the dotted lines were essential for explaining the images but could interrupt their interpretation, we reduced the thickness of the lines.

We apologize for any inconvenience caused by the misinterpretation. Also, it seemed that the overall resolution of figures in MS Word manuscripts could be affected by PDF conversion: we found that the image quality was significantly reduced after PDF conversion, particularly in the BeWo fusion images, which require precise delineation of nuclei or cell boundaries. We therefore replaced the fusion images with higher resolution ones. Problems caused by PDF conversion can be addressed through future consultations with the publishing team. The revised manuscript now includes figures or figure components with images related to BeWo fusion.

- How does the “heart organoid” model differ from traditional EB differentiation experiments towards cardiomyocytes? Is this terminology merely a ‘sexing up’ of the experimental strategy? Using “EB differentiation” as term is fine.

→ The conventional protocols for differentiating EB or iPSC-derived two-dimensional cardiomyocytes require a metabolic purification using non-glucose media supplemented with sodium lactate to achieve high purity^{25,26}.

However, the hHO utilized in this study is a platform that can reflect three-dimensional morphological and functional characteristics of the heart *in vitro*. In particular, the EBs (>2 mm in diameter) with three-dimensional structural complexity were produced using an extracellular matrix constituent Matrigel (10 %) as the generated heart organoids have recently been shown to recapitulate morphological and functional aspects of the developing heart⁶. Their differentiation was done without the CM purification procedure to preserve the interactions among heart-resident cells, and the resulting hHOs were composed of various types of cardiac cells, including cardiomyocytes, endothelial cells, pericytes, cardiac fibroblasts, and epicardial cells⁶.

Therefore, we applied this model to our study, demonstrating the heart development-promoting effect of sPIBF *in vitro*.

- The methods around the generation of CRISPR KOs in hTSCs must be expanded, as this is by no means a trivial approach with hTSCs being somewhat refractory to single cell cloning. The authors should expand on how these cells were generated, how many KO

clones they obtained, how many were tested and how many exhibited the same phenotype. An N=3 should be used at a minimum.

→ Thank you for the detailed suggestion. As you mentioned, generating *PIBF1* KO hTSCs can be somewhat challenging because they need to maintain hTS characteristics during single-cell cloning and subsequent experiments. However, these characteristics were successfully obtained using CRISPR/Cpf1 and were utilized in further experimental procedures in this study. The representative data on the generated hTSC (exon 2 of h*PIBF1* targeted; labeled as KO line #2-21) were presented in the original manuscript (Fig. 1, Fig. 2, and Supplementary Fig. 3). Following your suggestion, we expanded the descriptions on how *PIBF1* KO hTS cells were generated and tested whether the other *PIBF1* KO hTS lines exhibited the same phenotype as in lines #2-21.

First, regarding the generation of *PIBF1* KO hTS, the CRISPR/Cpf1-mediated KO procedure was used to target human *PIBF1* exon2 or exon 4; thereafter, puromycin selection was performed and single-cell colonies (n=60 colonies of each targeted exon) were selected after seeding them in a 96-well plate. *PIBF1* deficiency was screened by measuring the expression levels of full-length *PIBF1* (90 kDa) using Western blot analysis (**Fig. R3a**). The KO candidates targeted with exon 2 (#2-24, #2-38) or exon 4 (#4-26) were further analyzed using Sanger sequencing (n=20-25 clones per colony), revealing two distinct mutant alleles carrying frameshift-causing indels inducing a different premature stop codon in each colony (**Fig. R3b**), thereby confirming the successful establishment of *PIBF1* KO TS lines. Again, the data regarding the TS KO line #2-21 were described in the original manuscript. This workflow was also applied in the BeWo cell line.

Second, we further analyzed hTS and differentiation capacities in the KO line #2-24 to address data replication among different lines. As a result, increased expressions of TS markers were detected in a stem cell state (**Fig. R3c**), and its EVT differentiation capacity was impaired (**Fig. R3d**). Most importantly, TS-derived SynT both in 2D and 3D cultures displayed a declined expression of SynT markers (**Fig. R3e**) along with increased expression of TS markers (**Fig. R3f**), demonstrating that *PIBF1* KO hTSC line #2-24 replicated the phenotypes as seen in the former #2-21 and those phenotypes were in consequence of *PIBF1* KO. The *PIBF1* KO hTSC line #2-24 was presented only in this Response Letter, and the methods regarding the generation of *PIBF1* KO hTSC were expanded in the revised manuscript.

All of the experimental procedures above were carried out before reaching passage 9: the assays regarding TS and differentiation phenotyping were performed between passages 6 and 8.

Fig. R3. CRISPR/Cpf1-generated *PIBF1* KO hTSCs are defective in SynT differentiation.

a *PIBF1* expression in *PIBF1* KO candidate hTSC colonies. Red arrows indicate a band size of full-length *PIBF1* (90 kDa). Suspected colonies are colored in red. Colonies marked with a red dotted box were further analyzed using Sanger sequencing. Note that the data regarding TS KO lines #2-21 were described in Fig. 1, Fig. 2, and Supplementary Figure. 3 of the main manuscript. **b** Mutant *PIBF1* sequence observed in the *PIBF1* KO hTSC lines using CRISPR/Cpf1 targeting human *PIBF1* exon 2 or 4. **c** Expressions of TS markers in *PIBF1* KO TSC #2-24 in a stem cell condition. **d** Expressions of *HLA-G* and *MMP2* in EVT derived from *PIBF1* KO TSC #2-24. **e, f** Expressions of SynT (**e**) and TS (**f**) markers in SynT derived from *PIBF1* KO TSC #2-24. Transcript expression data were normalized to *GAPDH* and expressed relative to WT (**c-f**). Data are presented as mean \pm SEM (**c-f**); * $P < 0.05$; ** $P < 0.01$; *** $P < 0.001$; **** $P < 0.0001$ in Student's t-test (**c-f**).

- How were the recombinant proteins isolated and purified, and at what concentration were they used? Are the pro-angiogenic and heart differentiation-promoting effects dose-dependent?

➔ We have added the detailed preparation of recombinant *PIBF1* in the Materials & Methods section of the revised manuscript.

Concerning studies evaluating the angiogenic properties of *PIBF1* in HUAEC, HUVEC, and placental pericytes, the concentration of *PIBF1* was indicated in the figures (Fig. 2e-g, i-k and Supplementary Fig. 5d-l) and the Method section: rs*PIBF1*, rf*PIBF1*, ra*PIBF1*,

and rcPIBF at a dose of 20-200 ng/ml for cell migration assay, and 50-400 ng/ml for tube forming assay; the results showed a dose-dependent angiogenic effect except for rcPIBF; the C-terminal PIBF1 fragment did not show significant changes in endothelial migration, tube formation, or pericyte migration (Supplementary fig. 5f-l).

Meanwhile, in the studies using hHO, we demonstrated the heart-differentiation-promoting effect of rsPIBF at 200 ng/ml, a dose validated in the cell migration and tube-forming assays; however, studies regarding its dose-related effect were missing in the original manuscript. Therefore, as you suggested, we added an experiment regarding the dose-dependency of rsPIBF at doses of 50, 200, and 1000 ng/ml in hHO. As a result, the dose range (50-1000 ng/ml) of rsPIBF significantly increased the distributions of endothelial cells (CD31-positive) and cardiomyocytes (cTnT-positive) within hHO (Supplementary Fig. 6a); notably, the sprout length of hHO increased in a dose-dependent manner with rsPIBF (Supplementary Fig. 6b). These results strengthen our argument for the placenta-heart axis during embryogenesis following increased secretion of PIBF1 after SynT formation using the MRM method.

The results and related methods have been included in the Results and Methods sections of the revised manuscript, respectively.

- The display for qPCR (and other) datasets differs almost between every figure element, and the replicate data points of the controls cannot be made out in many of these graphs. Please homogenize the display to at least a common display type for each type of data.

→ According to the reviewer's suggestion, we have corrected the figures in accordance with the journal's policy. We synchronized the display style of qPCR datasets into a blue or skyblue box-and-whisker plot with dots. Especially in the case of endothelial vasculature markers in laser micro-dissected placenta sections (Fig. 4l, 5i), the assay was re-performed with an increased sample size ($n=3 \rightarrow n=4$ per group) and presented as a boxplot after statistical analysis. The results with $n=4$ /group showed almost the same pattern as in the previous one ($n=3$ /group): reduced placental vasculature in *Pibf1* KO was labyrinth-specific, which was rescued in *Pibf1^{Meox2}* cKO conceptuses.

- Given the fact that RNA-seq was performed on heart organoids, it is barely surprising that heart-related terms stand out in the GO terms. This should be reflected in the description of these data.

→ Thank you very much for pointing this out. We should have provided a sufficient explanation. To investigate the effect of trophoblast-derived secreted PIBF1 on the development of human heart organoids, we conducted a comparative experiment involving a control group in order to identify and describe the gene expression changes induced by secreted PIBF. We have rewritten it in the Results section of the revised manuscripts to convey as much meaning as possible. Please review it, and if we have misunderstood the reviewer's opinion, please let us know. We will be glad to conduct any additional experiment that you deem necessary.

Thank you again for your valuable suggestions on our manuscript, which were vital in significantly improving our study as a whole. We hope that our responses, newly conducted experiments, and the corresponding revisions are satisfactory. We would be happy to make any further revisions that you may deem necessary.

References used in this response letter

- 1 Perez-Garcia, V. *et al.* Placentation defects are highly prevalent in embryonic lethal mouse mutants. *Nature* **555**, 463-468 (2018).
- 2 Hemberger, M., Hanna, C. W. & Dean, W. Mechanisms of early placental development in mouse and humans. *Nat Rev Genet* (2019).
- 3 Radford, B. N. *et al.* Defects in placental syncytiotrophoblast cells are a common cause of developmental heart disease. *Nat Commun* **14**, 1174 (2023).
- 4 Ji, R. P. *et al.* Onset of cardiac function during early mouse embryogenesis coincides with entry of primitive erythroblasts into the embryo proper. *Circ Res* **92**, 133-135 (2003).
- 5 Savolainen, S. M., Foley, J. F. & Elmore, S. A. Histology atlas of the developing mouse heart with emphasis on E11.5 to E18.5. *Toxicol Pathol* **37**, 395-414 (2009).
- 6 Lee, S. G. *et al.* Generation of human iPSCs derived heart organoids structurally and functionally similar to heart. *Biomaterials* **290**, 121860 (2022).
- 7 Lee, S. G. *et al.* Development and validation of dual-cardiotoxicity evaluation method based on analysis of field potential and contractile force of human iPSC-derived cardiomyocytes / multielectrode assay platform. *Biochem Biophys Res Commun* **555**, 67-73 (2021).
- 8 Metais, A. *et al.* Asb2alpha-Filamin A Axis Is Essential for Actin Cytoskeleton Remodeling During Heart Development. *Circ Res* **122**, e34-e48 (2018).
- 9 Park, S. G. *et al.* Heart defects and embryonic lethality in Asb2 knock out mice correlate with placental defects. *Cells Dev* **165**, 203663 (2021).
- 10 Yamak, A. *et al.* Loss of Asb2 Impairs Cardiomyocyte Differentiation and Leads to Congenital Double Outlet Right Ventricle. *iScience* **23**, 100959 (2020).
- 11 Szekeres-Bartho, J., Wilczynski, J. R., Basta, P. & Kalinka, J. Role of progesterone and progestin therapy in threatened abortion and preterm labour. *Front Biosci* **13**, 1981-1990 (2008).
- 12 Mi, S. *et al.* Syncytin is a captive retroviral envelope protein involved in human placental morphogenesis. *Nature* **403**, 785-789 (2000).
- 13 Dupressoir, A. *et al.* Syncytin-A knockout mice demonstrate the critical role in placentation of a fusogenic, endogenous retrovirus-derived, envelope gene. *Proc Natl Acad Sci U S A* **106**, 12127-12132 (2009).
- 14 Zhang, Y. *et al.* TMEM16F phospholipid scramblase mediates trophoblast fusion and placental development. *Sci Adv* **6**, eaba0310 (2020).
- 15 Woods, L., Perez-Garcia, V. & Hemberger, M. Regulation of Placental Development and Its Impact on Fetal Growth-New Insights From Mouse Models. *Front Endocrinol (Lausanne)* **9**, 570 (2018).
- 16 Baczyk, D. *et al.* Complex patterns of GCM1 mRNA and protein in villous and extravillous trophoblast cells of the human placenta. *Placenta* **25**, 553-559 (2004).
- 17 Nait-Oumesmar, B., Copperman, A. B. & Lazzarini, R. A. Placental expression and chromosomal localization of the human Gcm 1 gene. *J Histochem Cytochem* **48**, 915-922 (2000).
- 18 Jeyarajah, M. J. *et al.* The multifaceted role of GCM1 during trophoblast differentiation in the human placenta. *Proc Natl Acad Sci U S A* **119**, e2203071119 (2022).
- 19 Kim, K., Lee, K. & Rhee, K. CEP90 is required for the assembly and centrosomal accumulation of centriolar satellites, which is essential for primary cilia formation. *PLoS One* **7**, e48196 (2012).
- 20 Bangs, F. K., Schrode, N., Hadjantonakis, A. K. & Anderson, K. V. Lineage specificity of primary cilia in the mouse embryo. *Nat Cell Biol* **17**, 113-122 (2015).
- 21 Wang, C. Y., Tsai, H. L., Syu, J. S., Chen, T. Y. & Su, M. T. Primary Cilium-Regulated EG-VEGF Signaling Facilitates Trophoblast Invasion. *J Cell Physiol* **232**, 1467-1477 (2017).
- 22 Ritter, A. *et al.* Primary Cilia in Trophoblastic Cells: Potential Involvement in Preeclampsia. *Hypertension* **76**, 1491-1505 (2020).
- 23 Lachmann, M. *et al.* PIBF (progesterone induced blocking factor) is overexpressed in highly proliferating cells and associated with the centrosome. *Int J Cancer* **112**, 51-60 (2004).

- 24 Zhang, Y. & Yang, H. A simple and robust fluorescent labeling method to quantify trophoblast fusion. *Placenta* **77**, 16-18 (2019).
- 25 Afjeh-Dana, E. *et al.* Stem Cell Differentiation into Cardiomyocytes: Current Methods and Emerging Approaches. *Stem Cell Rev Rep* **18**, 2566-2592 (2022).
- 26 Burridge, P. W. *et al.* Chemically defined generation of human cardiomyocytes. *Nat Methods* **11**, 855-860 (2014).

Point-by-point response to reviewer comments

Reviewer #2 (Remarks to the Author):

In this study, the authors explore the role for trophoblast-derived PIBF-1 in differentiation of placental syncytiotrophoblast and the link to vascularization and cardiogenesis in development. Using cell culture approaches, it was demonstrated in a series of elegant experiments that PIBF-1 is essential for syncytialisation of human trophoblast cells. Co-culture and sPIBF-treatment approaches using human foetal endothelial cell lines and iPSC-derived heart organoids, the authors show the impact of PIBF on angiogenesis and cardiogenesis in vitro. The assessment of a *Pibf* KO mouse showed a remarkable co-occurrence of a syncytiotrophoblast layer II defect in the placenta and abnormal heart formation. A conditional embryo-specific KO demonstrated that a wildtype placenta not only recovered the placental defect, but rescued the heart defect as well. Together, this work provides a compelling mechanism linking the placenta-heart axis in development, a significant contribution to the field.

Major comments:

1. The data presented in section 2 of the results show the effect of syncytialised BeWo cells on the angiogenic properties on foetal endothelial cell lines (lines 137-175).

a) Given that synT cells are adjacent to maternal endometrial vasculature, synT-produced PIBF-1 may also be involved in remodelling maternal vascular endothelium. Is there an appropriate cellular model to assess this aspect or can the authors comment on this?

→ Thank you for reviewing our study in detail and providing helpful comments.

As the reviewer indicated, some previous studies described the essential role of PIBF1 in endometrial remodeling. PIBF1 is highly expressed in endometrial stromal cells during the implantation window in both humans and mice, and plays a vital role in the decidualization process, as demonstrated in studies using *PIBF1* KD human endometrial stromal cells¹ and rPIBF-treated mouse endometrial stromal cells². Moreover, in a previous study, we utilized progesterone receptor (*Pgr*)-cre mice, which express Cre activities in the female reproductive tracts, including the uterus and showed that *Pgr*^{cre}-mediated *Pibf1* cKO mice exhibited female infertility without defects in ovarian functions (e.g., oogenesis and fertilization). Interestingly, the number of implantation sites was not significantly different between control and cKO^{*Pgr*}, and embryos in cKO^{*Pgr*} females died *in utero* at mid-gestational ages (data not shown), suggesting the essential role of PIBF1 in the uterus during pregnancy. In addition, decidual immune cells derived from the maternal immune system, especially B cells, are a significant source of PIBF1, which protects against preterm labor caused by uterine stress such as inflammation and infection³.

Separately from the above subject (roles of PIBF1 in the uterine environment), another topic of interest is whether trophoblast-derived PIBF1 is involved in maternal vascular remodeling within the endometrium. This subject can be divided into two categories: 1) remodeling of the spiral arteries and 2) differentiation of SynT-I. There was some indirect evidence in this study and previous literature that provided a glimpse into each category.

- First, in terms of spiral artery remodeling, TGC markers, including *Plf* and *Pl2* were not changed in both *Pibf1* KO and cKO^{*Meox2*} mouse placentas (Fig. 4g and 5g), suggesting that canal-associated TGC (C-TGC) or spiral artery-associated TGC (SpA-TGC) in the decidua or junctional zone were not affected by the *Pibf1* status of trophoblasts. On the other hand, *PIBF1* KO human TS cells displayed discernible phenotype in terms of EVT differentiation (Supplementary Fig. 3), which is responsible for invading the uterine wall and remodeling spiral arteries.
- Second, in terms of SynT-I, which directly encounters maternal vasculature in mice and is highly analogous to the SynT layer in humans, we demonstrated the essential roles of PIBF1 in syncytialization and sPIBF1 in vascular/perivascular recruitment in human cells (Fig. 1 and 2), and lack of syncytialization was also detected in *Pibf1*-null mouse placentas (Fig. 4). Along with the maldevelopment of SynT-II, defects in SynT-I were detected at an ultrastructural level (Fig. 4j). Meanwhile, Radford *et al.* recently identified the defective SynT-I as the primary placental abnormality associated with developmental heart defects⁴.

Many technical breakthroughs have greatly facilitated studies in placentology, including the hTSCs-derived first-trimester placenta⁵ used in this study. As implantation and placentation initiates with EVT invasion and spiral artery remodeling, *in vitro* platforms for EVT-endothelial interaction have been continuously advanced⁶: for example, a recently developed implantation-on-a-chip device compartmentalizing EVT, ECM, and endometrial vasculature can mimic EVT invasion and its interaction with vascular endothelium in the maternal-fetal interface⁷.

Likewise, the concept of "placenta-on-a-chip" models has also been applied in SynT⁸⁻¹¹, and several trials have reported the development of *in vitro* systems that combine multiple placental cell types to explore the transport or barrier functions of the placenta. For example, a co-culture system using fibroblast, endothelial cells, and trophoblasts has been proven to mimic SynT barrier development^{12,13}.

Therefore, it may be possible to address the impact of SynT-derived factors on the endometrial vasculature by co-culturing SynT cells with maternal endometrial endothelial cells. However, in this study, we did not examine the "trophoblast to maternal" interactions, but rather "trophoblast to maternal" interactions. Additionally, studies on maternal spiral artery remodeling have primarily focused on EVT cells, given the anatomical structure of the human placenta. Although our *in vitro* results using hTSCs showed poor EVT differentiation depending on PIBF1 KO, *in vivo* studies using the KO mouse model showed no differences in maternal blood sinuses within the labyrinth, even with PIBF1 deletion. Those partly conflicting results from mouse and human studies might arise from the comparative developmental anatomy of placentas in the two species, which share both anatomical and physiological similarities and differences^{14,15}. This issue may draw the attention of researchers in the field of placentology and needs further investigation.

Due to the unresolved problems or hurdles and the data that were not fully demonstrated, the discussions mentioned above were only cited in this response letter and not in the revised manuscript.

b) Can the authors also justify why syncytialised BeWo cells were used instead of syncytialised hTSCs, the latter of which may be more physiologically similar to primary cells?

→ As you indicated, our study did not investigate whether hTSC-derived SynT could recruit vascular endothelial cells. Therefore, we conducted an additional vascular endothelial recruiting assay using a SynT-HUVEC co-culture system, comparing undifferentiated hTSC and its SynT derivative.

As a result, SynT derived from hTSCs recruited a significantly higher number of HUVECs compared to undifferentiated cells (Supplementary Fig. 5a), and the endothelial recruitment ability was defective in *PIBF1* KO hTSC-derived SynT (Fig. 2c), which was also seen in *PIBF1* KO BeWo (Fig. 2d). Methodologically, to explain the difference in SynT differentiation protocol between conventional and our co-culture system, we expanded on the associated parts. For example, there was a slight modification in the SynT differentiation medium recipe and the incubation time was optimized for SynT-HUVEC co-culture.

The data have been included in the revised manuscript.

2. Is there evidence that PIBF is secreted into human foetal circulation, facilitating its potential impact on cardiomyocyte differentiation, given the effects observed in vitro (Figure 3)?

→ Thank you for the valuable comment. Based on *in vitro* and *in vivo* studies, we proposed the possible contribution of SynT in placental development and the placenta-heart axis: SynT mediates the formation of the interhaemal membrane by participating as a constituent, recruiting perivascular/vascular cells, and promoting angiogenesis while SynT-secreted PIBF1 impacts cardiovascular development. Identifying SynT-derived sPIBF1 in the developing fetus, especially in the vasculature, is the best way to directly confirm this information. However, as with other areas of obstetrics, there are many ethical and experimental research limitations. In this study, rather than focusing on how PIBF1 enters fetal circulation, we concentrated on the fundamental physiological meaning of PIBF1: where PIBF1 is developmentally required, which cells produce PIBF1, what kind of developmental role PIBF1 has, and ultimately how it works in the placenta-heart axis.

In this regard, as the reviewer indicated, we explored whether sPIBF1 could directly impact cardiomyocyte differentiation. Considering the characteristics of hHO, which is a mixed population of several cell types such as endothelial cells and cardiomyocytes¹⁶, it was not proven in the original manuscript whether the effect of PIBF1 in hHO is due to its direct effect on cardiomyocytes or an indirect effect resulting from enhanced endothelial angiogenesis, which can raise a question of

whether the effect is direct or indirect. To address this, we treated human iPSCs with sPIBF1 (200 ng/ml) during their differentiation into cardiomyocytes in a 2D culture, following the previously described protocol¹⁷. As a result, it was revealed that sPIBF1 significantly increased the number of cTnT and α -actinin-positive cardiomyocytes derived from hiPSCs (Fig. 3c in the revised manuscript), indicating the potential direct impact of PIBF1 on cardiomyocytes during embryo development.

These data and the corresponding methods have been included in the Results and Methods sections of the revised manuscript, respectively.

3. The authors use *Meox2-cre* to generate embryo-specific cKOs, a nice strategy to demonstrate the importance of placental PIBF and the Cre appears to be fully penetrant by E13.5 in the genotyping provided (Supp Figure 9b). To exclude the possibility that early expression of *Pibf* in the embryo is impacting the embryonic phenotype, can the authors report at what stage in development the *Meox2-Cre* become expressed? Is there any evidence that *Pibf* is expressed in any embryonic cell type up to this stage?

- To determine the expression of PIBF1 during embryogenesis, we conducted additional IHC and ISH experiments. *Pibf1* was expressed ubiquitously in all cell types throughout the placenta and embryo at E10.5 (**Fig. R1**). Moreover, the expression profile of *Pibf1* isoforms in whole mouse embryos has been reported at later gestational stages (E12-19)¹⁸.

As the reviewer indicated, since its initial establishment by Tallquist and Soriano, the *Meox2-cre* line has been widely utilized for targeting epiblast-derived cells with unmodified extraembryonic lineages, which is a strategy for demonstrating the developmentally essential role of specific genes in the placenta. *Meox2-cre* activity is first detected at E5.5 in the epiblast, after which it becomes positive in all epiblast derivatives. In the placenta, *Meox2-cre* is expressed in epiblast derivatives, including chorionic plate, yolk sac mesoderm, and fetal-derived endothelial cells, while all other regions of the placenta are negative¹⁹, which is reconfirmed at E9.5 embryo and placenta²⁰. Also, as expected, *Meox2-cre* is not detected in SynTs²¹.

Therefore, considering its spatiotemporal profile of Cre activity (specific in epiblast lineages since E5.5) and manifestations of *Pibf1* KO phenotypes (E10.5), the *Meox2-cre* line is suitable for our study for generating embryo proper-specific *Pibf1* cKO. Likewise, endothelial-specific cre lines used in this study, such as *Tie2-cre*, display their cre activities as early as E7.5²², far before the abnormal manifestation of *Pibf1* KO embryos.

As expected, the *Meox2-cre* line exhibits its activity in the majority of fetal endothelial cells in the placenta²¹. In this regard, to distinguish the endothelial lineage from the whole body of the embryo proper, we further used endothelial-specific cre lines (*Cdh5-* or *Tie2-cre*), demonstrating that the existence of *Pibf1* itself in the embryo proper or cardiovascular system did not affect embryo viability, reinforcing the critical role of SynT-derived PIBF1.

The discussion above was only stated in this Response Letter.

Fig. R1. Ubiquitous expression of Pibf1 in the mouse placenta and embryo at embryonic day (E) 10.5.

a *In situ* hybridization of *Pibf1* using a *Pibf1*-specific probe. Probes detecting mouse *Ppib* and bacterial *dapb* were used as positive and negative controls. Note that the signals (brown dots) were ubiquitously present throughout the placenta and embryo. a, atrium. v, ventricle.
b Immunohistochemistry of *Pibf1* showing ubiquitous signals throughout the placenta and embryo. Sections without primary antibody were used as a negative control. dec, decidua. jz, junctional zone. lab, labyrinth. ra and la, right and left atrium. rv and lv, right and left ventricle.

4. The contrast to the *synb* KO in the discussion is valuable (lines 345-353). However, the lack of embryonic lethality seen in the *synb* KO may not necessarily exclude that PIBF from the *synT* cells is the primary contributor of CHD. It is my understanding that the *synb* KO does not completely lack the *SynT-II* compartment, rather that the cells just fail to fuse but may still differentiate. Hence it is possible in this model, PIBF is still sufficiently produced. Whereas the stainings in the *Pibf* null model suggest that *SynT-II* almost completely fails to differentiate or form appropriately.

- ➔ According to the literature²³ that we discussed, regardless of the different gestational stages of their manifestations, the phenotypes of both *Pibf1* KO and *Synb* KO *SynT-II* were highly similar in terms of their ultrastructures, characterized by impaired fusion in the *SynT-II* layer. The authors also proposed a gap junction-mediated compensatory mechanism to counteract the defective fusion, which explains the absence of highly penetrant growth retardation or *in utero* lethality, which is different from that observed in *Pibf1* KO mice.

This discrepancy may be due to the intrinsic properties of genes essential for syncytialization, which are required in part or in whole for pregnancy. However, this can also be explained by SynT-derived factors, such as PIBF1, which was the main focus of this study.

We demonstrated that the formation of SynT can increase secretory PIBF1 and subsequently induce changes in fetal vasculature (**Fig. 2** in the revised manuscript). This result highlights the importance of syncytialization in trophoblast stem cells, in that without the differentiation of TSCs and formation of SynT, there would not be enough sPIBF1 produced and secreted. Additionally, based on validated data, genetic ablation of Synb may decrease the functionality of trophoblast-secreted factors, including PIBF1, in the undifferentiation of SynT cells.

Minor comment:

1. The authors could choose a more representative figure to show the measured relative increase in cTnT staining in sPIBF-treated hHOs (Figure 3b), as the current image shows no apparent difference.

→ Based on the reviewer's suggestion, we changed the figure.

2. The font in Figures 3e and 3f is so small that it is illegible.

→ As the reviewer suggested, we re-sized the fonts in Fig. 3f and 3g in the revised manuscript (Fig. 3e and 3f in the original manuscript).

Also, it seems that the overall resolution of figures in MS Word manuscripts could be compromised during PDF conversion. We discovered that the image quality significantly decreased after conversion, particularly in images with small text, such as Fig. 3f and 3g, making it difficult to read. We therefore replaced the images with higher resolution. The revised high-resolution images of Fig. 3f and 3g have been included in the revised manuscript. Problems caused by PDF conversion can be addressed through future consultations with the publishing team.

3. Can the authors show full images of the placental sections shown in Figures 4b, 5c, and 5m?

→ As the reviewer requested, the full images of the placental sections in Fig. 4b, 5c, and 5m are shown below. Dotted boxes indicate the region magnified and presented in each original figure.

These images were provided only in this response letter and not in the revised manuscript.

Fig. 4b, lower magnification

Fig. 5c, lower magnification

Fig. 5m, lower magnification

4. It is unclear which placental stages have been analysed throughout Figure 4, the text provides a range of E9.5-E10.5, but the embryos appear to be E10.5. Can the authors please add this to the figure legend and labels?

- ➔ Samples used in the experiments performed throughout Fig. 4 were obtained at E10.5, except for the measurement of fatty acid contents (Fig. 4e), where embryos and placentas were collected at E9.5 and E10.5.

As the reviewer pointed out, insufficient description regarding the embryonic stage could be confusing. We therefore added the term "E10.5" in the title of Fig. 4 and included a statement at the end of its figure legend that reads, "Experimental samples were obtained at E10.5 unless otherwise indicated." This statement was

also included in the figure legends of Fig. 5 and Fig. 6.

Thank you again for your valuable suggestions on our manuscript, which were vital in significantly improving our study as a whole. We hope that our responses, newly conducted experiments, and the corresponding revisions are satisfactory. We would be happy to make any further revisions that you may deem necessary.

References used in this response letter

- 1 Zhou, M. *et al.* Decreased PIBF1/IL6/p-STAT3 during the mid-secretory phase inhibits human endometrial stromal cell proliferation and decidualization. *J Adv Res* **30**, 15-25 (2021).
- 2 Mulac-Jericevic, B., Sucurovic, S., Gulic, T. & Szekeres-Bartho, J. The involvement of the progesterone receptor in PIBF and Gal-1 expression in the mouse endometrium. *Am J Reprod Immunol* **81**, e13104 (2019).
- 3 Huang, B. *et al.* Interleukin-33-induced expression of PIBF1 by decidual B cells protects against preterm labor. *Nat Med* **23**, 128-135 (2017).
- 4 Radford, B. N. *et al.* Defects in placental syncytiotrophoblast cells are a common cause of developmental heart disease. *Nat Commun* **14**, 1174 (2023).
- 5 Okae, H. *et al.* Derivation of Human Trophoblast Stem Cells. *Cell Stem Cell* **22**, 50-63 e56 (2018).
- 6 James, J. L., Lissaman, A., Nursalim, Y. N. S. & Chamley, L. W. Modelling human placental villous development: designing cultures that reflect anatomy. *Cell Mol Life Sci* **79**, 384 (2022).
- 7 Park, J. Y. *et al.* A microphysiological model of human trophoblast invasion during implantation. *Nat Commun* **13**, 1252 (2022).
- 8 Blundell, C. *et al.* Placental Drug Transport-on-a-Chip: A Microengineered In Vitro Model of Transporter-Mediated Drug Efflux in the Human Placental Barrier. *Adv Healthc Mater* **7** (2018).
- 9 Blundell, C. *et al.* A microphysiological model of the human placental barrier. *Lab Chip* **16**, 3065-3073 (2016).
- 10 Lee, J. S. *et al.* Placenta-on-a-chip: a novel platform to study the biology of the human placenta. *J Matern Fetal Neonatal Med* **29**, 1046-1054 (2016).
- 11 Mosavati, B., Oleinikov, A. V. & Du, E. Development of an Organ-on-a-Chip-Device for Study of Placental Pathologies. *Int J Mol Sci* **21** (2020).
- 12 McConkey, C. A. *et al.* A three-dimensional culture system recapitulates placental syncytiotrophoblast development and microbial resistance. *Sci Adv* **2**, e1501462 (2016).
- 13 Nishiguchi, A. *et al.* In vitro placenta barrier model using primary human trophoblasts, underlying connective tissue and vascular endothelium. *Biomaterials* **192**, 140-148 (2019).
- 14 Hemberger, M., Hanna, C. W. & Dean, W. Mechanisms of early placental development in mouse and humans. *Nat Rev Genet* (2019).
- 15 Rusidze, M. *et al.* Estrogen Actions in Placental Vascular Morphogenesis and Spiral Artery Remodeling: A Comparative View between Humans and Mice. *Cells* **12** (2023).
- 16 Lee, S. G. *et al.* Generation of human iPSCs derived heart organoids structurally and functionally similar to heart. *Biomaterials* **290**, 121860 (2022).
- 17 Lee, S. G. *et al.* Development and validation of dual-cardiotoxicity evaluation method based on analysis of field potential and contractile force of human iPSC-derived cardiomyocytes / multielectrode assay platform. *Biochem Biophys Res Commun* **555**, 67-73 (2021).
- 18 Bogdan, A., Polgar, B. & Szekeres-Bartho, J. Progesterone induced blocking factor isoforms in normal and failed murine pregnancies. *Am J Reprod Immunol* **71**, 131-136 (2014).
- 19 Tallquist, M. D. & Soriano, P. Epiblast-restricted Cre expression in MORE mice: a tool to distinguish embryonic vs. extra-embryonic gene function. *Genesis* **26**, 113-115 (2000).
- 20 Yang, F. *et al.* Inositol 1,4,5-trisphosphate receptors are essential for fetal-maternal connection and embryo viability. *PLoS Genet* **16**, e1008739 (2020).
- 21 Sandovici, I. *et al.* The imprinted Igf2-Igf2r axis is critical for matching placental microvasculature expansion to fetal growth. *Dev Cell* **57**, 63-79 e68 (2022).
- 22 Kisanuki, Y. Y. *et al.* Tie2-Cre transgenic mice: a new model for endothelial cell-lineage analysis in vivo. *Dev Biol* **230**, 230-242 (2001).

- 23 Dupressoir, A. *et al.* A pair of co-opted retroviral envelope syncytin genes is required for formation of the two-layered murine placental syncytiotrophoblast. *Proc Natl Acad Sci U S A* **108**, E1164-1173 (2011).

Point-by-point response to reviewer comments

Reviewer #3 (Remarks to the Author):

This study shows that PIBF is indispensable for trophoblast differentiation and syncytiotrophoblast formation. Syncytiotrophoblast and adjacent vascular cells communicate via PIBF, to promote the development of the vascular network in the primary placenta. This process plays a role in the early development of the embryonic cardiovascular system.

This is a very good and extremely meticulous work. The study is well designed the methods are appropriate, the data are interesting and sound. The paper is fairly well written. The conclusions are reasonable, except for those concerning the involvement of the immune system.

- (p. 8) the authors investigate whether the decidual NK cells contributed to the early defects of the *Pibf1*^{-/-} embryo, and found no significant changes either in the % of NK cells among immune cells or their absolute number in *Pibf1*^{+/-} decidua harbouring *Pibf1*^{-/-} embryos. Furthermore, in a Th17-deficient maternal immune background, *Pibf1*^{-/-} embryos still exhibited morphological abnormalities.

They conclude that the maternal immunological environment did not play a role in the early defects of the *Pibf1*^{-/-} embryo. This is likely however, the absolute number or % of decidual NK cells is not meaningful without functional changes (altered lytic activity), which they did not investigate.

➔ Thank you very much for reviewing our study in detail and providing helpful comments.

As the reviewer pointed out, the number or proportion of uNK cells may not accurately reflect their functional state. To support this, we also investigated perforin-positive uNK cells (Supplementary Fig. 9e); however, regardless of the abundance of perforin contents in uNK cells during pregnancy¹, counting only perforin-positive uNK cells may also be insufficient for explaining their functional changes.

To address this issue, we conducted FACS analysis of uNK cells using multiple markers that can represent their functional state. In our study, which explores whether embryonic loss of *Pibf1* affects the uNK population or function, we tested three groups of uNK cells sampled from *Pibf1*^{+/-} decidua including 1) *Pibf1*^{+/+}, 2) *Pibf1*^{+/-}, and 3) *Pibf1*^{-/-} embryos. As a result, there were no significant changes among the three groups in terms of granzyme B (cytotoxic granule contents¹), TNF- α and IFN- γ (pro-inflammatory cytokines²), and CD107a (a proxy for degranulation³) (Supplementary Fig. 9f), demonstrating that embryonic loss of *Pibf1* had no substantial impact on uNK population and function.

We have added the data in the revised manuscript.

- (p.12) the authors addressed whether the loss of *Pibf1* in the maternal immune system affects pregnancy and the offspring. They show that *Pibf1* Δ/Δ ;*Vav1*^{cre/+} females (*Pibf1*-deficient in hematopoietic/immune cells) displayed normal reproductive functions, and their

pups showed normal growth rate, indicating that *Pibf1* is dispensable in the maternal immune system for maintaining pregnancy, at least under an unchallenged condition. This conclusion is not justified.

The fact that hematopoietic/immune cells are PIBF deficient, does not mean that their function cannot be affected by PIBF produced by the neighbouring cells.

Though uterine NK cells do not express progesterone receptors (Henderson et al., Steroid receptor expression in uterine natural killer cells J Clin Endocrinol Metab . 2003; 88(1):440-9.) thus they cannot produce PIBF, they still do contain PIBF in their cytoplasmic granules (Bogdan et al., PIBF positive uterine NK cells in the mouse decidua. J Reprod Immunol. 2017 Feb;119:38-43.), Furthermore in PIBF deficient mice, PIBF+ decidual NK cells decrease in number together with an increased decidual NK activity (Csabai et al., Altered Immune Response and Implantation Failure in Progesterone-Induced Blocking Factor-Deficient Mice. J. Front Immunol. 2020 Mar 11;11:349.)

Knocking out the PIBF receptors on immune cells would be a more relevant approach to investigate the involvement of the maternal immune system.

- Thank you very much for the keen and valuable comment. Previous reports have indicated the pivotal role of PIBF1 in successful pregnancy maintenance and its association with pregnancy-related complications such as miscarriage, pre-eclampsia, and preterm labor⁴⁻⁶, where the roles of PIBF1 are described in the maternal immune system during threatened pregnancy circumstances. However, the *Vav1* cre-mediated cKO strategy in this study was intended to explore the impact of *Pibf1* ablation in maternal hematopoietic/immune cells, which focused on normal unchallenged pregnant conditions unlike previous studies.

Meanwhile, as the reviewer mentioned, pregnant lymphocyte-produced PIBF1, which is induced by progesterone in a PR-dependent manner, suppresses uNK cytotoxicity⁷, and intriguingly, mouse uNK cells also harbor PIBF-positive cytoplasmic granules⁸ despite not expressing PR⁹. However, the uNK cell is not the sole *Pibf1*-deficient immune population in cKO^{*Vav1*} mice, as *Vav1*-cre is expressed in hematopoietic/immune cells¹⁰. In the literature that the reviewer mentioned¹¹, the authors used a pregnant mouse model treated with an antibody against PIBF1 (2 µg intraperitoneally injected at E1.5 and 4.5), resulting in implantation failure and increased resorption rate at E10.5 and, most importantly, increased uNK activity. Intraperitoneal administration of exogenous anti-PIBF1 might have led to masking PIBF1 in the maternal immune system and whole female reproductive tracts, which is quite different from our cKO^{*Vav1*} mice in terms of the *Pibf1* genetic state considering that *Vav1*-cre only targets maternal hematopoietic/immune cells. These differences (harmful/normal pregnant condition, different *Pibf1* depletion strategy) might result in disparate outcomes, so the data in the study by Csabai *et al.*¹⁰ and our cKO^{*Vav1*} data should be interpreted differently.

Moreover, according to our unpublished study, progesterone receptor (*Pgr*)-cre mice, which express Cre activities in the female reproductive tracts including the uterus, showed that *Pgr*^{cre}-mediated *Pibf1* cKO led to female infertility without defects in ovarian functions (e.g., oogenesis and fertilization). Interestingly, the number of implantation sites was not significantly different between control and cKO^{*Pgr*}, and embryos in cKO^{*Pgr*} females died in utero at mid-gestational ages (data

not shown), suggesting a new role for PIBF1 in the maternal reproductive organs as well as immune cells during pregnancy.

Additionally, knocking out PIBF1 receptors on immune cells, as suggested by the reviewer, would be an excellent approach to investigating the role of PIBF1 in the maternal immune system. However, despite the identification of PIBF1 receptor^{12,13}, deletion of the receptor does not result in the embryonic lethal phenotype of *Pibf1* KO¹⁴. These results indicate the existence of a previously unidentified receptor. Therefore, future research is needed to discover new PIBF1 receptors and study their functions.

In order to avoid misunderstanding, the text has been revised as follows:

- “...indicating that *Pibf1* is dispensable in the maternal immune system for maintaining pregnancy, at least under unchallenged conditions.”
- “...indicating that the genetic deletion of *Pibf1* in immune cells does not directly affect the maintenance of pregnancy in mice, at least under unchallenged and physiologically normal maternal conditions.”

In conclusion; this paper shows that the inability of the trophoblast to produce PIBF interferes with organogenesis. This in itself is an important new information, sufficient to merit publication.

- ➔ Thank you again for your valuable suggestions on our manuscript, which were vital in significantly improving our study as a whole. We hope that our responses, newly conducted experiments, and the corresponding revisions are satisfactory. We would be happy to make any further revisions that you may deem necessary.

References used in this response letter

- 1 Wang, F., Qualls, A. E., Marques-Fernandez, L. & Colucci, F. Biology and pathology of the uterine microenvironment and its natural killer cells. *Cell Mol Immunol* **18**, 2101-2113 (2021).
- 2 Perdu, S. *et al.* Maternal obesity drives functional alterations in uterine NK cells. *JCI Insight* **1**, e85560 (2016).
- 3 Whettlock, E. M. *et al.* Dynamic Changes in Uterine NK Cell Subset Frequency and Function Over the Menstrual Cycle and Pregnancy. *Front Immunol* **13**, 880438 (2022).
- 4 Raghupathy, R. *et al.* Progesterone-induced blocking factor (PIBF) modulates cytokine production by lymphocytes from women with recurrent miscarriage or preterm delivery. *J Reprod Immunol* **80**, 91-99 (2009).
- 5 Lim, M. K. *et al.* Characterisation of serum progesterone and progesterone-induced blocking factor (PIBF) levels across trimesters in healthy pregnant women. *Sci Rep* **10**, 3840 (2020).
- 6 Huang, B. *et al.* Interleukin-33-induced expression of PIBF1 by decidual B cells protects against preterm labor. *Nat Med* **23**, 128-135 (2017).
- 7 Szekeres-Bartho, J., Sucurovic, S. & Mulac-Jericevic, B. The Role of Extracellular Vesicles and PIBF in Embryo-Maternal Immune-Interactions. *Front Immunol* **9**, 2890 (2018).
- 8 Bogdan, A., Berta, G. & Szekeres-Bartho, J. PIBF positive uterine NK cells in the mouse decidua. *J Reprod Immunol* **119**, 38-43 (2017).
- 9 Henderson, T. A., Saunders, P. T., Moffett-King, A., Groome, N. P. & Critchley, H. O. Steroid receptor expression in uterine natural killer cells. *J Clin Endocrinol Metab* **88**, 440-449 (2003).
- 10 de Boer, J. *et al.* Transgenic mice with hematopoietic and lymphoid specific expression of Cre. *Eur J Immunol* **33**, 314-325 (2003).

- 11 Csabai, T. *et al.* Altered Immune Response and Implantation Failure in Progesterone-Induced Blocking Factor-Deficient Mice. *Front Immunol* **11**, 349 (2020).
- 12 de la Haba, C. *et al.* Oxidative stress effect on progesterone-induced blocking factor (PIBF) binding to PIBF-receptor in lymphocytes. *Biochim Biophys Acta* **1838**, 148-157 (2014).
- 13 Kozma, N. *et al.* Progesterone-induced blocking factor activates STAT6 via binding to a novel IL-4 receptor. *J Immunol* **176**, 819-826 (2006).
- 14 Mohrs, M. *et al.* Differences between IL-4- and IL-4 receptor alpha-deficient mice in chronic leishmaniasis reveal a protective role for IL-13 receptor signaling. *J Immunol* **162**, 7302-7308 (1999).

REVIEWER COMMENTS

Reviewer #1 (Remarks to the Author):

This revised manuscript by Lee et al entitled "Trophoblast syncytialization maintains the placenta-heart axis" has been marginally improved, although for the most part the authors simply argue most of the points raised and have made minimal changes to their manuscript. Undoubtedly, the authors have performed a large amount of work. However, major concerns remain regarding the over-interpretation of data and the graphical summary.

1) There is NO PROOF that soluble PIBF1 travels from the placenta to the fetus to contribute to angiogenesis and heart formation. What is shown is that Pibf1 is critical for normal placentation, and that the restoration of a normal placenta also rescues the embryonic phenotype.

This is a massive discrepancy in the interpretation of data and must be addressed. This is not to contradict the authors' conclusion that exogenously supplied sPIBF1 may affect cardiomyocyte maturation. But that the assumption that the source of this PIBF1 is in any significant amount coming from the placenta is invalid. The two cell types (trophoblast, cardiomyocytes) are not in direct contact or in diffusional distance to each other. The rebuttal does not address the concerns raised around this point.

2) The diagram must be changed so to take out the PIBF1 circles travelling to the embryo and in particular to the heart. The placental effect on the heart could be mediated by anything; the fact that syncytiotrophoblast cells secrete PIBF1 in vitro does not demonstrate that this protein indeed crosses multiple layers (basement membrane, endothelial cells, in humans also some villous stromal cells) to ever reach the fetal circulation.

3) As mentioned before, the embryo phenotype is extremely severe (lack of turning, developmental stalling at around E8) and goes well beyond some heart defects. A simple rescue of the embryonic heart or vasculature will not restore viability. By the same token of the authors' argument, placental PIBF1 then also affects neurogenesis, overall cell proliferation and pretty much any other aspect of embryogenesis.

Therefore, the heart-centric interpretation must be changed and rephrased. The authors do show that PIBF1 is essential for normal formation of the placental labyrinth. The restoration of this placental layer rescues the lethality – most likely, because it restores normal nutrient and gas provision to the embryo.

For example, this statement in the manuscript is correct: Lines 325ff "Collectively, these results indicate that normally syncytialized trophoblasts and their interaction with endothelium, orchestrated by Pibf1, are prerequisites for establishing a materno-fetal interface, a functional placenta, and, ultimately, embryo survival."

By contrast, this one is not: Lines 381ff "Notably, through the connection between the placenta and embryo circulation, SynT-derived PIBF1 plays an endocrine role in supporting early embryo cardiovascular system development...".

4) VALIDITY OF DATA IN OTHER MODELS: The authors do not address at all whether Pibf1 expression is altered in any other models of the "placenta-heart" axis. Yet a quick look at published data (e.g., Pparg) shows that this is not the case – if anything, Pibf1 is up-regulated in the Pparg KO (Shalom-Barak et al, 2012). Such meta-analyses are essential to include in the Discussion.

5) This sentence is incorrect: Lines 223-233: "We next investigated whether the loss of Pibf1 in trophoblasts affects placentation and subsequent embryogenesis by evaluating Pibf1-null placenta and embryos." Investigation of a constitutive KO does not address whether placental Pibf1 affects

embryogenesis.

6) There is still close to no information on how the KOs were generated. It is highly surprising that the authors obtained 60 hTSC colonies after single cell sorting (hTSCs are extremely difficult to expand from single cells and the clonal derivation rate is low) and moreover, that they obtained full KOs amongst these. How many hTSC KO clones were analyzed? Only a single one? Please specify.

7) There is a clear tendency for decreased SynT-I marker gene expression in the KOs also, this could be mentioned.

8) The Pibf1 expression data shown in the rebuttal should be added to the manuscript.

9) It is still the case that the data display obscures detail– in particular Figure 4, every graph is displayed differently and data points cannot be seen. Also, many of the figures are miniscule.

10) The DiI-guided assessment of fused vs non-fused cells is extremely unreliable and cell membrane staining impossible to appreciate in the figures shown, these data should be performed with an established membrane marker.

11) Fig. 1f, h The WT data of both 2D and 3D time points have been set to 1, this is highly confusing as it obscures the normal trajectory of marker gene expression over differentiation. All data should be normalized only to 2D WT.

12) Line 158-161, the notion that rsPIBF1 is so much better in 'recruiting' HUVEC cells than rPIBF1 is heard to appreciate from the graphs in Fig. 2d, and the $p < 0.001$ significance value seems unlikely and should be verified.

13) Line 182, "embryoid" bodies and following sentence, the "distribution" cannot be "higher", but the authors presumably mean that the protein is more abundant.

Reviewer #2 (Remarks to the Author):

Thank you to the authors for the detailed responses, in particular about the timing and specificity of the Meox2-Cre. In light of this response, I do find that there is insufficient evidence provided to demonstrate that the presence of an intact Pibf1 gene in embryonic cells prior to E5.5 (when the Cre becomes active) is not impacting the phenotype in the cKOs. Single cell data (<https://endoderm-explorer.com/>) shows that Pibf1 is highly expressed in the inner cell mass and epiblast cells before E5.5 (Nowotschin et al. 2019 Nature). Thus it is conceivable that the apparent 'rescue' of the phenotype observed in the Meox2-Cre Pibf1 conditional KOs is in fact because PIBF1 plays a critical role in the specification and identity of embryonic stem cells in the peri-implantation embryo, in addition to a potential role in trophoblast differentiation.

I also echo the concern of reviewer 1: the phenotypes and developmental delay are so severe in the Pibf1^{-/-} embryos that attributing the role of placental PIBF to heart development seems to overlook the profound impairment of development in general. The authors suggest that the lack of rescue of the heart phenotype in the Meox2-Cre Asb2 conditional KO model (described in the response letter) as evidence for the placenta-heart link in the Pibf1^{-/-} despite the developmental abnormalities. However, it is valuable to note that Asb2 is not expressed in early embryos. Hence, differing function in early embryogenesis prior to Cre activation could be an alternative explanation for the apparent discrepancy between these two Meox2-Cre models.

Taken together, I find that the in vivo data does not necessarily support a definitive link between placental PIBF and heart development. And as suggested by other reviewers, the in vitro data

demonstrates an essential role for PIBF in placental development and cardiogenesis, but cannot attribute the source of PIBF, which is widely expressed across embryonic and extra-embryonic cell types and stages. Thus, these developmental processes may in fact be parallel, mutually exclusive dependencies on PIBF. The conclusions should reflect that there can be more than one interpretation of these results, and, in light of this, the singular focus on the connection between placenta and heart may be overstated.

Reviewer #3 (Remarks to the Author):

I thank for the answers for the authors, which I find partly satisfactory.

The authors attempted to assess the function of decidual NK cells by detecting different markers. Perforin and granzyme content of the cells in itself does not say much, the question is, whether it would be released upon encounter with the target cell. I still do not understand, why they did not perform a simple NK cytotoxicity assay.

On the other hand, it is clear, that embryonic loss of PIBF would result in developmental abnormalities -among others of the placenta-heart axis, without substantially affecting the maternal immune system. Therefore, I accept the following statement: embryonic loss of Pibf1 did not have a substantial impact on the population and function of uNK cells.

However, the following statement requires to be further expanded.

"the genetic deletion of Pibf1 in immune cells does not directly affect the maintenance of pregnancy in mice, at least under unchallenged and physiologically normal maternal conditions."

They refer to their own unpublished results showing that "Pgrcre-mediated Pibf1 cKO led to female infertility". This means that along with other systems the immune cells were also PIBF-deficient, and the complete absence of PIBF in the mother resulted in infertility.

Therefore, I suggest to modify this sentence as follows; the genetic deletion of Pibf1 in immune cells does not directly affect the maintenance of pregnancy in mice, however their function might be affected by PIBF produced by other systems, e.g., decidual cells.

Response to Reviewer comments

Reviewer #1 (Remarks to the Author):

This revised manuscript by Lee et al entitled "Trophoblast syncytialization maintains the placenta-heart axis" has been marginally improved, although for the most part the authors simply argue most of the points raised and have made minimal changes to their manuscript. Undoubtedly, the authors have performed a large amount of work. However, major concerns remain regarding the over-interpretation of data and the graphical summary.

1) There is NO PROOF that soluble PIBF1 travels from the placenta to the fetus to contribute to angiogenesis and heart formation. What is shown is that Pibf1 is critical for normal placentation, and that the restoration of a normal placenta also rescues the embryonic phenotype.

This is a massive discrepancy in the interpretation of data and must be addressed. This is not to contradict the authors' conclusion that exogenously supplied sPIBF1 may affect cardiomyocyte maturation. But that the assumption that the source of this PIBF1 is in any significant amount coming from the placenta is invalid. The two cell types (trophoblast, cardiomyocytes) are not in direct contact or in diffusional distance to each other. The rebuttal does not address the concerns raised around this point.

→ We are very grateful to the reviewer for providing critical comments on our manuscript in this second round of revision, and we sincerely hope that our responses and the corresponding revisions may now be deemed acceptable. We also appreciate that the reviewer intended "not to contradict the authors' conclusion that exogenous sPIBF1 may affect cardiomyocyte maturation". Regardless of the intensive criticisms in the second review round, we could sense the reviewer's respect, passion, and insights as an expert in this field.

Despite our efforts to clarify the connection between the placenta and the cardiovascular system during development, our data was insufficient to provide direct evidence regarding the significant concerns, including (1) how PIBF1 is able to cross the multiple layers of the placenta and enter the circulatory system of the embryo, and (2) how SynT-derived PIBF1 directly affects embryonic cardiovascular development, particularly the heart.

PIBF1 is detectable in the circulatory system and urine of non-pregnant or pregnant mothers, and its correlation with pregnancy state has been revealed^{1,2,3}; however, it is largely unknown whether PIBF1 is detectable in embryo/placenta circulation and what its normal/pathological levels are depending on the stage/state/outcome of pregnancy. Unfortunately, there is no literature providing direct evidence of the concerns raised by the reviewers. Additionally, we do not have access to experimental tools to uncover such evidence. Therefore, following the reviewer's suggestion, we carefully reviewed our logic and data again and revised our conclusion as follows:

"Our findings suggest that a healthy placenta with proper SynT is required for normal early embryonic organogenesis, including cardiovascular development. The enhanced manifestation of cardiovascular features seen

in PIBF1-treated heart organoids further implies the possible role of PIBF1 in cardiovascular development.”

We also believe that our current title, “**Trophoblast syncytialization maintains the placenta-heart axis**”, is somewhat broad but does not overstate the conclusion of our findings.

Please review our responses below and let us know if they are appropriate. We would be happy to make any further necessary amendments to the manuscript as deemed necessary by the reviewer.

2) The diagram must be changed so to take out the PIBF1 circles travelling to the embryo and in particular to the heart. The placental effect on the heart could be mediated by anything; the fact that syncytiotrophoblast cells secrete PIBF1 in vitro does not demonstrate that this protein indeed crosses multiple layers (basement membrane, endothelial cells, in humans also some villous stromal cells) to ever reach the fetal circulation.

→ As the reviewer suggested with emphasis, we changed the diagram in Fig. 7, shifting from a single heart-centric stationary interpretation to multiple possibilities that syncytialized trophoblasts can influence embryo development.

Even if detectable levels of PIBF1 are present in maternal blood and placenta during pregnancy, our study does not provide direct evidence that PIBF1 passes through multiple cell layers at the molecular level, as the reviewer stated.

We therefore focused on what could be adequately supported by our data: PIBF1 mediates trophoblast syncytialization, arrangement of vascular/perivascular cells around SynT via SynT-derived PIBF1, and restoration of abnormalities seen in Pibf1 KO placentas (e.g., vascularization and placental barrier formation, nutrient transport, and hematopoiesis) by Pibf1-intact and properly syncytialized trophoblasts within the labyrinth.

3) As mentioned before, the embryo phenotype is extremely severe (lack of turning, developmental stalling at around E8) and goes well beyond some heart defects. A simple rescue of the embryonic heart or vasculature will not restore viability. By the same token of the authors' argument, placental PIBF1 then also affects neurogenesis, overall cell proliferation and pretty much any other aspect of embryogenesis.

Therefore, the heart-centric interpretation must be changed and rephrased. The authors do show that PIBF1 is essential for normal formation of the placental labyrinth. The restoration of this placental layer rescues the lethality – most likely, because it restores normal nutrient and gas provision to the embryo.

For example, this statement in the manuscript is correct: Lines 325ff “Collectively, these results indicate that normally syncytialized trophoblasts and their interaction with endothelium, orchestrated by Pibf1, are prerequisites for establishing a materno-fetal interface, a functional placenta, and, ultimately, embryo survival.”

By contrast, this one is not: Lines 381ff “Notably, through the connection between the placenta and embryo circulation, SynT-derived PIBF1 plays an endocrine role in supporting early embryo cardiovascular system development...”.

- ➔ Thank you again for pointing out the limitations of our study. We acknowledge that there is a discrepancy between our findings (i.e., cardiovascular data in vitro and in vivo) and what we intended to clarify (i.e., a direct impact of SynT-derived PIBF1 on cardiovascular development).

Again, the major concerns raised by reviewers are as follows:

How does PIBF1 cross the multiple layers of the placenta and enter the circulatory system of the embryo, and does SynT-derived PIBF1 directly affect embryonic cardiovascular development, particularly the heart?

To address these issues, we believe that the following experimental tools may be applicable:

- Injecting labeled PIBF (stably labeled with a radioisotope or any other available trafficking system) directly into the labyrinthine layer and quantitatively measuring its presence in cord blood, embryonic circulating blood, and the heart.
- Establishing a mutant line that expresses fluorescence exclusively in trophoblastic PIBF and quantifying its presence in cord blood, embryonic circulating blood, and the heart.

However, the options mentioned above are not currently available. So, following your suggestion, we have made changes and rephrased the interpretation to focus on the heart. For this purpose, although this does not fit the temporal order of our experiments, we relocated the hHO data from Figure 3 to Figure 6 in order to describe the "potential" influence of SynT-derived PIBF1, as there is no direct evidence that SynT-derived PIBF1 could have any impact on cardiogenesis in vivo. In line with this, we have retracted the parts that could cause confusion, such as the suggestion that SynT-derived PIBF1 ultimately controls the cardiovascular development of the embryo. Instead, we attempted to propose the possibility that PIBF1 may play a role in the cardiovascular development of the embryo, at least in part, providing a “thin edge of the wedge” clue to the vast and complex question, “What is responsible for mediating the placenta-heart axis?”

- ➔ In this regard, we implore the reviewer to consider the following point of our study. Our study shows that PIBF1 mediates trophoblast syncytialization and that SynT is a critical participant in organizing and establishing multiple layers of the placental barrier in vitro and in vivo. Placental nutritional transport and gas exchange primarily occur in the labyrinth or villi, which are essential structures for the fetomaternal vascular network. Recently, there have been efforts to clarify the roles of each placental cell, such as SynT, in the placenta-heart axis paradigm⁴. In line with the previous literature⁴ and our data (hHO development enhanced by PIBF in vitro), beyond the widely known and accepted functions of the placenta (e.g., nutritional

supply, gas exchange), we assume that there are other factors derived from the placenta, as previously reviewed by Hemberger et al.⁵.

In the trophoblastic *Pibf1* rescue study using a *Meox2-cre* driver line, the only genetic difference between global KO and *cKOMeox2* embryos was the "trophoblastic *Pibf1* state". Restored nutrient transport is the result of this single genetic modification event. Although we were unable to clearly demonstrate the delivery of SynT-derived PIBF across the placental barrier and its presence in the embryonic circulation and heart, it is evident that the placenta-bearing PIBF-intact and properly syncytialized trophoblasts restored not only the nutrient supply but also the cardiovascular phenotypes of *Pibf1* KO embryos. As the reviewer indicated, "a simple rescue of the embryonic heart or vasculature will not restore viability". Likewise, it appears that a simple rescue of "normal nutrition and gas provision" may not be enough to correct the abnormalities observed in *Pibf1* KO embryos, because PIBF1 plays unique roles in cardiovascular development, as demonstrated by *in vitro* data using vascular/perivascular cells and hHOs.

Multiple cell types and factors are involved in the placenta-heart axis, and our data suggests a bias towards specific compartments. Investigating SynT and its derived factors, such as PIBF1, cannot solely explain the complex paradigm; nevertheless, they may provide some perspective to approach the enormous question, "What is responsible for mediating the placenta-heart axis?"

We therefore incorporated this concept into the first paragraph of the Discussion section as follows:

"PIBF1-intact trophoblast cells led to their syncytialization and played a role in the formation of multiple layers within the interhaemal membrane, which is essential for the proper development of the labyrinthine structure through PIBF1 derived from SynT. Under the influence of the placenta harboring properly syncytialized trophoblasts, the abnormalities observed in Pibf1-null early embryos were restored in vivo, and severe cardiovascular defects were also recovered by the functional placenta. Further in vitro evidence using human heart organoids proposes a possible role of PIBF1 in cardiovascular development. These data collectively indicate the essential roles of SynT in primary placentation and embryo survival, and imply the possible existence of placenta-derived factors that affect embryogenesis, including cardiovascular development."

We have retracted our previous strong opinion in the manuscript, which claimed that SynT-derived PIBF1 directly regulates cardiovascular development in an endocrine manner. Instead, we have shifted our focus to what has been experimentally proven in our study: (1) the rescue of placental and cardiovascular phenotypes in *cKOMeox2* mutants, and (2) the enhancement of cardiovascular features in hHO through PIBF1.

The conclusion we rephrased above does not mean that SynT directly regulates or governs embryonic heart development. Instead, we wanted to point out that the placenta with functional SynT is able to maintain normal embryogenesis, including

cardiovascular development. SynT-derived factors may mediate communication between two distant compartments, and PIBF1 might be one of those factors. However, the unique molecular and physiological properties of PIBF1 make it difficult to address directly, which is why this issue remains elusive. So, even though we were unable to demonstrate it, we may still discuss its potential implications.

The descriptions mentioned above, as well as the limitations of this study, were included in the Results and Discussion sections. We hope that these revisions bring balance to our conclusions so that there would not be an overstatement or an element of confusion.

4) VALIDITY OF DATA IN OTHER MODELS: The authors do not address at all whether *Pibf1* expression is altered in any other models of the “placenta-heart” axis. Yet a quick look at published data (e.g., *Pparg*) shows that this is not the case – if anything, *Pibf1* is up-regulated in the *Pparg* KO (Shalom-Barak et al, 2012). Such meta-analyses are essential to include in the Discussion.

- ➔ The study on PPAR-gamma KO mutants is one of the first pieces of literature to highlight the placenta-heart connection⁶. Microarray analysis revealed increased expression of *Pibf1* in the *Pparg* KO placenta, which is interesting but not in line with our data and without statistical significance.

As the reviewer pointed out, we had initially planned to conduct a meta-analysis of *Pibf1* in various mutant murine models. However, as revealed in this study, the role of placental-derived factors such as PIBF1 is challenging to assess for the following reasons: (1) PIBF1's ubiquitous expression within the placenta, (2) the presence of multiple isoforms is difficult to distinguish, and (3) the functions of the isoforms within and outside trophoblasts. Overall, the meta-analysis of PIBF1 in other mutant models with placental insufficiency is not convincing in explaining the unique properties of PIBF1, as we did not conduct such an analysis.

5) This sentence is incorrect: Lines 223-233: “We next investigated whether the loss of *Pibf1* in trophoblasts affects placentation and subsequent embryogenesis by evaluating *Pibf1*-null placenta and embryos.” Investigation of a constitutive KO does not address whether placental *Pibf1* affects embryogenesis.

- ➔ Thank you for the detailed suggestion. We changed the sentence as follows:

*"We next investigated whether the loss of *Pibf1* affects placentation and subsequent embryogenesis by evaluating *Pibf1*-null placentas and embryos."*

6) There is still close to no information on how the KOs were generated. It is highly surprising that the authors obtained 60 hTSC colonies after single cell sorting (hTSCs are extremely difficult to expand from single cells and the clonal derivation rate is low) and moreover, that they obtained full KOs amongst these. How many hTSC KO clones were analyzed? Only a single one? Please specify.

→ hTSCs were grown in a 35 mm dish, and each dish containing hTSCs was considered a separate "batch." The individual batches (batch n=3-4 per targeted exon 2 and 4, respectively) were transduced with hPIBF1 exon2 or exon4 targeting Cpf1-expressing vectors and then selected with puromycin. The transduced and selected hTSCs in the batch were dissociated from the 35-mm dish using TrypLE and then seeded in three to four 100-mm dishes with the lowest possible cell concentration that allows for single-cell growth. The suitable cell concentration for single-cell colonization varied from batch to batch.

Adherence of a single cell to the surface of a 100-mm dish was confirmed through visual inspection under a phase-contrast microscope after 6-8 h of seeding. The cell was allowed to grow until it formed a visible "colony." The colonies that were macroscopically detectable were then transferred to a 96-well plate, with approximately 10-12 colonies being transferred per 100-mm dish. The cells were further expanded in larger culture vessels.

As a result, 3-4 batches per target (exon 2 or 4) yielded 3-4 100-mm dishes per batch with 10-12 colonies per 100-mm per dish. Theoretically, it is possible to obtain >100 colonies per targeted exon, but not all colonies were grown sufficiently for further expansion. Hence, the number of colonies decreased to n=60 per targeted exon. If the preliminary studies had been included, the number of batches and colonies would have been increased.

During the expansion steps, a certain number of hTSCs were grown in a separate dish to assess the expression of PIBF1 protein through Western blotting. Among them, only the colonies that passed the Western blot screening were subjected to sequence analysis, after which PIBF1 exon 2 or 4 KO was confirmed only in four colonies (#2-21, #2-24, #2-38 in exon2 targeted, #4-26 in exon4 targeted were labelled as KO line #1, 2, 3, and 4 in the revised manuscript, respectively) (Supplementary Fig. 4). Among the four established PIBF1 KO TS lines, we evaluated TS stemness and SynT/EVT differentiation capacity in two lines: KO line #1 (Fig. 1e-m, Supplementary Fig. 3) and #2 (Supplementary Fig. 4b-f).

Supplementary Fig. 4. CRISPR/Cpf1-generated *PIBF1* KO hTSCs are defective in differentiation into EVT and Synt.

➔ Indeed, as the reviewer mentioned, establishing KO lines of hTSC was challenging due to the low induction rate compared to other cell lines. We therefore increased the size of the experiment, which resulted in a tight schedule for arranging batches due to the labor-intensive nature of the experiment. The order of the procedure is summarized in Fig. R1.

Fig. R1. Schematic diagram for generating CRISPR/Cpf1-mediated *PIBF1* KO hTSC cell lines.

➔ The candidate colonies that did not pass gene expression screening by Western blotting were discarded during the expansion step, and the ones that did pass were subjected to sequencing and further expanded (Fig. R1). The time required for the entire generation could be shortened by simultaneously conducting the expansion and knockout screening steps. As the methodological description above was too long, we have summarized the key methodological items for reproducing the

generation of the PIBF1 KO hTSC line in the revised manuscript.

7) There is a clear tendency for decreased SynT-I marker gene expression in the KOs also, this could be mentioned.

→ We appreciate the reviewer's detailed comment. Based on the observed pathological phenotypes in electron microscopy of PIBF1 KO placentas, we expected a decrease in SynT-I markers in PIBF1-deficient placentas. However, no significant difference was observed, as shown in the q-PCR raw data below (Fig. R1a).

Because the Y-axis settings of the graph were confusing the interpretation of SynT-I marker expression, we made the following changes to the graph settings. We normalized the values on the Y-axis, as shown in Figure R2. The mRNA expression data from the *Pibf1* KO placenta were normalized to the WT control. The data for both WT and KO were plotted on separate graphs and then combined into a single graph using GraphPad Prism 9.

a

	+/+				+/-				P value in t-test
Cox2	1.49259	1.05885	0.70989	0.73868	1.74821	0.52092	1.35496	1.66878	0.37133
Eomes	0.86197	0.97704	1.03668	1.12431	0.78642	0.97119	0.88345	0.92974	0.16502
Esrrb	1.10065	1.05807	0.88916	0.95212	1.26765	0.59175	1.57341	1.45264	0.36193
Pif	1.11884	0.95424	0.91219	1.01473	1.27539	0.59539	2.34580	1.93358	0.21346
Pif1	0.98997	0.90030	0.98827	1.12146	1.04289	0.79523	1.94933	1.28770	0.32697
Pif2	1.72095	1.00556	0.59882	0.67469	1.29534	0.47027	2.31217	2.14830	0.30502
Ctcf	1.92668	0.90356	0.60290	0.66769	1.62787	0.57516	1.32390	1.93871	0.40224
Gja3	1.25151	1.14438	0.81042	0.75369	1.09495	0.64156	0.80351	1.22656	0.75356
Pcdh12	1.31046	1.01069	0.84732	0.83153	1.51150	0.92546	1.30897	1.36820	0.14711
Ascl2	1.03973	1.22122	0.82923	0.90983	1.34278	0.63999	1.06004	1.26659	0.68121
Tbipa	1.23113	1.27089	0.82849	0.66950	1.40419	0.52263	1.38980	1.79356	0.40066
Syna	1.22813	1.18055	0.78600	0.82522	0.83483	0.56116	0.42716	1.13304	0.23244
Mct1	0.86298	1.25710	0.88108	0.99884	1.11600	0.70043	0.85334	0.96307	0.49471
Gcm1	1.24174	0.98878	0.87994	0.88954	0.46289	0.39632	0.32768	0.52481	0.00091
Synb	1.27210	0.93588	0.90483	0.88719	0.54477	0.49560	0.34051	0.67985	0.00560
Mct4	1.05017	1.09879	0.93461	0.91644	0.65722	0.87984	0.55348	0.76011	0.01326

Fig. R2. Generating a graph of Fig. 3g in the revised manuscript by combining WT and *Pibf1* KO datasets. **a** Raw data on expressions of markers for TSC and its lineages in WT and *Pibf1* KO placenta at E10.5. **b, c** Graphs generated in GraphPad Prism 9. A graph representing *Pibf1* KO data (b1, c1) was combined with that of WT (b2, c2), generating a combined format (b3, c3).

8) The *Pibf1* expression data shown in the rebuttal should be added to the manuscript.

→ Following the reviewer's suggestion, we added the *Pibf1* expression data as supplementary data (Supplementary Fig. 9) and its associated descriptions in the Results section.

9) It is still the case that the data display obscures detail— in particular Figure 4, every graph is displayed differently and data points cannot be seen. Also, many of the figures are miniscule.

10) The Dil-guided assessment of fused vs non-fused cells is extremely unreliable and cell membrane staining impossible to appreciate in the figures shown, these data should be performed with an established membrane marker.

→ Below are our responses to questions 9 and 10.

■ Regarding the resolution of figures (including Di-8-stained BeWo data)

This issue was also raised in the reviewer's comments during the first-round revision, and we noted that the conversion process from Word to PDF significantly reduced the resolution of the figure components.

Unfortunately, modifying the size of images, increasing their resolution before converting to PDF, or saving them as a PDF file with high image resolution (an additional option provided in MS Word) did not significantly improve the quality of the final PDF version. Ultimately, only the PDF conversion process affected the resolution. We can address the problems caused by PDF conversion through future consultations with the publishing team at the journal. However, in addition to the BeWo fusion images, where the plasma membranes of non-fused or fused cells were stained with Di-8-ANEPPS, a fluorescent dye that is effective in the BeWo cell line^{8,9,10} and was also used in this study, our explanations in the first-round response letter may not have adequately addressed the issue.

The PDF version of our manuscript cannot clearly display the figures while maintaining their appropriate resolution, especially during the revision step, which is quite different from the publishing process, where the publishing team ensures high-resolution image quality. So, we included individual image files of the main figures at a publishable resolution of 300 dpi.

■ Regarding the size and styles of graphs (including data points)

We agree that some of the minor figure panels are rather small, especially in Figure 4. However, when we printed out the figures in their native high resolution (unlike the PDF-converted version in the revision round), there were no issues in appreciating and understanding the contents of each figure panel. Moreover, if the manuscript gets published, the readers will be able to zoom into the high-resolution PDF files so there won't be any issues. Please see the individual images of the main figures enclosed with this response letter.

In terms of graph styles, the graphs were not expressed in a single style because we intended to present data depending on the type of experiments. For example, among the graphs of Fig. 3 in the revised manuscript (Fig. 3c,d,e,f,g,j,l), gross/microscopic morphological assessment data were presented as a bar graph with dots (c,d,f,j), GC-MS of fatty acids data as a “combined” bar graph (e), and the qPCR data as a “combined” box-and-whisker with dots (g,l). This method of data presentation was applied throughout all figures in this manuscript.

Regarding the data points, among the graphs in Fig. 3, 3e is the only one without dots. It is possible that the low resolution of the PDF version might have hindered the visibility of graphs in Fig. 3 other than 3e. In the case of some combined graph styles in this manuscript, data points were omitted if necessary, for example, in Fig. 3e (GC-MS analysis of fatty acids) and Fig. 4j (CFU assay). Please see below:

Fig. R3. Schematic diagram describing how the data in Fig. 3e (a) and Fig. 4j (b) were presented as the combined graph style without data points.

When the combined graphs had a large amount of data (e.g., Fig. R3), graphs were simplified by omitting data points and presenting them as bar graphs with average and SEM, as in the cases of Fig. 3e and Fig. 4j.

Again, we apologize for any inconvenience caused by the difficulty in reading the figures. Considering the descriptions provided above, please review our manuscript

once again, along with the attached image files of the figures.

11) Fig. 1f, h The WT data of both 2D and 3D time points have been set to 1, this is highly confusing as it obscures the normal trajectory of marker gene expression over differentiation. All data should be normalized only to 2D WT.

→ Thank you for the detailed suggestion. We agree that presenting both 2D and 3D differentiation data in a single graph could be more effective. However, we believe that normalizing all data only to 2D WT may not be appropriate for displaying our results. Although 2D and 3D TSC cultures share the same timeline for SynT differentiation (cultured in 2D and 3D SynT differentiation medium for 6 days, respectively), the two experiments were conducted as separate experimental sets as those assays were not intended to compare 2D vs. 3D groups, but rather WT vs. KO groups in 2D and 3D culture, respectively. Therefore, the 2D and 3D datasets could not be analyzed en bloc by normalizing only to 2D WT. Instead, the two datasets should be separated, normalized to each WT, and presented as individual graphs. Therefore, we presented 2D and 3D data separately in Fig. 1f and h. In line with this, 2D and 3D datasets were also separately presented in Fig. 1g, which was applied in the revised manuscript.

12) Line 158-161, the notion that rsPIBF1 is so much better in 'recruiting' HUVEC cells than rPIBF1 is heard to appreciate from the graphs in Fig. 2d, and the p<0.001 significance value seems unlikely and should be verified.

→ We appreciate the reviewer's comments. Along with issue #7 raised by the reviewer, we rechecked the data in Fig. 2d and performed the suggested statistical analysis again. The screenshot below shows the statistical analysis of the dataset in Fig. 2d.

Ordinary one-way ANOVA									
Multiple comparisons									
1	Number of families	1							
2	Number of comparisons per family	10							
3	Alpha	0.05							
4									
5	Tukey's multiple comparisons test	Mean Diff.	95.00% CI of diff.	Below threshold?	Summary	Adjusted P Value			
6	WT vs. KO	86.88	80.77 to 92.99	Yes	****	<0.0001		A-B	
7	WT vs. KO+fPIBF	26.05	19.94 to 32.17	Yes	****	<0.0001		A-C	
8	WT vs. KO+sPIBF	14.93	8.890 to 20.97	Yes	****	<0.0001		A-D	
9	WT vs. KO+f/sPIBF	8.201	2.284 to 14.12	Yes	**	0.0019		A-E	
10	KO vs. KO+fPIBF	-60.82	-67.07 to -54.58	Yes	****	<0.0001		B-C	
11	KO vs. KO+sPIBF	-71.95	-78.12 to -65.77	Yes	****	<0.0001		B-D	
12	KO vs. KO+f/sPIBF	-78.68	-84.73 to -72.63	Yes	****	<0.0001		B-E	
13	KO+fPIBF vs. KO+sPIBF	-11.12	-17.30 to -4.948	Yes	****	<0.0001		C-D	
14	KO+fPIBF vs. KO+f/sPIBF	-17.85	-23.91 to -11.80	Yes	****	<0.0001		C-E	
15	KO+sPIBF vs. KO+f/sPIBF	-6.730	-12.71 to -0.7478	Yes	*	0.0192		D-E	
16									
17	Test details	Mean 1	Mean 2	Mean Diff.	SE of diff.	n1	n2	q	DF
18	WT vs. KO	100.0	13.12	86.88	2.204	24	22	55.75	111
19	WT vs. KO+fPIBF	100.0	73.95	26.05	2.204	24	22	16.72	111
20	WT vs. KO+sPIBF	100.0	85.07	14.93	2.179	24	23	9.692	111
21	WT vs. KO+f/sPIBF	100.0	91.80	8.201	2.134	24	25	5.436	111
22	KO vs. KO+fPIBF	13.12	73.95	-60.82	2.251	22	22	38.21	111
23	KO vs. KO+sPIBF	13.12	85.07	-71.95	2.227	22	23	45.70	111
24	KO vs. KO+f/sPIBF	13.12	91.80	-78.68	2.183	22	25	50.98	111
25	KO+fPIBF vs. KO+sPIBF	73.95	85.07	-11.12	2.227	22	23	7.065	111
26	KO+fPIBF vs. KO+f/sPIBF	73.95	91.80	-17.85	2.183	22	25	11.57	111
27	KO+sPIBF vs. KO+f/sPIBF	85.07	91.80	-6.730	2.157	23	25	4.412	111
28									

Fig. R4. Summary of statistical dataset analysis in Fig. 2d using GraphPad Prism 9.

In the chart, the data labeled as columns A to E represent the WT (A), KO (B), KO+fPIBF (C), KO+sPIBF (D), and KO+f/sPIBF (E) groups, respectively. Therefore, the C-D comparison refers to the comparison between KO+fPIBF and KO+sPIBF. The statistics comparing the two groups in the dataset were highlighted in red. It was confirmed that the significance value between the rsPIBF and rPIBF groups ($P < 0.0001$ in one-way ANOVA followed by Tukey's multiple comparison test) was statistically significant. This was one of the crucial findings that prompted us to investigate the roles of sPIBF in vascular/perivascular cell recruitment.

Additionally, we considered that the issue raised by the reviewer may have originated from the presentation of Fig. 2d, which was visually insufficient to demonstrate the significant difference between the rPIBF and rsPIBF groups. To address this, we modified the graph style by incorporating a y-axis break for improved visualization, as follows:

The revised graph was applied in Fig. 2d of the revised manuscript. Thank you again for reviewing our manuscript in such detail.

13) Line 182, “embryoid” bodies and following sentence, the “distribution” cannot be “higher”, but the authors presumably mean that the protein is more abundant.

→ Thank you for the keen comment. We revised it as follows:

Previous) *the distribution of cTnT, a cardiomyocyte-specific marker, was significantly higher in sPIBF-treated hHOs than in controls (Fig. 3b).*

Revised) *cTnT-positive cardiomyocytes were significantly more abundant in sPIBF-treated hHOs than in controls (Fig. 6b).*

Thank you again very much for providing critically valuable comments for our study as a whole. We tried our best to address your concerns and revise our manuscript accordingly. We hope that our responses and the corresponding revisions are satisfactory. If the reviewer deems that further changes are necessary, we would be glad to comply in order to ensure that our manuscript is considered appropriate for publication.

Reviewer #2 (Remarks to the Author):

Thank you to the authors for the detailed responses, in particular about the timing and specificity of the Meox2-Cre. In light of this response, I do find that there is insufficient evidence provided to demonstrate that the presence of an intact Pibf1 gene in embryonic cells prior to E5.5 (when the Cre becomes active) is not impacting the phenotype in the cKOs. Single cell data (<https://endoderm-explorer.com/>) shows that Pibf1 is highly expressed in the inner cell mass and epiblast cells before E5.5 (Nowotschin et al. 2019 Nature). Thus it is conceivable that the apparent 'rescue' of the phenotype observed in the Meox2-Cre Pibf1 conditional KOs is in fact because PIBF1 plays a critical role in the specification and identity of embryonic stem cells in the peri-implantation embryo, in addition to a potential role in trophoblast differentiation.

→ Thank you for reviewing our manuscript and response letter in such detail.

As the reviewer indicated, our manuscript lacked a thorough discussion of the temporal discrepancy between Pibf1 expression in the early embryo and the periods when Meox2-cre becomes active. Embryo outgrowth culture demonstrated that Pibf1-null embryos displayed unchanged implantation potential with normal ICM/TE morphological specification in vitro (Supplementary Fig. 7b in the revised manuscript) and normal gross phenotypes until E8.5 (Supplementary Fig. 7c in the revised manuscript), which was not relevant to our study. Subsequently, the central point we addressed in our in vivo study was whether the extraembryonic Pibf1 affects the developmental process of trophoblast syncytialization in the placenta and subsequent embryogenesis.

However, the intracellular events in peri-implantation embryos that can affect the fate decisions of ESCs and subsequent lineages cannot be determined by gross phenotype detection alone. For example, in the case of fibroblast growth factor (FGF)/FGF receptor (FGFR) signaling, the use of Fgfr1 and Fgfr2 siRNA-treated mouse blastocysts showed impairment in the formation of the primitive endoderm lineage, but the ratio of ICM/TE remained unchanged¹¹ despite its essential role in the maintenance of pluripotency in ESCs¹².

The absence of this type of in-depth discussion led to a somewhat biased interpretation of the data, causing us to overlook the potential roles of Pibf1 in the blastocyst stage. We therefore included this as a limitation of this study in the Discussion section of the revised manuscript.

I also echo the concern of reviewer 1: the phenotypes and developmental delay are so severe in the Pibf1^{-/-} embryos that attributing the role of placental PIBF to heart development seems to overlook the profound impairment of development in general. The authors suggest that the lack of rescue of the heart phenotype in the Meox2-Cre Asb2 conditional KO model (described in the response letter) as evidence for the placenta-heart link in the Pibf1^{-/-} despite the developmental abnormalities. However, it is valuable to note that Asb2 is not expressed in early embryos. Hence, differing function in early embryogenesis prior to Cre activation could be an alternative explanation for the apparent discrepancy between these two Meox2-Cre models.

Taken together, I find that the in vivo data does not necessarily support a definitive link between placental PIBF and heart development. And as suggested by other reviewers, the in vitro data demonstrates an essential role for PIBF in placental development and cardiogenesis, but cannot attribute the source of PIBF, which is widely expressed across embryonic and extra-embryonic cell types and stages. Thus, these developmental processes may in fact be parallel, mutually exclusive dependencies on PIBF. The conclusions should reflect that there can be more than one interpretation of these results, and, in light of this, the singular focus on the connection between placenta and heart may be overstated.

→ We appreciate the reviewer's criticism and agree with the suggestion.

We acknowledge that mentioning the *Asb2* KO mutant as an example in the first-round response letter was somewhat inappropriate. After considering the valuable suggestion from the reviewer, we have taken into account the temporal discrepancy between the gene expression pattern in early embryos and the *Meox2*-cre activity window, which should be interpreted differently from the *Pibf1* case, as the reviewer pointed out.

Despite our efforts to clarify the relationship between the placenta and the embryonic cardiovascular system during development, we must admit that our perspective was somewhat limited. We proposed a new role for syncytialized trophoblasts and their derived PIBF1. However, our data were insufficient to provide direct evidence for the major concerns raised by the reviewers, which should be clarified in vivo or clinically:

PIBF1 is detectable in the circulatory system and urine of non-pregnant or pregnant mothers, and its correlation with pregnancy state has been revealed^{1,2,3}; however, it is largely unknown whether PIBF1 is detectable in embryo/placenta circulation and what its normal/pathological levels are depending on the stage/state/outcome of pregnancy. Unfortunately, there is no literature providing direct evidence of the concerns raised by the reviewers. Additionally, we do not have access to experimental tools to uncover such evidence. Therefore, following the reviewer's suggestion, we carefully reviewed our logic and data again and revised our conclusion as follows:

"Our findings suggest that a healthy placenta with proper SynT is required for early embryonic normal organogenesis, including cardiovascular development. The enhanced manifestation of cardiovascular features seen in PIBF1-treated heart organoids further implies the possible role of PIBF1 in cardiovascular development."

We also believe that our current title, "**Trophoblast syncytialization maintains the placenta-heart axis**", is somewhat broad but does not overstate the conclusion of our findings.

Moreover, although this does not fit the temporal order of our experiments, we relocated the hHO data from Figure 3 to Figure 6 in order to describe the "potential" influence of SynT-derived PIBF1, as there is no direct evidence that SynT-derived

PIBF1 could have any impact on cardiogenesis in vivo.

We added the *Pibf1* expression data, which was presented in the first-round response letter, as a supplement in the revised manuscript (Supplementary Fig. 9) and its associated descriptions in the Results section due to its essential state in building up our storyline, as the reviewer indicated.

In line with this, we have retracted the parts that could cause confusion, such as the suggestion that SynT-derived PIBF1 ultimately controls the cardiovascular development of the embryo. Instead, we attempted to propose the possibility that PIBF1 may play a role in the cardiovascular development of the embryo, at least in part, providing a “thin edge of the wedge” clue to the vast and complex question, “What is responsible for mediating the placenta-heart axis?”

We incorporated this concept into the first paragraph of the Discussion section as follows:

*“PIBF1-intact trophoblast cells led to their syncytialization and played a role in the formation of multiple layers within the interhaemal membrane, which is essential for the proper development of the labyrinthine structure through PIBF1 derived from SynT. Under the influence of the placenta harboring properly syncytialized trophoblasts, the abnormalities observed in *Pibf1*-null early embryos were restored in vivo, and severe cardiovascular defects were also recovered by the functional placenta. Further in vitro evidence using human heart organoids proposes a possible role of PIBF1 in cardiovascular development. These data collectively indicate the essential roles of SynT in primary placentation and embryo survival. They imply the possible existence of placenta-derived factors that affect embryogenesis, including cardiovascular development.”*

We have retracted our previous strong opinion in the manuscript, which claimed that SynT-derived PIBF1 directly regulates cardiovascular development in an endocrine manner. Instead, we have shifted our focus to what has been experimentally proven in our study: (1) the rescue of placental and cardiovascular phenotypes in cKOMEox2 mutants, and (2) the enhancement of cardiovascular features in hHO through PIBF1.

Instead, we wanted to point out that the placenta with functional SynT is able to maintain normal embryogenesis, including cardiovascular development. SynT-derived factors may mediate communication between two distant compartments, and PIBF1 might be one of those factors. However, the unique molecular and physiological properties of PIBF1 make it difficult to address directly, which is why this issue remains elusive. So, even though we were unable to demonstrate it, we may still discuss its potential implications.

According to the revised conclusion above, we made changes to the diagram in Fig. 7, shifting from a single heart-centric stationary interpretation to multiple possibilities that syncytialized trophoblasts can influence embryo development. We therefore focused on what could be adequately supported by our data: PIBF1 mediates

trophoblast syncytialization, arrangement of vascular/perivascular cells around SynT via SynT-derived PIBF1, and restoration of abnormalities seen in Pibf1 KO placentas (e.g., vascularization and placental barrier formation, nutrient transport, and hematopoiesis) by Pibf1-intact and properly syncytialized trophoblasts within the labyrinth.

The descriptions mentioned above, as well as the limitations of this study, were included in the Results and Discussion sections of the revised manuscript. We hope that these revisions bring balance to our conclusions so that there would not be an overstatement or an element of confusion.

Thank you again very much for providing critically valuable comments for our study as a whole. We tried our best to address your concerns and revise our manuscript accordingly. We hope that our responses and the corresponding revisions are satisfactory. If the reviewer deems that further changes are necessary, we would be glad to comply in order to ensure that our manuscript is considered appropriate for publication.

Reviewer #3 (Remarks to the Author):

I thank for the answers for the authors, which I find partly satisfactory.

The authors attempted to assess the function of decidual NK cells by detecting different markers. Perforin and granzyme content of the cells in itself does not say much, the question is, whether it would be released upon encounter with the target cell. I still do not understand, why they did not perform a simple NK cytotoxicity assay.

On the other hand, it is clear, that embryonic loss of PIBF would result in developmental abnormalities -among others of the placenta-heart axis, without substantially affecting the maternal immune system. Therefore, I accept the following statement: embryonic loss of Pibf1 did not have a substantial impact on the population and function of uNK cells.

- Thank you very much for reviewing our study in detail and providing valuable comments. As the reviewer indicated, we did not perform the NK cytotoxicity assay, which the reviewer suggested using to estimate functional changes in uNK cells. Based on the preliminary publications on the role of PIBF1 in regulating immune function, we made efforts to identify immunological differences in our initial studies using conventional PIBF1 KO embryos/placentas. However, after conducting an investigation, no significant differences were found in histopathological observations related to immune cells and various cytokine-related analyses that regulate the cytotoxicity of NK cells. As a result, the study took a different direction.

We kindly ask for the reviewer's understanding, and once again, we appreciate their agreement despite the limitations of our study.

However, the following statement requires to be further expanded.

“the genetic deletion of Pibf1 in immune cells does not directly affect the maintenance of pregnancy in mice, at least under unchallenged and physiologically normal maternal conditions.”

They refer to their own unpublished results showing that “Pgrcre-mediated Pibf1 cKO led to female infertility”. This means that along with other systems the immune cells were also PIBF-deficient, and the complete absence of PIBF in the mother resulted in infertility.

Therefore, I suggest to modify this sentence as follows; the genetic deletion of Pibf1 in immune cells does not directly affect the maintenance of pregnancy in mice, however their function might be affected by PIBF produced by other systems, e.g., decidual cells.

- We appreciate the reviewer's detailed comments and highly agree with the suggestion. Our cKOVav1 data could be viewed as conflicting with the previously well-known fact that PIBF1 plays essential roles in the maternal immune system if the different experimental conditions used in this study were not thoroughly considered. However, we believe that the reviewer has offered a valuable solution to this concern.

Following your suggestion, we have modified the sentence as follows:

“...indicating that the genetic deletion of Pibf1 in immune cells does not directly affect the maintenance of pregnancy in mice. Meanwhile, the

function of immune cells might also be affected by PIBF1, which can be produced by other cells or tissues. For example, PIBF1 is highly expressed in endometrial stromal cells during the mid-secretory phase of the endometrium and is required for their decidualization¹³. Therefore, further investigation is needed to understand the role of PIBF1 in maintaining pregnancy in maternal environments.”

Thank you again for the reviewer's valuable. We hope that our responses and the corresponding revisions are satisfactory.

References cited in the response letter

1. Polgar, B., Nagy, E., Miko, E., Varga, P. & Szekeres-Bartho, J. Urinary progesterone-induced blocking factor concentration is related to pregnancy outcome. *Biol Reprod* **71**, 1699-1705 (2004).
2. Hudic, I. et al. Maternal serum progesterone-induced blocking factor (PIBF) in the prediction of preterm birth. *J Reprod Immunol* **109**, 36-40 (2015).
3. Lim, M. K., Ku, C. W., Tan, T. C., Lee, Y. H. J., Allen, J. C. & Tan, N. S. Characterisation of serum progesterone and progesterone-induced blocking factor (PIBF) levels across trimesters in healthy pregnant women. *Sci Rep* **10**, 3840 (2020).
4. Radford, B. N. et al. Defects in placental syncytiotrophoblast cells are a common cause of developmental heart disease. *Nat Commun* **14**, 1174 (2023).
5. Hemberger, M., Hanna, C. W. & Dean, W. Mechanisms of early placental development in mouse and humans. *Nat Rev Genet*, (2019).
6. Barak, Y. et al. PPAR gamma is required for placental, cardiac, and adipose tissue development. *Mol Cell* **4**, 585-595 (1999).
7. Shalom-Barak, T. et al. Placental PPARgamma regulates spatiotemporally diverse genes and a unique metabolic network. *Dev Biol* **372**, 143-155 (2012).
8. Zhang, Y. & Yang, H. A simple and robust fluorescent labeling method to quantify trophoblast fusion. *Placenta* **77**, 16-18 (2019).
9. Zhang, Y. et al. TMEM16F phospholipid scramblase mediates trophoblast fusion and placental development. *Sci Adv* **6**, eaba0310 (2020).
10. Zhang, Y. et al. Functional coupling between TRPV4 channel and TMEM16F modulates human trophoblast fusion. *Elife* **11**, (2022).
11. Krawczyk, K., Wilczak, K., Szczepanska, K., Maleszewski, M. & Suwinska, A. Paracrine interactions through FGFR1 and FGFR2 receptors regulate the development of preimplantation mouse chimaeric embryo. *Open Biol* **12**, 220193 (2022).
12. Mossahebi-Mohammadi, M., Quan, M., Zhang, J. S. & Li, X. FGF Signaling Pathway: A Key Regulator of Stem Cell Pluripotency. *Front Cell Dev Biol* **8**, 79 (2020).
13. Zhou, M. et al. Decreased PIBF1/IL6/p-STAT3 during the mid-secretory phase inhibits human endometrial stromal cell proliferation and decidualization. *J Adv Res* **30**, 15-25 (2021).

REVIEWERS' COMMENTS

Reviewer #1 (Remarks to the Author):

This is a second revision of the manuscript by Lee et al entitled "Trophoblast syncytialization maintains the placenta-heart axis"

Although fundamentally remaining unchanged, the manuscript has been improved by amendments to the text that tone down the previously over-stated role of placental PIBF1 in heart development. The diagram in Figure 7, in particular, has been critically improved.

It is truly disappointing, however, that even on second review the authors simply choose to ignore a significant number of comments.

Largely, this revised version reflects the findings in a more accurate light.

Remaining necessary changes:

All individual data points must be visible. The unchanged display of graphs, despite repeated requests, is unacceptable.

Line 151-152: CTBs do not "act" as trophoblast stem cells. A small fraction of first trimester CTB cells retains trophoblast stem and progenitor cell properties that allows derivation of hTSC lines in vitro.

Line 162: "...implying a potential role of trophoblast-derived PIBF1 in placental development"
Delete this part of the sentence. The mere fact that some factor is produced by trophoblast cells does not imply a function.

Lines 443-448 "However,requires further investigation"

Replace with: However, in this study we did not prove that placentally produced PIBF1 enters the fetal blood circulation to directly exert effects on the developing fetal heart. Thus, it is possible that the dysfunctional placenta of Pibf1 mutants affects heart development in ways unrelated to secreted PIBF1.

Line 518-520: "After selection with puromycin (2 µg/ml; A1113803, 519 Gibco) for 48 h, single-cell colonies (n=60 of each targeted exon) were selected after seeding them in a 96-well plate."

There is absolutely no way that this statement is correct for human TSCs. How and when were these cells seeded into 96-well plates, how were colonies picked, how was it confirmed that they were single-cell derived? With a doubling time of 20-28h in optimal conditions (Okae et al 2018), a single cell would merely have produced 4 cells after 48h. This statement alone casts immense doubt on the entire experimental rigor of this study.

Reviewer #2 (Remarks to the Author):

I find the authors responses and additions to the abstract, results and discussion sections suitably address the reviewers' comments. I appreciate the experimental limitations in trying to further demonstrate that SynT-produced PIBF is exerting a function on cardiogenesis in vivo, and I think the additional paragraph on study limitations is well-discussed. While I agree that the title is broad and does not necessarily overstate the findings, I would suggest that the title be revised for publication, to better represent the study findings.

Reviewer #3 (Remarks to the Author):

The authors have addressed my comments in a satisfactory manner.

Point-by-point responses to Reviewers' Comments

Reviewer #1 (Remarks to the Author):

This is a second revision of the manuscript by Lee et al entitled "Trophoblast syncytialization maintains the placenta-heart axis"

Although fundamentally remaining unchanged, the manuscript has been improved by amendments to the text that tone down the previously over-stated role of placental PIBF1 in heart development. The diagram in Figure 7, in particular, has been critically improved.

It is truly disappointing, however, that even on second review the authors simply choose to ignore a significant number of comments.

Largely, this revised version reflects the findings in a more accurate light.

Remaining necessary changes:

All individual data points must be visible. The unchanged display of graphs, despite repeated requests, is unacceptable.

→ Thank you again for providing important comments on our study. We apologize for not fully meeting the requirements in terms of the display of graphs in the previous revision. We changed the graph styles according to the requests from the editorial team of the journal and supplied the Source Data files for all data presented within the figures.

Line 151-152: CTBs do not "act" as trophoblast stem cells. A small fraction of first trimester CTB cells retains trophoblast stem and progenitor cell properties that allows derivation of hTSC lines in vitro.

→ As you suggested, we changed the sentence as follows:

"In the developing human placenta, a small fraction of mononuclear cytotrophoblasts (CTBs) retain the properties of trophoblast stem cells (TSCs) and differentiate into multinucleated syncytiotrophoblasts (SynT), ~."

Line 162: '...implying a potential role of trophoblast-derived PIBF1 in placental development"

Delete this part of the sentence. The mere fact that some factor is produced by trophoblast cells does not imply a function.

→ We deleted the sentence following your suggestion.

Lines 443-448 "However,requires further investigation"

Replace with: However, in this study we did not prove that placentally produced PIBF1 enters the fetal blood circulation to directly exert effects on the developing fetal heart. Thus, it is possible that the dysfunctional placenta of Pibf1 mutants affects heart development in ways unrelated to secreted PIBF1.

→ We replaced the paragraphs following your suggestion.

Line 518-520: "After selection with puromycin (2 µg/ml; A1113803, 519 Gibco) for 48 h, single-cell colonies (n=60 of each targeted exon) were selected after seeding them in a 96-well plate."

There is absolutely no way that this statement is correct for human TSCs. How and when were these cells seeded into 96-well plates, how were colonies picked, how was it confirmed that they were single-cell derived? With a doubling time of 20-28h in optimal conditions (Okae et al 2018), a single cell would merely have produced 4 cells after 48h. This statement alone casts immense doubt on the entire experimental rigor of this study.

→ To avoid misinterpretation, we added the following sentence:

"After selection with puromycin (2 µg/ml; A1113803, Gibco) for 48 h, the transduced and selected cells were dissociated and seeded in 100-mm dishes at the lowest possible cell concentration that allows for single-cell growth. The cell was then allowed to grow until it formed a visible colony, and the single-cell colonies (n=60 for each targeted exon) were selected after being seeded in a 96-well plate."

Reviewer #2 (Remarks to the Author):

I find the authors responses and additions to the abstract, results and discussion sections suitably address the reviewers' comments. I appreciate the experimental limitations in trying to further demonstrate that SynT-produced PIBF is exerting a function on cardiogenesis in vivo, and I think the additional paragraph on study limitations is well-discussed. While I agree that the title is broad and does not necessarily overstate the findings, I would suggest that the title be revised for publication, to better represent the study findings.

→ Thank you very much for reviewing our study in detail and providing valuable comments. We revised the title following your suggestion: "PIBF1 regulates trophoblast syncytialization and promotes cardiovascular development"

Reviewer #3 (Remarks to the Author):

The authors have addressed my comments in a satisfactory manner.

→ Thank you very much for reviewing our study in detail and providing valuable comments.